# RoboInter: A Holistic Intermediate Representation Suite Towards Robotic Manipulation

**Hao Li**[1,2,∗]  **Ziqin Wang**[3,2,∗]  **Zi-han Ding**[4]  **Shuai Yang**[5]  **Yilun Chen**[2,‡]  **Yang Tian**[2]
**Xiaolin Hu**[6]  **Tai Wang**[2]  **Dahua Lin**[7]  **Feng Zhao**[1,†]  **Si Liu**[3,†]  **Jiangmiao Pang**[2,†]
[1]University of Science and Technology of China, [2]Shanghai Artificial Intelligence Laboratory,
[3]Beihang University, [4]Nanyang Technological University, [5]Zhejiang University,
[6]Tsinghua University, [7]The Chinese University of Hong Kong
**Project page:** https://lihaohn.github.io/RoboInter.github.io

## ABSTRACT

Advances in large vision-language models (VLMs) have stimulated growing interest in vision-language-action (VLA) systems for robot manipulation. However, existing manipulation datasets remain costly to curate, highly embodiment-specific, and insufficient in coverage and diversity, thereby hindering the generalization of VLA models. Recent approaches attempt to mitigate these limitations via a plan-then-execute paradigm, where high-level plans (e.g., subtasks, trace) are first generated and subsequently translated into low-level actions, but they critically rely on extra intermediate supervision, which is largely absent from existing datasets. To bridge this gap, we introduce the ***RoboInter Manipulation Suite***, a unified resource including data, benchmarks, and models of **intermediate representations** for manipulation. It comprises ***RoboInter-Tool***, a lightweight GUI that enables semi-automatic annotation of diverse representations, and ***RoboInter-Data***, a large-scale dataset containing over 230k episodes across 571 diverse scenes, which provides dense per-frame annotations over more than 10 categories of intermediate representations, substantially exceeding prior work in scale and annotation quality. Building upon this foundation, ***RoboInter-VQA*** introduces 9 spatial and 20 temporal embodied VQA categories to systematically benchmark and enhance the embodied reasoning capabilities of VLMs. Meanwhile, ***RoboInter-VLA*** offers an integrated plan-then-execute framework, supporting modular and end-to-end VLA variants that bridge high-level planning with low-level execution via intermediate supervision. In total, RoboInter establishes a practical foundation for advancing robust and generalizable robotic learning via fine-grained and diverse intermediate representations.

## 1 INTRODUCTION

The remarkable generalization of large language models (LLMs) and vision-language models (VLMs) through large-scale pretraining has inspired efforts to extend this paradigm to robotics, giving rise to end-to-end vision-language-action (VLA) models (Brohan et al., 2023; Kim et al., 2024; Black et al., 2024; Bjorck et al., 2025; Lu et al., 2025). Although web-scale multimodal data enables broad semantic reasoning, existing large-scale robot datasets (et al., 2023; Khazatsky et al., 2024; Wu et al., 2024; Bu et al., 2025) remain costly and tightly coupled to specific embodiments despite massive efforts in data collection (Fu et al., 2024b), leaving a significant gap in generalization.

To address this gap, recent research has explored frameworks that separate planning from execution. **Modular approaches** (Huang et al., 2023; Belkhale et al., 2024; Huang et al., 2024a; Liu et al., 2024a; Nasiriany et al., 2024) infer high-level structures before translating them into low-level actions, enabling better generalization but often relying on rule-based design. Meanwhile, many **end-to-end VLAs** (Zhou et al., 2025b; Yang et al., 2025b; Zawalski et al., 2024; Shi et al., 2025; Lin et al., 2025; Deng et al., 2025; Cen et al., 2025; Wu et al., 2025; Du et al., 2023) introduce intermediate representations (e.g., subtasks, visual traces, 2D/3D grounding, etc.) as extra input conditions

---

∗Equal contributions. ‡Project leader. †Corresponding authors.

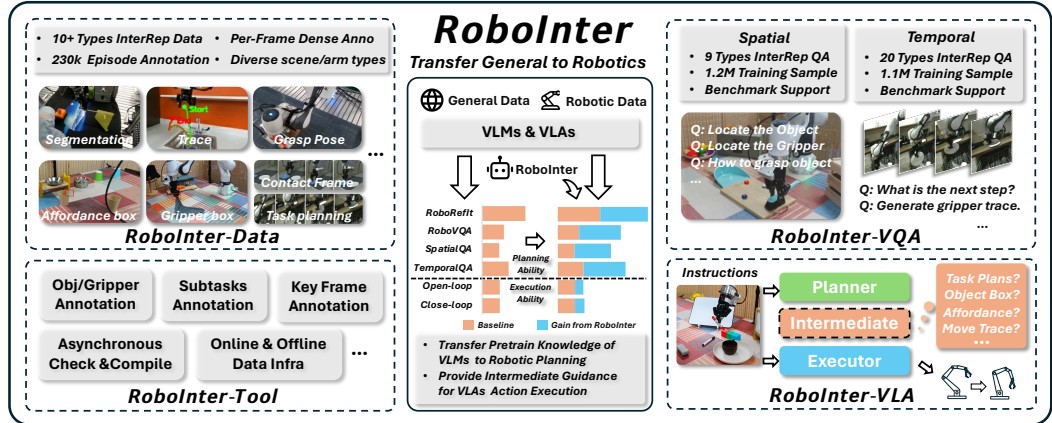

Figure 1: **RoboInter manipulation suite** includes annotation tools, annotated data, curated VQA dataset, and their applications in VLMs and VLAs. RoboInter provides a dataset with over 230k episodes (mainly from Droid (Khazatsky et al., 2024) and RH20T (Fang et al., 2023)) and 10+ types of intermediate representation annotations, named *RoboInter-Data*; a curated embodied VQA benchmark and dataset covering 29 spatial- and temporal-level categories, *RoboInter-VQA*; and an integrated *plan-then-execute* framework for training VLM and VLA models, *RoboInter-VLA*.

or supervision signals. Despite their differences, both directions converge on introducing planning as an intermediate representation, which we summarize as the ***plan-then-execute*** paradigm.

The effectiveness of this paradigm critically depends on high-quality intermediate representations. Existing datasets (et al., 2023; Khazatsky et al., 2024) typically pair visual inputs with overall instructions and robot actions, but they rarely provide the fine-grained intermediates required for ***plan-then-execute***. Collecting and annotating new data remains costly and infrastructure-intensive, which fails to leverage the advantages of community-driven open-source data and makes it unavailable for large-scale pretraining. Recent efforts (Li et al., 2025b; Yuan et al., 2024) have explored automated annotation for existing datasets, yet with limited success. For example, LLARVA (Niu et al., 2024) leverages a pretrained gripper detector to generate large-scale traces, but it is sensitive to distribution shift. ECoT (Zawalski et al., 2024) annotates pseudo-intermediate textual planning with object grounding via Gemini (Team, 2023). ShareRobot (Ji et al., 2025) combines automated annotation with manual verification, but only at a small scale and with labels misaligned to step-wise actions. Overall, current datasets lack large-scale, high-quality annotations, which limits their value for advancing research on intermediate representations for VLMs and ***plan-then-execute*** VLAs.

To address this gap, we propose the ***RoboInter Manipulation Suite***, illustrated in Figure.1. Built on **RoboInter-Tool**, a lightweight GUI for semi-automatic per-frame annotation of embodied videos, we introduce **RoboInter-Data**, a large-scale, per-frame annotation dataset of intermediate representations for robotic manipulation. As shown in Table.1, *RoboInter-Data* provides over 230k episodes across 571 distinct scenes, surpassing LLARVA (Niu et al., 2024), ECoT (Zawalski et al., 2024), and ShareRobot (Ji et al., 2025) in both scale and diversity. Unlike prior datasets that either cover a limited number of scenes (Chen et al., 2024a) or rely solely on automatic annotation (Li et al., 2025b; Zawalski et al., 2024), *RoboInter-Data* uniquely combines large-scale coverage with human-in-the-loop verification. To the best of our knowledge, this is the first real-world manipulation dataset that provides dense, per-frame annotations of intermediate representations spanning more than ten categories, including subtasks, primitive skills, segmentation masks, gripper/object bounding boxes, placement proposals, affordance boxes, grasp poses, traces, and contact points. All annotations are temporally synchronized with executed actions and robot states, together with two-view observations (one third-person and one wrist-view camera), enabling end-to-end action learning.

Leveraging these fine-grained annotations, we introduce **RoboInter-VQA**, which exposes and benchmarks existing VLMs from the perspectives of embodied generation and understanding, comprising 9 spatial and 20 temporal embodied-scene VQA categories to enhance the embodied abilities of VLMs. Built upon the high-level VLM planner trained on these curated VQA data, we introduce **RoboInter-VLA**, an integrated plan-then-execute framework that supports both modular and end-

Table 1: **Comparison of embodied annotations datasets.** *Emb.-VQA* denotes the availability of curated embodied VQA benchmarks and datasets; *E2E-ACT* indicates whether the dataset temporally aligned with executed actions; *Curated-CoT* specifies multi-intermediate chain-of-thought support.

| Dataset | #Video | #Scene | Dense | Emb. -VQA | E2E -ACT | Curated -CoT | Subtask &Skill | Affor- -dance | Contact Point | Gripper Box | Object Box | Trace | Annotation Type |
|---|---|---|---|---|---|---|---|---|---|---|---|---|---|
| LLARVA | – | 311 | ✗ | ✗ | ✓ | ✗ | ✗ | ✗ | ✗ | ✗ | ✗ | ✓ | Auto |
| Hamster | 136k | – | ✓ | ✓ | ✓ | ✗ | ✗ | ✗ | ✗ | ✗ | ✗ | ✓ | Auto |
| RH20T-P | 38k | 7 | ✗ | ✓ | ✗ | ✗ | ✓ | ✗ | ✓ | ✗ | ✗ | ✗ | Human+Auto |
| ECoT | 60k | 12 | ✓ | ✗ | ✓ | ✓ | ✓ | ✗ | ✗ | ✓ | ✓ | ✗ | Auto |
| AgiBot-World | 1M | 106 | ✓ | ✗ | ✓ | ✗ | ✓ | ✗ | ✗ | ✗ | ✗ | ✗ | Human |
| VLA-OS | 10k | – | ✓ | ✗ | ✓ | ✓ | ✓ | ✓ | ✓ | ✓ | ✓ | ✓ | Auto |
| ShareRobot | 51k | 102 | ✗ | ✓ | ✗ | ✗ | ✓ | ✓ | ✓ | ✗ | ✗ | ✓ | Human+Auto |
| Robo2VLM | 176k | 463 | ✗ | ✓ | ✗ | ✗ | ✓ | ✗ | ✓ | ✗ | ✗ | ✓ | Auto |
| VeBrain | 12k | – | ✓ | ✓ | ✗ | ✗ | ✓ | ✗ | ✓ | ✗ | ✗ | ✗ | Human |
| **Ours** | **230k** | **571** | ✓ | ✓ | ✓ | ✓ | ✓ | ✓ | ✓ | ✓ | ✓ | ✓ | **Human+Auto** |

to-end VLA variants, enabling rapid adaptation from planning to execution, and then we systematically investigate the impact of intermediate representations on the generalization and controllability of VLA variants. Through extensive experiments, we show that RoboInter-Data substantially improves the reasoning and grounding capabilities of VLM planners, particularly in understanding and generating various embodied intermediate representations for manipulation, thereby strengthening VLMs' foundational embodied abilities. Open- and closed-loop evaluations of manipulation tasks further demonstrate that the pretrained planners and our intermediate representation data provide performance and generalization gains to the VLAs. By analyzing the trade-offs among different VLA variants, we establish a unified foundation for leveraging these data in future research. We release RoboInter to the community, including its corresponding data, benchmarks, and models, and hope they can pave the way for applications of intermediate representations for Embodied AI.

## 2 RELATED WORKS

**Embodied intermediate representation.** Embodied intermediate representations have been widely explored. Prior work leverages 2D trace (Gu et al., 2023), optical flow (Xu et al., 2024), subtasks (Zhang et al., 2024; Belkhale et al., 2024), key points (Wen et al., 2023), future images (Zhao et al., 2025; Lv et al., 2025; Cai et al., 2026) and grounding box (Sundaresan et al., 2023; Huang et al., 2025) to guide action generation. While some representations (e.g., grounding box and flow) can be estimated by vision foundation models (Ravi et al., 2024; Dong & Fu, 2024), the results are not always reliable across different embodied scenes. Web-scale resources remain under-aligned with manipulation tasks, so embodied alternatives such as RoboBrain (Team et al., 2025) and Ve-Brain (Luo et al., 2025) curate task-specific embodied datasets and train VLMs to generate intermediate representations. Another line (Li et al., 2025b; Yuan et al., 2024) collects large-scale datasets annotated with a single type of intermediate representation and leverages them to generate actions.

**Robotic manipulation datasets.** Numerous prior works (Fang et al., 2023; Khazatsky et al., 2024; et al., 2023; Gao et al., 2025b; Tian et al., 2025) have introduced real-world or simulated robot datasets encompassing diverse sources, scenarios, and skills. While these data do not provide native intermediate representation labels for real-world manipulation tasks, subsequent efforts have provided some annotations. RH20T-P (Chen et al., 2024a) augments RH20T with primitive-level labels of subtasks, RT-H (Belkhale et al., 2024) provides language descriptions of grasp poses and motions. LLaRVA (Niu et al., 2024) and Hamster (Li et al., 2025b) leverage gripper tracking to generate large-scale 2D trajectories for pretraining. High-level planning approaches, such as Robo2VLM (Chen et al., 2025a), sample frames and then supply diverse VQA annotations to enhance embodied scene understanding. RoboInter introduces ***per-frame*** dense annotations data across varied intermediate representations to advance both embodied understanding and end-to-end action learning.

**Embodied planning for execution.** Embodied planning is important in diverse and complex real-world settings. ***Plan-then-execute*** systems can be categorized into implicit and explicit forms. Implicit methods operate as black boxes, where these VLA methods (Black et al., 2024; Li et al., 2023a; 2025a) primarily rely on implicit reasoning by fine-tuning pretrained VLMs of various pretraining paradigms. Explicit methods generate interpretable intermediate representations and directly uti-

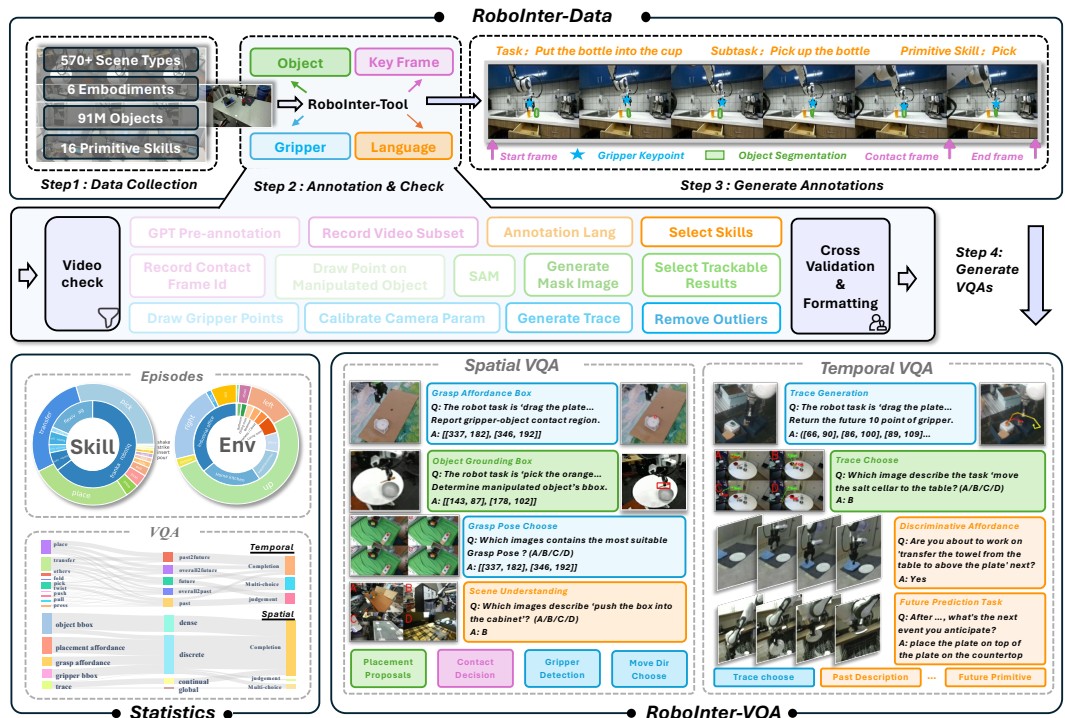

Figure 2: **Overview of RoboInter-Data and RoboInter-VQA.** We collect and annotate 230k manipulation episodes to obtain 10 types of intermediate representation annotations through Data Collection and Annotation & Check. We further construct a large-scale, diverse set of VQA spanning spatial and temporal dimensions. Statistics of raw episodes and the curated VQA are also provided.

lize the knowledge from VLMs, such as $\pi_{0.5}$ (Intelligence et al., 2025), and Rekep (Huang et al., 2024b). ECoT (Zawalski et al., 2024) introduces explicit CoT into VLA by prompting a VLM to autoregressively generate both CoT text and discrete action tokens. VLA-OS (Gao et al., 2025a) further explores various model designs to combine planning and action generation. We offer large-scale intermediate representation data and the pretrained VLM Planners, jointly enabling both implicit and explicit reasoning in VLA variants through efficient information aggregation.

## 3 DATASET

### 3.1 ROBOINTER-DATA

As illustrated in Figure 2, RoboInter-Data builds upon extensive manipulation datasets and provides large-scale, high-quality annotations of diverse intermediate representations.

**Data collection.** To enhance dataset diversity, we collected two types of raw manipulation data: (1) *In-the-Wild setting* (i.e., diverse indoor scenarios), emphasizing the diversity of scenes and instructions, mainly collected from Droid (Khazatsky et al., 2024) and OXE (et al., 2023). (2) *Table-Top setting* (i.e., tabletop interaction scenarios), highlighting the high quality and skills diversity, collected from RH20T (Fang et al., 2023). By integrating raw teleoperated video recordings of these datasets, followed by rigorous screening and pre-processing, we constructed a high-quality, large-scale database consisting of 230k manipulation episodes (third-person videos) in total.

**Annotations & Check with RoboInter-Tool.** For accurate and comprehensive labels, *RoboInter-Tool* is employed to perform the following annotations: *(1) Task decomposition & key-frame annotation.* Each manipulation video is decomposed into clips using 15 predefined primitive skills, and ChatGPT (OpenAI, 2023) is employed to produce preliminary references for language annotations. Human annotators utilize *RoboInter-Tool* to segment video clips, assigning each clip to a primitive skill, and simultaneously completing clip-level and video-level language annotations. The *contact*

*frame* where the robot arm contacts the manipulated object is also recorded. *(2) Recognizing the manipulated object.* After recording the object that the robot arm interacts with, *RoboInter-Tool* automatically transports the annotation to *SAM2* (Ravi et al., 2024) for object segmentation and tracking, and the result is asynchronously returned for review. A re-annotation and inspection mechanism reduces the impact of segmentation or tracking errors on the quality. *(3) Locating the end-effector.* As many raw recordings lack reliable camera parameters, directly projecting 3D coordinates to obtain accurate 2D end-effector traces is often infeasible. We estimate a calibration matrix to improve projection accuracy and complement parameter-missing episodes via gripper detection and point tracking, enabling reliable reconstruction of the 2D trace. Details are provided in Appendix A.6.

**Post-processed annotations.** By reorganizing the above annotations, we derive additional intermediate representations: *(1) Grasp annotation.* The grasp affordance box is inferred from the 2D end-effector location at the annotated contact frame. The contact points are the pre-defined key points of the gripper at the moment of contacting, while the corresponding robot state (i.e., the 6D end-effector pose) defines the grasp pose. *(2) Placement annotation.* The object position at the end of the subtask is treated as the target placement location. *(3) Gripper annotation.* Anchor points enclosing the gripper are identified and projected from 3D to 2D using camera parameters and robot states, forming the gripper bounding box on the 2D image coordinates.

**Statistics of RoboInter-Data.** As shown in Figure.2, we provide high-quality, scene-diverse intermediate representation annotations for 230k manipulation episodes. This dataset includes 6 types of robot arms, 571 types of scenes, and 15 types of primitive skills. With the assistance of the RoboInter-Tool, we produce nearly 61M-frame object grounding annotations, about 70M-frame gripper trace annotations, 190k affordance box and placement proposal annotations, and nearly 760k language clip annotations. Despite this scale, the annotations maintain high quality.

## 3.2 ROBOINTER-VQA

**VQA task construction.** As illustrated in the RoboInter-VQA section of Figure.2, we convert the annotations into diverse VQA tasks to enhance VLM capabilities. Tasks are organized along two axes: (i) intermediate representation type (spatial vs. temporal) and (ii) target capability (understanding vs. generation). *(1) Spatial VQA for understanding.* We design three selection tasks and one judgment task to train spatial comprehension, including selecting the correct object bounding box or grasp pose, matching scenes to instructions, and determining whether contact occurs. *(2) Spatial VQA for generation.* It includes five prediction tasks that require generating spatial intermediate representations for downstream execution and complete specific content based on spatial reasoning, which includes object bounding box, grasp pose, placement proposal, key points, and the gripper bounding box. *(3) Temporal VQA for understanding.* To evaluate how the VLMs understand motion traces and the relationships between subtasks and observations, we design selection tasks and judgment tasks for temporal information. We design five selection tasks for movement directions of grippers, the matching of trace and description, subtask/primitive discrimination, and execution stage identification. Four judgment tasks assessing task success and next-step feasibility. *(4) Temporal VQA for generation.* We formulate tasks that require trace generation and multi-step planning under varying levels of contextual completeness. Prompts condition on different amounts of prior information (e.g., past subtasks or overall instructions) and ask the model to predict the subsequent steps or multi-step planning. Video-based visual inputs are also used to summarize past events and predict feasible next steps. Trace generation is evaluated under both easy and challenging settings (with or without initial waypoints). Details are included in the Appendix.A.7.3.

**Statistics of RoboInter-VQA.** Our VQA data is also large in scale, comprising approximately 1M spatial generation entries, 172k spatial understanding entries, 131k temporal generation entries, and 935k temporal understanding entries. To prevent information leakage between training and validation, we carefully designate 7,246 videos as the evaluation pool with the remaining data used for training, and sample validation sets for each question category from this pool.

## 4 ROBOINTER-VLA

In this section, as illustrated in Figure.3, we present **RoboInter-VLA**, a family of models following a **plan-then-execute** paradigm, consisting of a *Planner* and an *Executor*. Rather than a monolithic

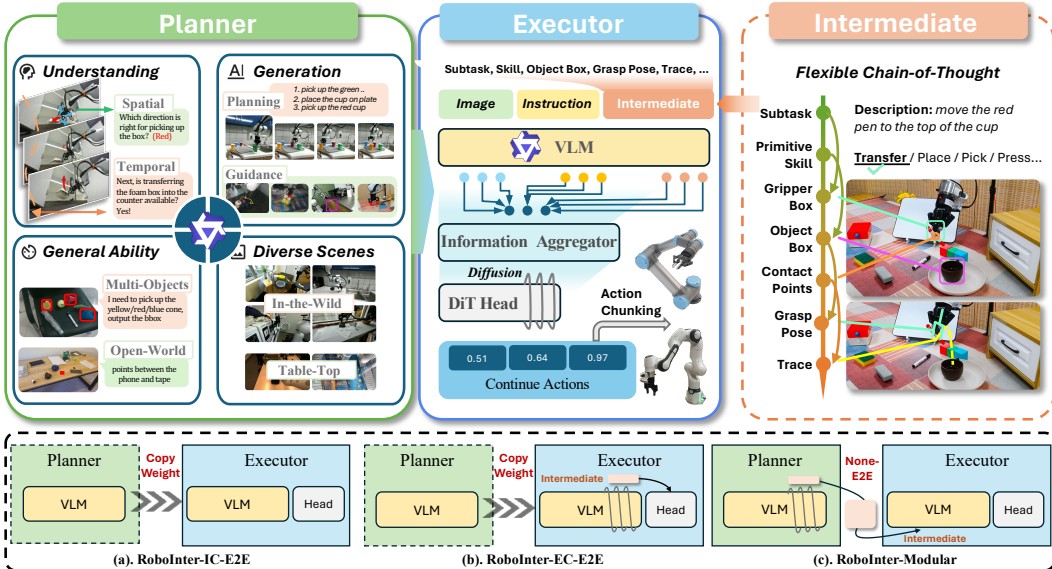

Figure 3: **Framework of RoboInter-VLA.** Our model follows a ***plan-then-execute*** paradigm with a VLM-based *Planner* and an *Executor*. The Planner exhibits enhanced understanding and generation for manipulation, strong general grounding abilities, and robust perception across diverse scenes. The Executor shares the VLM backbone with the Planner. Three variants are supported, and intermediate representations in Flexible Chain-of-Thought (F-CoT) bridge planning and execution.

design, RoboInter-VLA supports multiple variants and enables flexible adaptation from planning to execution. The *Planner* performs high-level decision-making by combining general and embodied reasoning to produce intermediate representations, which guide the *Executor* in translating multi-modal observations and language instructions into low-level actions.

## 4.1 MODEL ARCHITECTURE

**VLMs as Planner.** The Planner model acquires embodied capabilities through a visual question answering formulation with a co-training strategy. To capture both spatial and temporal information, we adopt VLM architectures that support single- and multi-image inputs, including the Qwen-VL series (Wang et al., 2024b) and LLaVA-One-Vision (Li et al., 2024a). Each model consists of a base LLM, a vision encoder, and an MLP-based vision–language projector. The Planner generates outputs autoregressively and is optimized using a cross-entropy loss.

**VLAs as Executor.** To systematically and lightweightly derive from the Planner, we build Executor on a Qwen2.5-VL backbone with a Diffusion Transformer (DiT) action head (Peebles & Xie, 2023). We further utilize an *information aggregator* that gathers the hidden states of all input and output tokens, as well as intermediate representations, and compresses them into conditioning features with a controllable length. The *Executor* consumes multi-view visual observations (e.g., primary and wrist), language instructions, and intermediate representations (based on primary observation), and produces multi-step action chunks via a diffusion loss.

## 4.2 PLAN-THEN-EXECUTE PARADIGMS

As shown in Figure.3, by leveraging the pretrained Planner, we provide three paradigms to enhance downstream action execution: (1) *RoboInter-IC-E2E (Implicitly-Conditioned End-to-End)*, which directly injects the VLM from a pretrained Planner into the end-to-end Executor, using it as a stronger vision-language feature extractor. This approach can yield robust embodied perception and more accurate task-relevant visual cues. (2) *RoboInter-EC-E2E (Explicitly-Conditioned End-to-End)*, where the Executor is initialized with the VLM of the Planner and jointly optimizes both intermediate-representation reasoning and action generation. (3) *RoboInter-Modular (Modular Planner-to-Executor)*, a non-E2E hierarchical design that treats the Planner and Executor as

Table 2: **Performance comparison on third-party benchmarks.** Including **Embodied**, **Grounding**, and **General** benchmarks for general VLMs (upper) and embodied VLMs (lower).

| Model Name | Embodied | | | Grounding | | | General | | | | | |
|---|---|---|---|---|---|---|---|---|---|---|---|---|
| | Where-2place↑ | RoboRefIt-test↑ | Robo-VQA↑ | Refcoco-g-val↑ | Refcoco+val↑ | Refcoco-val↑ | Text-VQA↑ | CO-CO↑ | OCR-bench↑ | MME↑ | MM-VET↑ | POPE↑ |
| InternVL3-1B | 2.65% | 7.0% | 30.5 | 79.8% | 73.2% | 83.0% | 75.1 | 23.7 | 798 | 1907 | 58.9% | 90.7% |
| InternVL3-2B | 1.86% | 27.5% | 27.7 | 87.6% | 84.0% | 85.8% | 77.0 | **27.9** | 835 | 2186 | 62.2% | 89.6% |
| InternVL3-8B | 1.95% | 27.7% | 27.9 | **89.6%** | **88.2%** | **92.5%** | 80.2 | 26.5 | 880 | **2410** | **81.3%** | 91.1% |
| QwenVL2.5-3B | 11.8% | 68.9% | 37.6 | 85.2% | 82.4% | 89.1% | 79.3 | 15.7 | 797 | 2175 | 61.8% | 85.9% |
| QwenVL2.5-7B | 18.9% | 75.8% | 38.4 | 87.2% | 84.2% | 90.2% | **84.9** | 15.0 | 864 | 2306 | 67.1% | 85.9% |
| LLaVA-OV-7B | 7.9% | 10.4% | 31.4 | 71.9% | 69.7% | 73.8% | 71.1 | 8.4 | **882** | 2307 | 67.3% | 86.4% |
| RoboBrain-2.0-3B | 59.8% | 30.9% | 30.6 | 55.0% | 51.5% | 50.9% | 81.0 | 27.2 | 811 | 2126 | 59.4% | 88.1% |
| RoboBrain-2.0-7B | 63.6% | 8.8% | 31.6 | 62.9% | 70.1% | 76.1% | 75.9 | 25.2 | 857 | 2076 | 61.4% | 86.2% |
| RoboInter-Qwen-3B | 58.3% | 80.0% | 43.3 | 87.9% | 85.8% | 89.5% | 78.9 | 15.6 | 787 | 2180 | 61.0% | 90.5% |
| RoboInter-Qwen-7B | 65.8% | 85.6% | 74.4 | 88.4% | 86.6% | 91.5% | 83.0 | 15.9 | 832 | 2281 | 62.3% | **91.4%** |
| RoboInter-LLaVAOV-7B | **66.3%** | **89.3%** | **74.5** | 87.3% | 84.2% | 91.3% | 72.2 | 15.8 | 725 | 2217 | 61.4% | 90.4% |

independent modules. During training, the Executor conditions on ground-truth intermediate representations to generate actions, whereas during inference it uses the predicted results of the Planner.

**Flexible chain-of-thought for intermediate representations.** To support the explicitly conditioned and modular architectures, we introduce **F-CoT**, a chain-of-thought composed of multiple intermediate representations. F-CoT plays two roles: (i) as VQA supervision for training the Planner, and (ii) as action-aligned guidance for the Executor. In *RoboInter-IC-E2E*, the VLM generates F-CoT content, which is directly consumed by the DiT head. In *RoboInter-Modular*, the Planner produces the F-CoT content and the Executor conditions on it. F-CoT flexibly combines representations such as subtasks, skills, object bounding boxes, affordance boxes, motion traces, etc., in textual or visual form, allowing users to select subsets tailored to specific embodied tasks. We denote textual F-CoT as *RoboInter-Te-Modular* and visual-prompted F-CoT as *RoboInter-Im-Modular*.

## 5 BENCHMARKING AND EXPERIMENTS

### 5.1 BENCHMARKING THE PLANNER

**Enhanced grounding and embodied capability.** As shown in Table 2, we evaluated on third-party spatial reasoning benchmarks, including Where2Place (Yuan et al., 2024) and RoboRefIt (Lu et al., 2023) (spatial point and grounding reasoning) and RoboVQA (Sermanet et al., 2024) (temporal task planning). Across all three benchmarks, our models substantially outperformed the base models (Qwen2.5-VL-3B/7B (Wang et al., 2024b) and LLaVA-OneVision-7B (Li et al., 2024a)). Notably, RoboBrain2.0 (Team et al., 2025) is also an embodied VLM (i.e., Planner). At the 3B scale, *RoboInter-Qwen-3B* achieved a 49.1% improvement over RoboBrain2.0 on RoboRefIt and a 12.7% improvement on RoboVQA. At the 7B scale, the corresponding gains reached 76.8% and 42.8%, respectively. For grounding, all three RoboInterVLM variants exceeded their respective base models on Refcoco (Lin et al., 2014). Particularly, *RoboInter-Qwen-7B* eventually ranked second overall, with a 27.4% relative improvement over RoboBrain2.0-7B. On general benchmarks, our models remained relatively stable on most benchmarks, indicating that ***our curated VQA data enhances the abilities of embodied reasoning and grounding***, meanwhile general capabilities of our VLMs are slightly affected.

**RoboInter-VQA benchmark at the spatial and temporal level.** As shown in Table 3, for spatial-based generation tasks, closed-source and general VLMs without embodied experience rarely produce accurate intermediates (typically below 40%), underscoring the importance of additional intermediate representation annotations. On simpler questions, Gemini-2.5-Flash and RoboBrain-2.0-7B lead on *Grasp Pose (choice)* with 32.7% and 23.3% ACC. For *Grounding Choice*, LLaVA-OV-7B achieves 31.9%, while Gemini-2.5-Flash is strongest at 69.4%; most other models remain near random choosing (25%) given limited understanding of manipulation scenes. For temporal, closed-source API and general VLMs largely fail to generate future traces or task planning; RoboBrain-2.0 attains a much better DTW in *Trace Generation* than Qwen-VL-2.5 (541 v.s. 1702). On *Visual Trace Choice*, Gemini-2.5-Flash remains competitive (49.4%). For *Planning Choice* and *T/F of Task Plan-*

Table 3: **Results of RoboInter-VQA *spatial* and *temporal* benchmark. G.D.** means grounding, **A.F.** denotes Affordance. Spatial generation uses ACC@IOU>0.1 (%↑), multiple choice and T/F use ACC (%↑). For temporal, Trace uses DTW (↓), other metrics use ACC or average BLEU (↑).

| Model Name | RoboInter-VQA Spatial | | | | | | | RoboInter-VQA Temporal | | | | |
| | Generation | | | | Multiple Choice | | T/F | Generation | | Multiple Choice | | T/F |
| | Object G.D.↑ | Grasp A.F.↑ | Place A.F.↑ | Gripper G.D.↑ | Grasp Pose↑ | G.D. Choice↑ | Contact↑ | Trace↓ | Task Planning↑ | Visual Trace↑ | Planning Choice↑ | Task Planning↑ |
|---|---|---|---|---|---|---|---|---|---|---|---|---|
| QwenVL2.5-3B | 46.6% | 12.2% | 34.1% | 6.1% | 21.1% | 21.9% | 50.9% | 2712 | 20.3 | 37.5% | 60.0% | 59.7% |
| QwenVL2.5-7B | 51.2% | 14.7% | 38.2% | 10.2% | 27.3% | 25.7% | 52.5% | 1702 | 22.4 | 39.0% | 64.5% | 60.5% |
| InternVL3-1B | 7.8% | 2.3% | 8.3% | 1.2% | 24.8% | 25.9% | 50.4% | – | 10.5 | 28.9% | 54.9% | 55.8% |
| InternVL3-2B | 20.6% | 3.1% | 17.9% | 1.9% | 25.5% | 27.3% | 50.1% | – | 7.7 | 35.2% | 59.3% | 59.4% |
| InternVL3-8B | 32.7% | 5.9% | 28.2% | 3.5% | 25.1% | 31.1% | 52.9% | 1035 | 8.1 | 34.0% | 71.5% | 60.0% |
| Llava-OV-7B | 25.8% | 5.5% | 23.7% | 1.6% | 24.5% | 31.9% | 54.4% | – | 11.0 | 37.7% | 44.9% | 63.5% |
| GPT4o-mini | 6.8% | – | 7.2% | 1.1% | 10.9% | 16.8% | 53.6% | 1736 | 14.7 | 28.4% | 66.6% | 63.9% |
| Gemini-2.5-flash | 1.7% | – | 1.2% | – | 32.7% | 69.4% | 65.5% | – | – | 49.4% | – | – |
| RoboBrain-2.0-3B | 15.2% | – | – | 2.8% | 25.5% | 26.5% | 50.4% | 595 | 16.0 | 29.7% | 48.2% | 46.8% |
| RoboBrain-2.0-7B | – | – | – | 2.5% | 23.3% | 21.5% | 49.2% | 541 | 15.3 | 29.5% | 57.8% | 46.4% |
| RoboInter-Qwen-3B | 76.1% | 34.9% | 52.7% | 61.6% | 74.0% | 73.4% | 75.2% | 332 | 61.2 | 78.8% | 82.2% | 88.7% |
| RoboInter-Qwen-7B | 75.1% | 37.8% | **56.9%** | 62.0% | **76.1%** | 75.7% | 75.6% | 323 | **63.4** | 81.9% | **86.5%** | **93.0%** |
| RoboInter-LlavaOV-7B | **82.9%** | **46.3%** | 55.1% | **70.1%** | 74.1% | **79.7%** | **76.3%** | **299** | 62.7 | 81.9% | 81.8% | 83.9% |

*ning*, as planning aligns closely with general LLM abilities, most models transfer common-sense knowledge and show better performance. Overall, ***current closed-source and general VLMs typically lack enough embodied abilities***. Curated from diverse annotations, ***RoboInter-VQA markedly improves the VLM abilities of understanding and generating intermediate representations***.

## 5.2 OPEN-LOOP EVALUATION OF THE EXECUTOR

**Experimental settings**. In this section, we examine how a pretrained Planner improves the Executor and compare different VLA paradigms. As our annotated corpus spans more than 500 distinct scenarios, comprehensive real-world validation across all scenarios is infeasible, as emphasized by HPT (Wang et al., 2024a). Following the discrete token-accuracy evaluation in Open-VLA (Kim et al., 2024), we utilize an ***Open-Loop Score (OLS)*** to evaluate the generation of *continuous action chunks*, in which per-step actions are assessed independently and compared with the ground-truth actions. OLS is computed as the average value over 100K transitions from evaluation videos, ensuring statistical stability. More details in Appendix.A.4.3. Nine Executor variants are evaluated: (a).*Vanilla*: omits any pretrained VLM from Planner, performing action learning only; (b-e).*RoboInter-IC-E2E*, *EC-E2E*, *Te-Modular* and *Im-Modular* are stated in Section.4.2; (f).*Oracle+Executor*: not end-to-end, both training and inference are guided by GT intermediate representations; (g).*QwenVL+Executor*: training is GT-guided, and inference employs intermediates from original Qwen2.5VL; (h).*VLA-OS*: we train VLA-OS (Gao et al., 2025a) in our setting and use it as an additional baseline; (i).*Oracle+VLA-OS*: not end-to-end, VLA-OS are guided by GT intermediate representations. Two open-loop evaluation settings: (1) *In-the-Wild*: focus on scene and object generalization, we compare the convergence performance under identical training steps. (2) *Table-Top*: focus on the tabletop environment and cross-embodiment ability, we mainly examine the evaluation curve during training. The annotated corpus is divided into *In-the-Wild* and *Table-Top* subsets. We sample approximately 10% of episodes from each subset for Executor training (25k in total), of which 8% are reserved for evaluation.

**Planner consistently improves the Executor's action generation**. The *In-the-Wild* setting is shown in Table.4. The *Vanilla* achieves a lower mOLS score than IC-E2E (0.3086 v.s. 0.3218), which incorporates intermediate representations, indicating that ***pretrained VLM Planner can enhance the learning capability of the VLA Executor***. The mOLS score of *EC-E2E* is higher than *IC-E2E* (0.3340 v.s. 0.3218), showing that the ***explicit intermediate representations are more helpful for action guidance than the implicit***. For *Oracle+Executor*, the non-E2E architectures, utilizing ground-truth annotation in the Executor, achieve substantially the highest scores. This indicates that ***our annotations are informative and stable***. *Te-Modular* surpasses *EC-E2E* (0.3543 v.s. 0.3340), implying that ***decoupling planning and execution facilitates dedicated optimization*** of each capability and mitigates mode conflict. *Im-Modular* performs slightly worse than *Te-Modular* (0.3439

Table 4: **Open-loop evaluation in In-the-Wild setting**. We report OLS with different error thresholds (@0.1 to @0.01) and the mean value.

| Method | Open-Loop Score (OLS) | | | | mOLS |
|---|---|---|---|---|---|
| | @0.1 | @0.05 | @0.03 | @0.01 | - |
| VLA-OS | 0.6180 | 0.3905 | 0.1928 | 0.0129 | 0.3035 |
| Vanilla | 0.6793 | 0.3608 | 0.1753 | 0.0189 | 0.3086 |
| RoboInter-IC-E2E | 0.6984 | 0.3810 | 0.1873 | 0.0204 | 0.3218 |
| RoboInter-EC-E2E | 0.7049 | 0.3930 | 0.2066 | 0.0314 | 0.3340 |
| QwenVL+Executor | 0.6749 | 0.3582 | 0.1777 | 0.0298 | 0.3102 |
| RoboInter-Te-Modular | 0.7124 | 0.4133 | 0.2332 | 0.0584 | 0.3543 |
| RoboInter-Im-Modular | 0.7056 | 0.4029 | 0.2240 | 0.0430 | 0.3439 |
| Oracle+VLA-OS | 0.7260 | 0.4928 | 0.2734 | 0.0200 | 0.3780 |
| Oracle+Executor | 0.7511 | 0.4640 | 0.2705 | 0.0587 | 0.3861 |

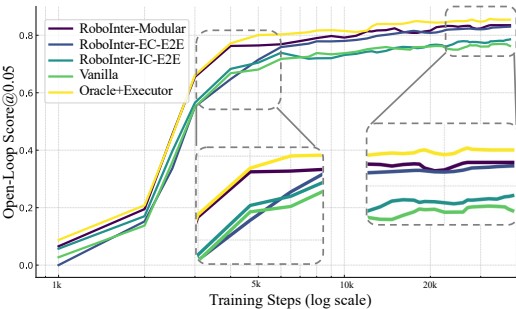

Figure 4: **Open-loop evaluation in TableTop setting**. We show the curve of OLS@0.05 from 1k to 40k training steps. We mainly report the five variances of RoboInter-VLA.

v.s. 0.3543), as visual prompting embeds within information-dense images, diluting their relative contribution. *QwenVL+Executor* employs Qwen2.5-VL as a zero-shot Planner, and its overall performance is not comparable with other Non-E2E models, ***showing that the embodied reasoning ability of our Planner is better than the general VLMs***. The evaluation curves of the *Table-Top* are shown in Figure.4, as Table-Top scenes are easier to interpret, most methods eventually achieve high scores. *RoboInter-Te-Modular* and *Oracle+Executor* converge faster and reach higher performance. *EC-E2E* converges more slowly but ultimately approaches the performance of *Te-Modular*. *IC-E2E* shows stronger early-stage results and maintains a consistent advantage over *Vanilla* after 20k steps.

**Ablations on intermediate representations**. We ablate combinations of different intermediate representations within the open-loop Oracle+Executor setting. As shown in Table 5, coarse-grained representations such as *Subtask* and *Primitive Skill* provide only marginal improvements, as they offer stage-level guidance with limited actionable constraints during execution. In contrast, spatially grounded representations (*Object Box*, *Gripper Box*, and *Affordance*) yield substantially larger gains by providing finer-grained cues. The most significant

Table 5: **Ablation of intermediate representation**. We report OLS under multiple thresholds. Six representations are evaluated, where finer-grained categories yield larger gains.

| Variant | OLS | | | | mOLS |
|---|---|---|---|---|---|
| | @0.1 | @0.05 | @0.03 | @0.01 | - |
| Vanilla | 0.6793 | 0.3608 | 0.1753 | 0.0189 | 0.3086 |
| + Subtask | 0.6965 | 0.3676 | 0.1770 | 0.0171 | 0.3146 |
| + S. + Primitive Skill | 0.6983 | 0.3681 | 0.1779 | 0.0194 | 0.3159 |
| + S. + P. + Object Box | 0.7025 | 0.3849 | 0.1988 | 0.0294 | 0.3289 |
| + S. + P. + O.B. + Gripper Box | 0.7212 | 0.4032 | 0.2048 | 0.0272 | 0.3391 |
| + S. + P. + O.B. + G.B. + Affordance | 0.7245 | 0.4083 | 0.2114 | 0.0297 | 0.3435 |
| + S. + P. + O.B. + G.B. + Aff. + Trace | **0.7511** | **0.4640** | **0.2705** | **0.0587** | **0.3861** |

improvement comes from *Trace*, which introduces dense, temporally grounded information and achieves the strongest overall performance. Additional results are provided in Appendix A.2.3.

## 5.3 CLOSED-LOOP REAL-WORLD EVALUATION OF THE EXECUTOR

**Experimental setting**. We study how our dataset and the pretrained Planner affect the closed-loop success rate. Experiments are conducted in a few-shot TableTop evaluation with a real-world Franka Research 3 arm. Observation input comprises a static third-person camera and a wrist-view camera, and no proprioceptive state. We focus on more practical E2E variants and evaluate five E2E models: (1) *OpenVLA* (Kim et al., 2024): Initialized from official pretrained weights and extended with an additional wrist-view input. (2) *Pi-0* (Black et al., 2024): Fine-tuned from the official checkpoints of Droid. (3) *Vanilla*: our baseline. (4) *RoboInter-IC-E2E*: Initialized from the Planner and further finetuned on in-distribution (ID) data. (5) *RoboInter-EC-E2E*: Initialized from the Planner, and jointly optimize action and CoT generation, with a 1:1 ratio of our annotated data and collected data (more details in A.1.1). We design four tasks: (a) Object Collecting: sequentially place three pens from a cluttered tabletop into a cup, out-of-distribution (OOD) tests are with novel objects, and alter spatial layouts. (b) Cup Stacking: Stack cups from left to right. OOD tests include novel objects, OOD positions, and continuous stacking. (c) Towel Folding: Fold two towels in sequence and stack

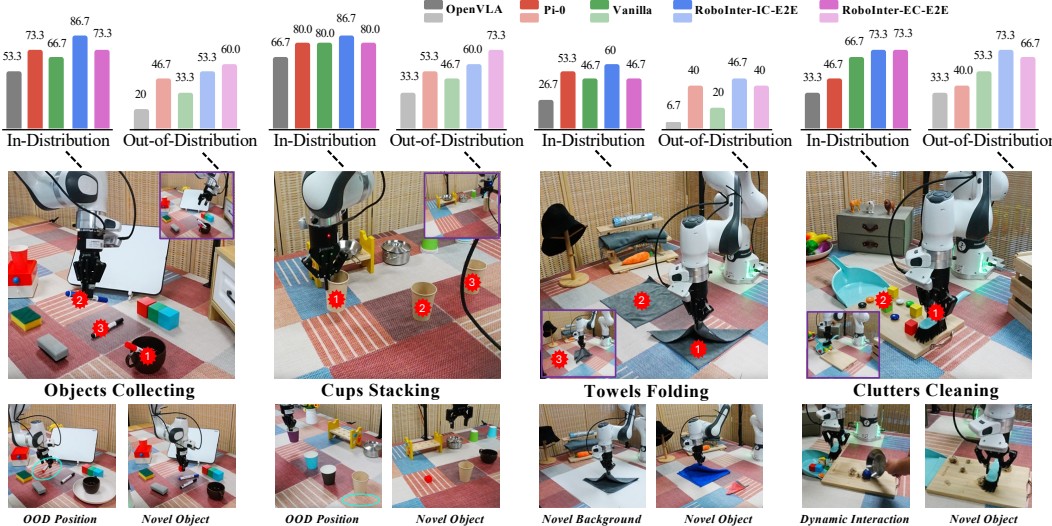

Figure 5: **Real-World Experiments.** The top charts present results from 15 in-distribution (ID) and 15 out-of-distribution (OOD) trials. The bottom panel illustrates the OOD test setup. Notably, the performance drop from ID to OOD reflects each model's generalization under distribution shift, where EC-E2E outperforms IC-E2E and exhibits a smaller ID→OOD degradation (8.3% vs. 19.0%), showing the consistent conclusion with the *Open-Loop Evaluation*. Key steps are marked with number, along with an end-execution thumbnail. Experiments of RoboInter-Modular are in Table.6.

them. OOD tests vary in the towel category and background. (d) Clutter Cleaning: Clean all items from the board with a brush. OOD tests introduce novel objects and disturbances.

**Experimental results on ID and OOD testing.** Across all tasks, *RoboInter-IC-E2E* consistently outperforms the *Vanilla*. In ID evaluations, IC-E2E attains an average success rate of 77.3%, compared with 65.0% of Vanilla. Under OOD conditions, the gap widens, and IC-E2E achieves a 58.3% success rate, while Vanilla reaches only 38.3%, indicating the superior generalization of *IC-E2E*. ***The pretrained VLM from the Planner is pre-exposed to rich embodied data and therefore provides stronger perceptual priors.*** Although Pi-0, which is pretrained on Droid, also demonstrates solid ID and OOD performance, the *IC-E2E* benefits from a broader representation training, thereby producing better overall results. *EC-E2E* records a lower ID success rate than *IC-E2E* (68.3% vs. 77.3%), which seems to be misaligned with the open-loop results in which *EC-E2E* was superior. Actually, the open-loop protocol enforces strict decouple between training and validation, and therefore functions more like an OOD test. Correspondingly, under real-world OOD conditions, *EC-E2E* exceeds *IC-E2E* in Object Collecting (60.0% vs. 53.3%) and Cup Stacking (73.3% vs. 60.0%), and achieves a higher average success rate (60.0% vs. 58.3%). The ID-to-OOD drop is only 8.3% for *EC-E2E*, whereas *IC-E2E* declines by 19%. We attribute *EC-E2E*'s weaker ID accuracy to the potential modality interference from the joint training of text generation and action prediction. ***Diverse OOD knowledge from our dataset contributes to superior OOD robustness and generalization.*** We provide more real-world results in Appendix A.1.

# 6 CONCLUSION

We presented the RoboInter Manipulation Suite, a unified platform designed to advance research on intermediate representations for the plan-then-execute VLA paradigm. At its core, RoboInter-Data offers over 230k episodes with dense, per-frame annotations spanning diverse intermediate representations, establishing a new scale and quality standard for real-world manipulation datasets. Built upon this foundation, RoboInter-VQA systematically benchmarks and improves the embodied generation and understanding capabilities of RoboInter-VLMs across rich spatial and temporal tasks. RoboInter-VLA further integrates intermediate representations into both modular and end-to-end frameworks, enabling a principled study of how representations influence execution performance.

ACKNOWLEDGMENTS

This work is funded in part by the National Key R&D Program of China (2022ZD0160201); Shanghai Artificial Intelligence Laboratory; the National Key Research and Development Program of China (No. 2024YFB4709800); the Special Program of the Graduate School, University of Science and Technology of China; National Key R&D Program of China (2022ZD0115502); Anhui Provincial Natural Science Foundation under Grant 2108085UD12; Ningbo Science and Technology Innovation 2025 Major Project (2025Z034); National Natural Science Foundation of China (No. 62461160308, U23B2010); "Pioneer" and "Leading Goose" R&D Program of Zhejiang (No. 2024C01161). We acknowledge the support of GPU cluster built by MCC Lab of Information Science and Technology Institution, USTC. The AI-driven experiments, simulations and model training were partly performed on the robotic AI-Scientist platform of Chinese Academy of Sciences.

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

# A  APPENDIX

## CONTENTS

## A.1 REAL-WORLD EXPERIMENTS FOR EXECUTOR

### A.1.1 EXPERIMENTAL SETTING

All Experiments are conducted in a few-shot TableTop evaluation with a real-world Franka Research 3 arm equipped with a Robotiq-2f-85 gripper. Controls are in the mode of delta end-effector motions. Observation input comprises a static third-person camera and a wrist-view camera, and no proprioceptive state is provided. Inference runs on an online A800 cluster and communicates with a local PC for robot control. To accommodate network latency, the control loop is limited to lower than 10 Hz, and demonstrations are collected using a SpaceMouse. We test: (1) OpenVLA (Kim et al., 2024): Initialized from publicly released OXE (et al., 2023) (including Droid (Khazatsky et al., 2024)) weights and extended to process both third-person and wrist-view images. (2) $\pi_0$ (Black et al., 2024): Fine-tuned from the official JAX checkpoints of the Droid dataset. (3) Vanilla: A from-scratch baseline without a pretrained Planner and without annotated intermediate representations. Our model designs mainly follow InternVLA-M1 (Chen et al., 2025b) and CogACT (Li et al., 2024b). (4) RoboInter-IC-E2E: Initialized from pretrained Planner weights and further finetuned on In-Distribution (ID) data. (5) RoboInter-EC-E2E: Initialized from the pretrained Planner, and action and CoT generation are jointly optimised during finetuning, with a 1:1 ratio of our annotated data and collected data. At inference time, we utilize a shorter CoT (only subtask, affordance box, and gripper box), as well as a caching mechanism that stores slowly varying components (e.g., subtask, affordance box) of the CoT sequence, and refreshes them every 30 control steps. (6) RoboInter-Modular: We use a 1:1 mixture of annotated and collected data to guide the learning of the Planner and Executor, respectively. The CoT sequence follows the same design as RoboInter-EC-E2E, while the Planner and Executor operate through an asynchronous inference process. We design four long-horizon, contact-rich tasks:

- **Object Collecting**. Sequentially place three pens from a cluttered tabletop into a cup, requiring the precise picking and placing; we collect 100 demonstrations. Out-of-distribution (OOD) tests are to replace the pens with novel objects, alter spatial layouts, and substitute the container.

- **Cup Stacking**. Stack cups from left to right across three $22cm \times 22cm$ zones, requiring precise spatial grounding ability; we collect 150 demonstrations. OOD tests include novel objects, OOD positions, and continuous stacking to evaluate the ability of re-planning.

- **Towel Folding**. Fold two towels in sequence and then stack them, evaluating the ability to manipulate deformable objects; we collect 64 demonstrations. OOD tests vary in the towel category, material, and background, evaluating the robustness.

- **Clutter Cleaning**. Clean all items from the board with a brush, and the task demands sustained operation and spatial perception; we collect 150 demonstrations. OOD tests introduce continuous streams of novel objects and random disturbances, evaluating a persistent cleaning process.

Table 6: **Real-world closed-loop performance**. We report success rates for four tasks under ID/OOD settings and the ID→OOD performance drop.

| Model | Objects Collect. | Cups Stack. | Towels Fold. | Clutters Clean. | ID→OOD Drop |
|---|---|---|---|---|---|
| | ID/OOD | ID/OOD | ID/OOD | ID/OOD | – |
| OpenVLA | 53.3 / 20.0 | 66.7 / 33.3 | 26.7 / 6.7 | 33.3 / 33.3 | 21.7 |
| $\pi_0$ | 73.3 / 46.7 | 80.0 / 53.3 | 53.3 / 40.0 | 46.7 / 40.0 | 18.3 |
| Vanilla | 66.7 / 33.3 | 80.0 / 46.7 | 46.7 / 20.0 | 66.7 / 53.3 | 26.7 |
| IC-E2E | 86.7 / 53.3 | 86.7 / 60.0 | 60.0 / 46.7 | 73.3 / 73.3 | 18.4 |
| EC-E2E | 73.3 / 60.0 | 80.0 / 73.3 | 46.7 / 40.0 | 73.3 / 66.7 | **8.3** |
| Modular | 66.7 / 53.3 | 86.7 / 73.3 | 53.3 / 40.0 | 80.0 / 73.3 | 11.7 |

Detailed results of closed-loop real-world evaluation are shown in Table 6. *IC-E2E* performs better than other variances across the ID settings of most tasks. *EC-E2E* exceeds *IC-E2E* in Object Collecting (60.0% vs. 53.3%) and Cup Stacking (73.3% vs. 60.0%) and achieves a higher average success rate (60.0% vs. 58.3%). The ID-to-OOD drop is only 8.3% for *EC-E2E*, whereas *IC-E2E* declines by 19%. The Modular variant achieves strong real-world performance and competitive out-of-distribution (OOD) generalization. Its slightly larger ID→OOD drop compared to EC-E2E indicates that the asynchronous two-module design is somewhat more sensitive to distribution shift, yet still substantially benefits from our intermediate representation.

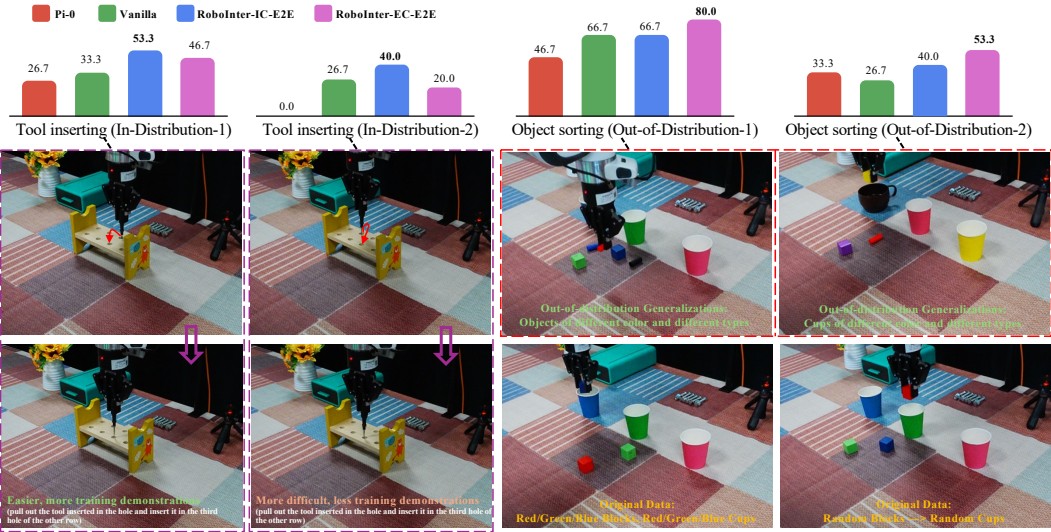

Figure 6: **Additional real-world ID and OOD validation.** (Left) *Tool Inserting*: a precision ID task requiring accurate contact handling and slot alignment. (Right) *Object Sorting*: an OOD task that tests language-guided generalization using novel objects and containers.

### A.1.2 ADDITIONAL REAL-WORLD ID AND OOD VALIDATION

To further disentangle in-distribution (ID) and out-of-distribution (OOD) generalization, we introduce two additional real-world tasks that emphasize different aspects of control and reasoning:

- **Tool Inserting (ID-oriented).** A precision-controlled manipulation task that evaluates the model's ability to fit fine-grained in-distribution actions. The robot must pull a metal tool out of a $1.5\,\mathrm{cm} \times 1.5\,\mathrm{cm}$ slot and re-insert it into another slot of the same size. Two ID variants with different initial positions and different training demonstrations are used, each requiring accurate contact handling.

- **Object Sorting (OOD-oriented).** A generalization task in which the robot must place objects into their corresponding target containers. Training demonstrations include only red, green, and blue objects and cups with simple pick-and-place motions. The OOD setting introduces novel objects and cups with novel colors, shapes, or types, assessing whether models can generalize sorting behaviors according to language instructions beyond the training distribution.

The results in Figure 6 show complementary strengths of the two variants: EC-E2E achieves stronger OOD performance owing to its explicit reasoning, whereas IC-E2E exhibits superior ID robustness. Overall, both variants outperform the Vanilla and $\pi_0$ baselines, demonstrating that intermediate representations significantly benefit both ID precision and OOD generalization.

### A.1.3 MORE RESULTS AND VISUALIZATION

We provide some visualizations of the real-world experimental process in Figures.7, 8, and 9.

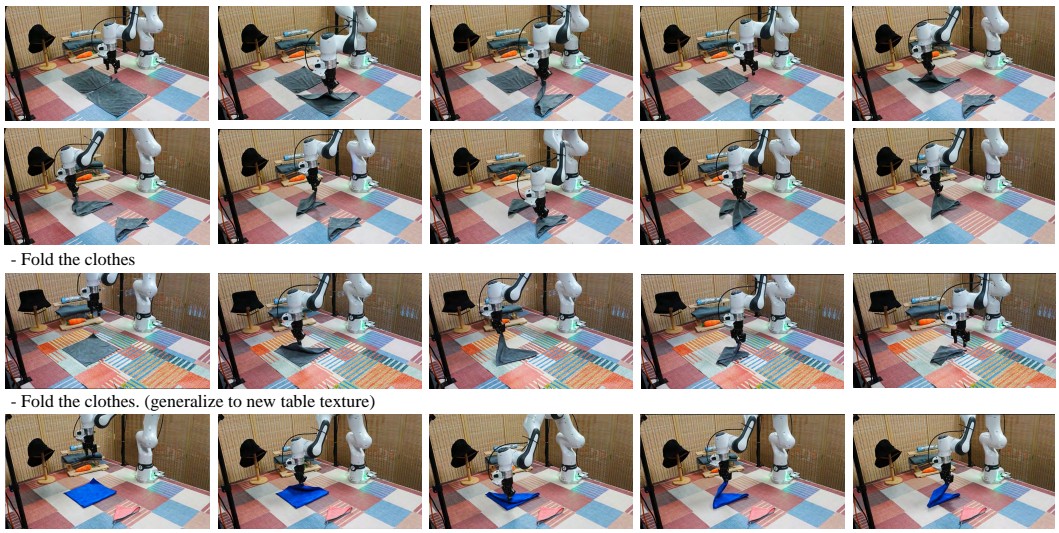

- Fold the clothes

- Fold the clothes. (generalize to new table texture)

- Fold the clothes. (generalize to new color)

Figure 7: **Real-world evaluation of folding deformable cloths.**

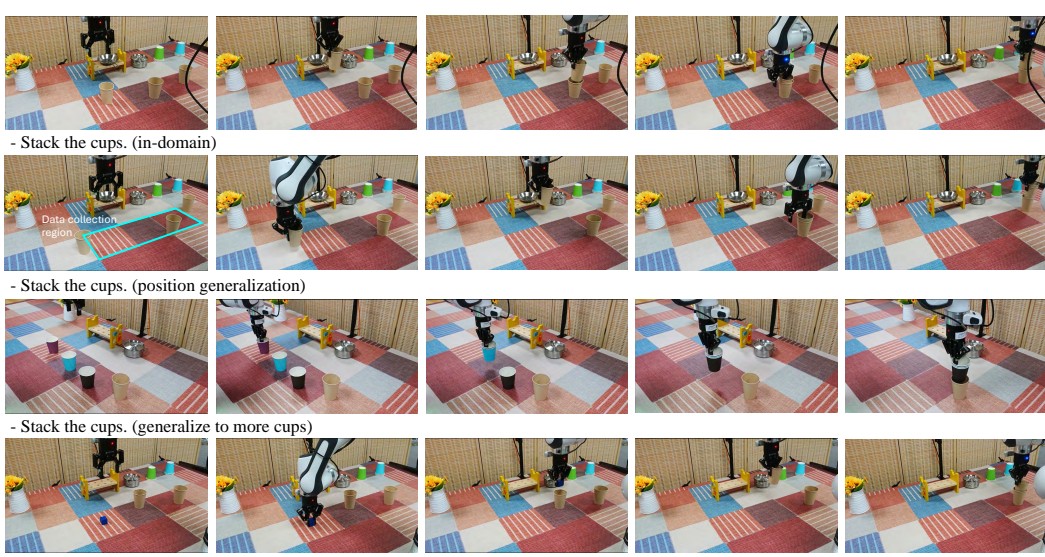

- Stack the cups. (in-domain)

- Stack the cups. (position generalization)

- Stack the cups. (generalize to more cups)

- Put the cube into the cup, then stack the cups. (generalize to novel objects)

Figure 8: **Real-world generalization evaluation of stacking three cups.** RoboInter-VLA generalizes to novel positions, four cups, unseen objects, and novel instructions.

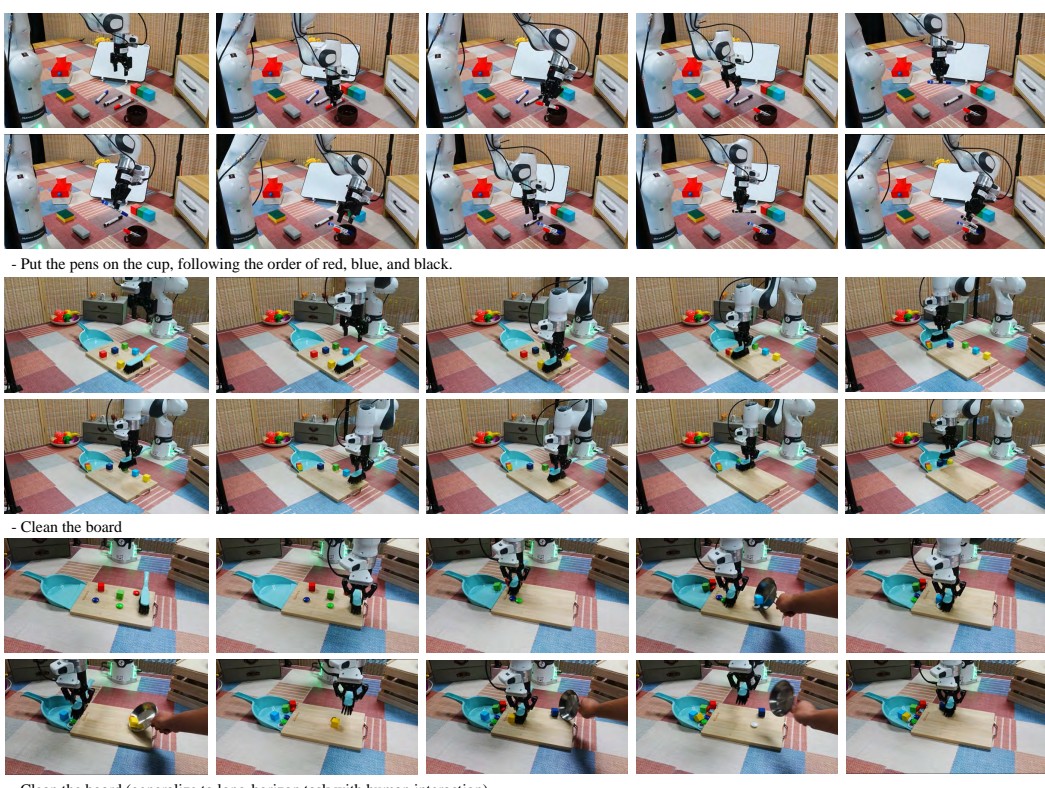

- Put the pens on the cup, following the order of red, blue, and black.

- Clean the board

- Clean the board (generalize to long-horizon task with human interaction)

Figure 9: **Real-world evaluation.** RoboInter-VLA demonstrates precise action generation (e.g., grasping a pen from the table while avoiding collision) and long-horizon capabilities, such as continuously cleaning the board.

## A.2 Additional Ablation Experiments for RoboInter-VLA

### A.2.1 Inference Time Analysis.

We report the inference-time characteristics of RoboInter-VLA in this section. For open-loop evaluation, our goal is to measure the accuracy of action prediction under comparable model sizes. As shown in Table 7, explicit reasoning significantly improves open-loop accuracy for both VLA-OS (Gao et al., 2025a) and RoboInter-VLA, but increases latency due to autoregressive CoT generation. For real-world deployment, we apply practical acceleration strategies, including textual caching and chunked execution for EC-E2E, and asynchronous dual-frequency execution for the Modular variant. Table 8 shows that these strategies substantially reduce latency while preserving strong ID and OOD performance. The general trend confirms that explicit reasoning enhances robustness at the cost of slower inference, motivating future work on more efficient execution.

Table 7: **Open-loop inference latency comparison**. Explicit reasoning improves accuracy but increases latency due to autoregressive generation.

| Model | Size | Reasoning Tokens | Latency (s) | OLS |
|---|---|---|---|---|
| VLA-OS (Vanilla) | 3B | 0 | 0.1647 | 0.3035 |
| VLA-OS-I-Explicit | 3B | 64 | 2.2274 | 0.3290 |
| Vanilla | 3B | 0 | 0.1309 | 0.3086 |
| RoboInter-IC-E2E | 3B | 64 | 0.1309 | 0.3218 |
| RoboInter-EC-E2E | 3B | 64 | 2.2187 | 0.3340 |
| RoboInter-Modular | 3B+3B | 64 | 2.3594 | 0.3543 |

Table 8: **Real-world inference speed with acceleration strategies**. Explicit reasoning increases robustness but incurs higher latency.

| Model | Speed (Hz) | Latency (ms) | ID Score | OOD Score |
|---|---|---|---|---|
| OpenVLA (+wrist) | 3.8 | 263.2 | 45.0 | 23.3 |
| Vanilla | 7.64 | 130.9 | 65.0 | 38.3 |
| RoboInter-IC-E2E | 7.64 | 130.9 | 76.7 | 58.3 |
| RoboInter-EC-E2E (cache+chunk) | 2.56 | 390.6 | 68.3 | 60.0 |
| RoboInter-Modular (async) | 6.92 | 144.5 | 73.3 | 60.0 |

### A.2.2 Experiment Results for Data Scaling Law

**Experiments for RoboInter-VQA scaling law.** As shown in Table 9, Table 10, and Table 11, we conduct data sampling rate experiments on RoboInter-VQA with QwenVL2.5-3B, varying the proportion of our data from 10% to 100%. The training set consists of general VQA data combined with the corresponding subset of our annotated data. The results clearly show that the model follows a scaling-law trend with the dataset size increases.

Table 9: **The scaling low with data of all visual generation VQA data.**

| Data Sampling Rate(%) | Object Grounding | Grasp Affordance | Placement Affordance | Gripper Detection | Trace Generation |
|---|---|---|---|---|---|
| 10% | 68.1% | 22.9% | 42.0% | 46.1% | 442 |
| 50% | 71.9% | 32.9% | 51.6% | 58.7% | 375 |
| 100% | 72.8% | 35.0% | 53.2% | 61.5% | 352 |

**Experiments for F-CoT data scaling law.** As shown in Tab. 12, we vary the proportion of paired F-CoT and action samples to study how data scale influences open-loop performance. We compare two settings: *Oracle+Executor* and *RoboInter-Te-Modular*. As the sampling ratio increases, both models improve; however, *Oracle+Executor* exhibits a larger gain ($0.2347 \rightarrow 0.2705$) than *RoboInter-Te-Modular* ($0.2109 \rightarrow 0.2332$). This suggests that, with more data, Executor directly conditioned on oracle intermediate signals benefits more readily, while the VLM Planner improves at a slower rate.

Table 10: **The scaling low with data of all visual understanding tasks.**

| Data Sampling Rate(%) | T/F | | Multiple Choice | | | | |
|---|---|---|---|---|---|---|---|
| | Contact Decide | | Grasp Pose | Object Grounding | Trace | Trace Dir | Trace Lang |
| 10% | 69.9% | | 30.6% | 74.8% | 43.6% | 53.3% | 65.8% |
| 50% | 82.9% | | 34.4% | 77.9% | 65.9 % | 73.6% | 77.1% |
| 100% | 85.4% | | 41.3% | 79.0% | 71.9% | 74.9% | 80.4% |

Table 11: **The scaling low with data of all planning tasks.**

| Data Sampling Rate(%) | Task Planning | | |
|---|---|---|---|
| | Multiple Choice | T/F | Planning |
| 10% | 63.5% | 66.0% | 54.8 |
| 50% | 69.2% | 73.5% | 59.8 |
| 100% | 72.1% | 75.1% | 62.1 |

Table 12: **F-CoT data scaling law** between Oracle+Executor and RoboInter-Te-Modular across varying data sampling rates.

| Data Sampling Rate (%) | Oracle+Executor | | RoboInter-Te-Modular | |
|---|---|---|---|---|
| | OLS@0.03 | OLS@0.01 | OLS@0.03 | OLS@0.01 |
| 10% | 0.2347 | 0.0236 | 0.2109 | 0.0249 |
| 30% | 0.2590 | 0.0280 | 0.2225 | 0.0285 |
| 50% | 0.2686 | 0.0400 | 0.2372 | 0.0429 |
| 100% | 0.2705 | 0.0587 | 0.2332 | 0.0584 |

### A.2.3 ABLATION FOR DESIGNS AND INTERMEDIATE REPRESENTATIONS TYPES OF F-CoT

**Ablation for designs.** F-CoT serves as a flexible carrier that composes multiple intermediate representations from RoboInter-Data. We evaluate two forms: (1) Textual F-CoT: Intermediate representations are serialized into autoregressive text tokens, used in RoboInter-Te-Modular. (2) Visual F-CoT: Intermediate representations are drawn as visual prompts on additional input images, used in RoboInter-Im-Modular. As shown in Table.4, Textual F-CoT (RoboInter-Te-Modular) consistently performs best among learning-based models, indicating that structured textual reasoning may be a more stable and expressive interface for composing heterogeneous intermediate representations.

**Ablation for intermediate representation types.** As shown in Table.5, we have ablated the contribution of each F-CoT component using the *Oracle+Executor* setting. We further ablate how each intermediate representation individually affects performance in Table.13. The trend is consistent: coarse-grained representations (Subtask, Primitive skill) give modest gains, whereas spatially precise cues (Object Box, Affordance) provide much greater improvements.

### A.2.4 DETAILED RESULTS OF PLANNER ON TEMPORAL ROBOINTER-VQA

We present more detailed results of the RoboInter-VQA benchmark in Table.14, Table.15, Table.16, Table.17, including entries that were omitted or only summarized by average scores in Table.3.

Table 13: **Ablation results of single intermediate representations**. We report OLS under four thresholds and the mean value (mOLS).

| Method | OLS | | | | mOLS |
|---|---|---|---|---|---|
| | @0.10 | @0.05 | @0.03 | @0.01 | – |
| Vanilla | 0.6793 | 0.3608 | 0.1753 | 0.0189 | 0.3086 |
| + Subtask | 0.6965 | 0.3676 | 0.1770 | 0.0171 | 0.3146 |
| + Primitive skill | 0.6925 | 0.3658 | 0.1793 | 0.0156 | 0.3133 |
| + Object Box | 0.7001 | 0.3793 | 0.1879 | 0.0202 | 0.3218 |
| + Affordance | 0.7054 | 0.3958 | 0.2012 | 0.0263 | 0.3322 |

Table 14: **Performance comparisons on temporal tasks** of the RoboInter-VQA benchmark. "–" indicates the model could not be accurately evaluated for that metric.

| Model Name | Generation | | Multiple Choice | | | T/F↑ |
|---|---|---|---|---|---|---|
| | Trace-Easy↓ | Trace-Hard↓ | Trace↑ | Trace Dir↑ | Traj Lang↑ | Contact |
| QwenVL2.5-3B | 2752 | 2672 | 26.5% | 35.6% | 50.5% | 50.1% |
| QwenVL2.5-7B | 1553 | 1850 | 26.4% | 35.7% | 54.8% | 54.3% |
| InternVL3-1B | – | – | 16.5% | 28.2% | 42.1% | 51.6% |
| InternVL3-2B | – | 1216 | 19.1% | 30.8% | 55.7% | 53.5% |
| InternVL3-8B | 656 | 1414 | 16.2% | 29.6% | 56.1% | 58.7% |
| Llava-OV-7B | – | – | 27.8% | 25.9% | 59.3% | 57.8% |
| Robobrain-3B-2.0 | 592 | 598 | 25.6% | 28.1% | 35.3% | 52.0% |
| Robobrain-7B-2.0 | 540 | 541 | 30.7% | 24.4% | 33.4% | 50.5% |
| GPT4o-mini | 1091 | 2381 | 9.4% | 21.4% | 54.3% | 53.6% |
| Gemini-2.5-flash | – | – | 44.4% | 38.7% | 65.2% | 65.5% |
| RoboInter-Qwen-3B | 315 | 350 | 73.9% | 81.5% | 81.1% | 72.5% |
| RoboInter-Qwen-7B | 302 | 344 | 81.1% | 80.4% | 84.3% | 74.4% |
| RoboInter-Llava-OV-7B | 291 | 306 | 83.6% | 83.5% | 78.5% | 76.3% |

Table 15: **Detailed performance on Multiple Choice of task planning (ACC%↑).**

| Task | Qwen VL2.5-3B | Qwen VL2.5-7B | Intern VL3-1B | Intern VL3-2B | Intern VL3-8B | Robo brain-3B | Robo brain-7B | Llava OV-7B | RoboInter Qwen-3B | RoboInter Qwen-7B | RoboInter LlavaOV-7B |
|---|---|---|---|---|---|---|---|---|---|---|---|
| Scene Understanding | 38.3 | 55.5 | 28.8 | 40.8 | 70.0 | 41.6 | 59.0 | 53.0 | 83.2 | **86.2** | 83.9 |
| Past Multi-task Sel. | 93.8 | 94.2 | 88.2 | 90.3 | 93.6 | 70.6 | 81.0 | 70.4 | 94.2 | **96.4** | 90.4 |
| Past Primitive Sel. | 64.3 | 65.0 | 53.0 | 60.2 | 68.8 | 48.4 | 53.2 | 56.0 | 84.2 | **88.9** | 81.7 |
| Future Multi-task Sel. | 83.4 | 85.9 | 82.0 | 81.9 | 84.2 | 52.2 | 68.1 | 25.2 | 90.5 | **93.6** | 90.1 |
| Future Primitive Sel. | 41.6 | 48.0 | 45.2 | 45.2 | 58.6 | 40.0 | 43.9 | 22.0 | 78.1 | **83.9** | 75.8 |
| Temporal Understand | 23.1 | 23.7 | 26.4 | 26.1 | 52.2 | 31.4 | 45.2 | 25.9 | 42.9 | 50.7 | **60.1** |
| Average (weighted) | 60.0 | 64.5 | 54.9 | 59.3 | 71.5 | 48.2 | 57.8 | 44.9 | 82.2 | **86.5** | 81.8 |

Table 16: **Detailed performance on T/F (Decide) Tasks of task planning (ACC%↑).**

| Task | Qwen VL2.5-3B | Qwen VL2.5-7B | Intern VL3-1B | Intern VL3-2B | Intern VL3-8B | Robo brain-3B | Robo brain-7B | Llava OV-7B | RoboInter Qwen-3B | RoboInter Qwen-7B | RoboInter LlavaOV-7B |
|---|---|---|---|---|---|---|---|---|---|---|---|
| Success Negative | 89.5 | 96.7 | 55.5 | 68.1 | 55.0 | 2.1 | 0.0 | 60.1 | 81.3 | 90.5 | 69.8 |
| Success Positive | 14.3 | 11.4 | 46.4 | 37.2 | 50.3 | 98.4 | 100.0 | 46.2 | 89.1 | 91.6 | 85.7 |
| Discrim. Afford. Neg. | 92.4 | 97.0 | 64.5 | 67.8 | 71.2 | 1.5 | 0.2 | 75.4 | 93.4 | 95.9 | 95.2 |
| Discrim. Afford. Pos. | 34.4 | 26.3 | 58.3 | 63.1 | 67.5 | 99.2 | 100.0 | 76.2 | 94.9 | 95.6 | 92.3 |
| Average (weighted) | 59.7 | 60.5 | 55.8 | 59.4 | 60.0 | 46.8 | 46.4 | 63.5 | 88.7 | 93.0 | 83.9 |

Table 17: **Detailed performance on Generation Tasks of task planning (Average results ↑ of BLEU-1 to BLEU-4 Score).**

| Task | Qwen VL2.5-3B | Qwen VL2.5-7B | Intern VL3-1B | Intern VL3-2B | Intern VL3-8B | Robo brain-3B | Robo brain-7B | Llava OV-7B | RoboInter Qwen-3B | RoboInter Qwen-7B | RoboInter LlavaOV-7B |
|---|---|---|---|---|---|---|---|---|---|---|---|
| Planning Task | 28.4 | 25.4 | 13.2 | 8.1 | 6.8 | 17.7 | 16.7 | 11.8 | 61.0 | 64.7 | 61.1 |
| Past Description | 8.5 | 14.4 | 3.3 | 2.0 | 2.8 | 6.5 | 6.9 | 0.8 | 46.1 | 51.7 | 48.9 |
| Planning w/ Context | 33.3 | 31.6 | 14.6 | 13.2 | 14.8 | 28.1 | 25.1 | 28.6 | 76.7 | 76.7 | 78.1 |
| Planning Remaining | 20.5 | 24.4 | 13.4 | 12.5 | 13.2 | 19.3 | 22.3 | 19.5 | 73.2 | 73.5 | 74.0 |
| Generative Affordance | 6.3 | 12.9 | 4.0 | 3.0 | 2.0 | 5.6 | 5.6 | 1.1 | 43.3 | 45.8 | 45.3 |
| Future Prediction | 22.3 | 25.4 | 13.8 | 7.8 | 10.0 | 18.8 | 16.1 | 5.7 | 69.0 | 68.1 | 71.2 |
| Average (weighted) | 20.3 | 22.4 | 10.5 | 7.7 | 8.1 | 16.0 | 15.3 | 11.0 | 61.2 | 63.4 | 62.7 |

## A.3 CROSS-PLATFORM VALIDATION

### A.3.1 OPEN-LOOP CROSS-PLATFORM EVALUATION

Our dataset spans over 570 scenes across multiple robotic embodiments, forming a hybrid collection that is both cross-platform and cross-scene. Because both the training and validation splits include diverse platforms, the open-loop evaluation naturally serves as cross-platform testing. We report platform-specific open-loop performance in Table 18, including five distinct embodiments. The results show that all RoboInterVLA variants consistently outperform the vanilla baseline across platforms, with the Modular configuration achieving the best overall accuracy among learned models. The Oracle+Executor serves as the upper bound.

Table 18: **Cross-platform open-loop evaluation across different robot embodiments.**

| Model | Franka+Panda | UR5+Robotiq | Flexiv+AG-95 | Kuka+Robotiq | Franka+Robotiq | mean OLS |
|---|---|---|---|---|---|---|
| Vanilla | 0.6756 | 0.9659 | 0.7646 | 0.7476 | 0.3608 | 0.7029 |
| RoboInter-IC-E2E | 0.7149 | 0.9929 | *0.7786* | 0.7564 | 0.3810 | 0.72476 |
| RoboInter-EC-E2E | 0.8098 | 0.9876 | 0.7723 | *0.7748* | 0.3930 | 0.7475 |
| RoboInter-Modular | *0.8572* | **0.9962** | 0.7743 | 0.7609 | *0.4133* | *0.76038* |
| Oracle+Executor | **0.8828** | *0.9942* | **0.7809** | **0.7848** | **0.4640** | **0.78134** |

### A.3.2 CLOSE-LOOP EVALUATION ON SIMPLERENV

**Evaluation setup in SimplerEnv.** We conduct a close-loop experiment within SimplerEnv (Li et al., 2024d), a benchmark to evaluate models across various tasks with the WidowX Robot (WR) and the Google Robot (GR). The GR environment includes two settings, Visual Matching (VM) and Variant Aggregation (VA). WidowX Robot environment only includes the VM settings. SimplerEnv covers more than 2k trails across different scenarios, objects, and tasks. We report the average success rate of each task. RT-1-X (et al., 2023), RT-2-X (et al., 2023), and Octo-Based (Octo Model Team et al., 2024) are early baselines, OpenVLA (Kim et al., 2024), CogACT (Li et al., 2024b), and Magma (Yang et al., 2025a) are trained on subsets of the OXE. $\pi_0$ and $\pi_0$-FAST are VLAs with additional robot state input. RoboVLMs (Li et al., 2024c), SpatialVLA (Qu et al., 2025), and TraceVLA (Zheng et al., 2024) are all trained on the Fractal and Bridge-V2 datasets.

Table 19: **Performance comparison on SimplerEnv.** The experiments are conducted across 12 tasks, including both Visual Matching and Visual Aggregation. The parameter size of LLM are shown behind the model name. % of success rate is omitted.

| | Google Robot | | | | | | | | | | WidowX Robot | | | | | |
|---|---|---|---|---|---|---|---|---|---|---|---|---|---|---|---|---|
| Methods | Open/Close Drawer | | Put in Drawer | | Pick Coke Can | | Move Near | | Avg | | Put Spoon | Put Carrot | Stack Blocks | Put Eggplant | Avg | Avg |
| | VM | VA | VM | VA | VM | VA | VM | VA | VM | VA | VM | | | | | |
| RT-1-X (et al., 2023) | 59.7 | 29.4 | 21.3 | 10.1 | 56.7 | 49.0 | 31.7 | 32.3 | 42.4 | 30.2 | 0.0 | 4.2 | 0.0 | 0.0 | 1.1 | 24.6 |
| RT-2-X (et al., 2023) | 25.0 | 35.5 | 3.7 | 20.6 | 78.7 | 82.3 | 77.9 | 79.2 | 46.3 | 54.4 | - | - | - | - | - | - |
| Octo-Base (Octo Model Team et al., 2024) | 22.7 | 1.1 | 0.0 | 0.0 | 17.0 | 0.6 | 4.2 | 3.1 | 11.0 | 1.2 | 15.8 | 12.5 | 0.0 | 41.7 | 17.5 | 9.9 |
| RoboVLMs (2B) (Li et al., 2024c) | 44.9 | 10.3 | 27.8 | 0.0 | 76.3 | 50.7 | 79.0 | 62.5 | 57.0 | 30.9 | 50 | 37.5 | 0.0 | 83.3 | 42.7 | 43.5 |
| SpatialVLA (3B) (Qu et al., 2025) | 54.6 | 39.2 | 0.0 | 6.3 | 79.3 | 78.7 | 90.0 | 83.0 | 56.0 | 51.8 | 20.8 | 37.5 | 41.7 | 83.3 | 45.8 | 51.2 |
| $\pi_0$ (3B) (Black et al., 2024) | 38.3 | 25.6 | 46.6 | 20.5 | 72.7 | 75.2 | 65.3 | 63.7 | 55.7 | 46.3 | 29.1 | 0.0 | 16.6 | 62.5 | 27.1 | 43.0 |
| $\pi_0$-FAST (3B) (Pertsch et al., 2025) | 42.9 | 31.3 | - | - | 75.3 | 77.6 | 67.5 | 68.2 | - | - | 29.1 | 21.9 | 10.8 | 66.6 | 32.1 | - |
| OpenVLA (7B) (Kim et al., 2024) | 59.7 | 23.5 | 0.0 | 2.9 | 25.7 | 54.1 | 55.0 | 63.0 | 35.1 | 35.9 | 8.3 | 4.2 | 0.0 | 0.0 | 3.1 | 24.7 |
| CogACT (7B) (Li et al., 2024b) | 71.8 | 28.3 | 50.9 | 46.6 | 91.3 | 89.6 | 85.0 | 80.8 | 74.8 | 61.3 | 75.0 | 50.0 | 16.7 | 79.2 | 55.2 | 63.8 |
| TraceVLA (7B) (Zheng et al., 2024) | 63.1 | 61.6 | 11.1 | 12.5 | 45.0 | 64.3 | 63.8 | 60.6 | 45.8 | 49.8 | 12.5 | 16.6 | 16.6 | 65.0 | 27.7 | 41.1 |
| Magma (8B) (Yang et al., 2025a) | 58.9 | 59.0 | 8.3 | 24.0 | 75.0 | 68.6 | 53.0 | 78.5 | 48.8 | 57.5 | 37.5 | 29.2 | 20.8 | 91.7 | 44.8 | 50.4 |
| Vanilla (3B) | 56.9 | 27.5 | 65.7 | 48.6 | 97.7 | 86.2 | 86.0 | 85.0 | 76.6 | 61.8 | 41.7 | 37.5 | 12.5 | 95.8 | 46.9 | 61.8 |
| **RoboInter-IC-E2E (3B)** | **73.6** | 55.3 | 48.1 | **58.3** | 94 | 87.7 | 88.0 | 84.0 | 75.9 | **71.3** | 54.2 | 25.0 | 29.2 | 91.7 | **50.0** | **65.7** |

**Results.** As shown in Table 19, we initialize our RoboInter-IC-E2E model with the pretrained VLM of *Planner* and perform implicitly conditioned end-to-end training on Fractal and Bridge data. Because *RoboInter-Data* does not include action annotations for WidowX or Google robots, this constitutes a strictly cross-embodiment evaluation. On *SimplerEnv*, our minimal *Vanilla* design outperforms common baselines ($\pi_0$, $\pi_0$-FAST), though it is slightly below CogACT (61.8 vs. 63.8). With a stronger embodied VLM initialization, *RoboInter-IC-E2E* improves to 65.7, surpassing CogACT (65.7 vs. 63.8). These results validate the implicit *planner-then-executor* and highlight the transferability of our data.

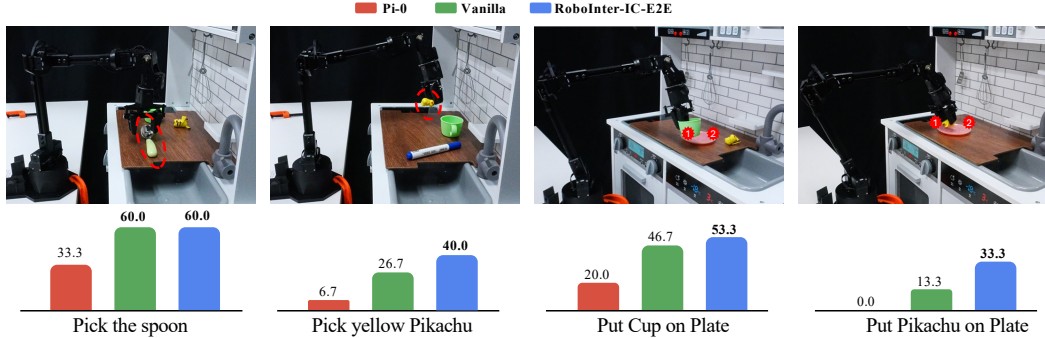

Figure 10: **Real-world WidowX Robot results.**

### A.3.3 CLOSE-LOOP EVALUATION ON REAL-WORLD WIDOWX ROBOT

To further assess real-world generalization capabilities, we conduct closed-loop zero-shot experiments on the WidowX-250 robotic arm. We evaluate three models: $\pi_0$ (Black et al., 2024), Vanilla baseline, and our RoboInter-IC-E2E. All models are pretrained on the BridgeV2 without further post-training or finetuning prior to deployment, and all experiments are executed using the same real-world setup. Our evaluation focuses on a kitchen environment, where we design four manipulation tasks, each executed 15 times:

- **Pick the Spoon:** The robot must grasp a metal spoon placed at arbitrary positions on a kitchen board.
- **Pick Yellow Pikachu:** This serves as an out-of-distribution (OOD) test, as similar objects (Pikachu) do not appear in the BridgeV2 dataset. The small toy increases the difficulty.
- **Put Cup on Plate:** A pick-and-place task where the robot must pick up a cup and accurately place it at the center of a plate.
- **Put Pikachu on Plate:** An OOD variant of the previous task, replacing the cup with the Pikachu toy, making the placement more challenging.

The experimental setup is illustrated in Figure 10. Across all tasks, our *RoboInter-IC-E2E* model demonstrates the strongest real-world performance, benefiting from its more powerful pretrained representation and showing superior generalization in closed-loop control.

### A.4 ADDITIONAL DETAILS OF TRAINING AND EVALUATION.

#### A.4.1 TRAINING DETAILS OF PLANNER AND EXECUTOR

**Planner training hyperparameters.** The RoboInterVLM(QwenVL2.5 7B) Planner is trained for one epoch with a global batch size of 128 and a per-device batch size of 4, without gradient accumulation. The base learning rate, as well as the learning rates for the vision tower and the multimodal projector, are all set to $3 \times 10^{-6}$. Training uses BF16 mixed precision, a maximum gradient norm of 1.0, zero weight decay, and a warmup ratio of 0.03. More details will be released in our codebase.

**Executor training hyperparameters.** The RoboInter-IC-E2E Executor is trained with a global batch size of 128 and a per-device batch size of 8. The base learning rate is $5 \times 10^{-5}$, with no warmup and no weight decay. Action head, the LLM backbone, and the vision backbone all remain trainable. The Q-Former operates between layers 36 and 37, predicting actions of 7 dimensions over a future action window of 15 steps. More details will be released in our codebase.

**VLM training data**. We partially follow the basic VLM training recipe of InternVL (Chen et al., 2024b), and as shown in Figure 11, to ensure that the Planner acquires both strong general perceptual understanding and embodied prior knowledge, we carefully select a broad range of VQA data for training.

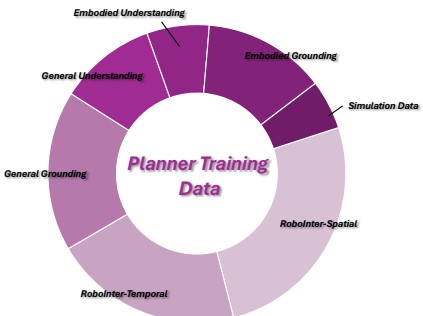

Figure 11: **Training data distribution for the Planner.** We cover a broad range of training data, including temporal/spatial understanding/general, embodied reasoning, and spatial grounding.

#### A.4.2 VQA BENCHMARKS

Recent benchmarks for spatial referring and multimodal reasoning can be broadly categorized into embodied and non-embodied settings. Embodied benchmarks such as Where2Place (Zhou et al., 2025a), RoboRefIt (Lu et al., 2023), RoboVQA (Sermanet et al., 2024), and RefSpatial-Bench (Lu et al., 2023) emphasize 3D-aware grounding, ambiguous object reasoning, and multi-step spatial understanding in realistic robotic contexts, thereby capturing the complexities of embodied interaction. In contrast, non-embodied benchmarks, which include RefCOCO/+/g (Lin et al., 2014) for referring expression comprehension, TextVQA (Singh et al., 2019), COCO (Lin et al., 2014), OCRBench (Liu et al., 2024b), MME (Fu et al., 2024a), MMVet (Yu et al., 2023), and POPE (Li et al., 2023b), have driven advances in multimodal reasoning but remain limited in sequential and interactive dimensions essential for robotic deployment.

#### A.4.3 METRIC DETAILS.

**Open-loop evaluation protocol** Our annotated corpus spans more than 500 distinct scenarios and contains roughly 200k demonstrations for large-scale robotic training. However, as emphasized by HPT (Wang et al., 2024a), comprehensive real-world validation across all scenarios is infeasible, and there is no practical approach for evaluating large-scale pretraining. Our primary objective is to quantify how intermediate representations benefit downstream action generation, so we adopt an open-loop evaluation protocol in which per-step actions are assessed independently: given the vision and language inputs, the model predicts a candidate action that is then compared with the ground-truth action. HPT compares *validation loss*; however, for the diffusion-based model, validation

loss is sensitive to the per-step noise schedule. Following the discrete token-accuracy evaluation in OpenVLA (Kim et al., 2024), we utilize an **Open-Loop Score (OLS)** for *continuous action chunks*:

$$\text{OLS} = \frac{1}{M} \sum_{i=1}^{M} \mathbf{1}\left\{ \frac{1}{T} \sum_{t=1}^{T} \mathbf{1}(|\hat{a}_i^{(t)} - a_i^{(t)}| < \tau) \geq \theta \right\}, \tag{1}$$

where $M$ is the number of evaluation transitions, $T$ is the length of each action chunking, $\tau$ denotes the error threshold, $\theta \in [0, 1]$ is the chunking-level tolerance (default = 1). For each frame, we fully denoise a $T$-step action chunk. If the mean stepwise error stays below $\theta$, the episode counts as 1, otherwise counts as 0 through $\mathbf{1}(*)$. OLS is the average value over all transitions. The validation set is independent of all training data and contains over 100K transitions, ensuring statistical stability.

## A.5 CASE STUDY

We present qualitative examples in Figure 12 and Figure 13. Although RoboBrain, which is trained on embodied datasets, achieves performance on par with Qwen2.5VL, this observation highlights the inherent limitations of the embodied data it relies upon. In contrast, RoboInter (RoboInterVLM Qwen2.5VL 7B) consistently demonstrates superior performance across different settings. These results indicate that our planner model is capable of intermediate-level representational reasoning and informed decision-making, thereby serving as an effective tool for data selection. Moreover, as shown in Figure 14, when evaluated on our self-collected zero-shot dataset, the RoboInterVLM model continues to exhibit strong generalization ability, maintaining robustness even when encountering entirely novel objects.

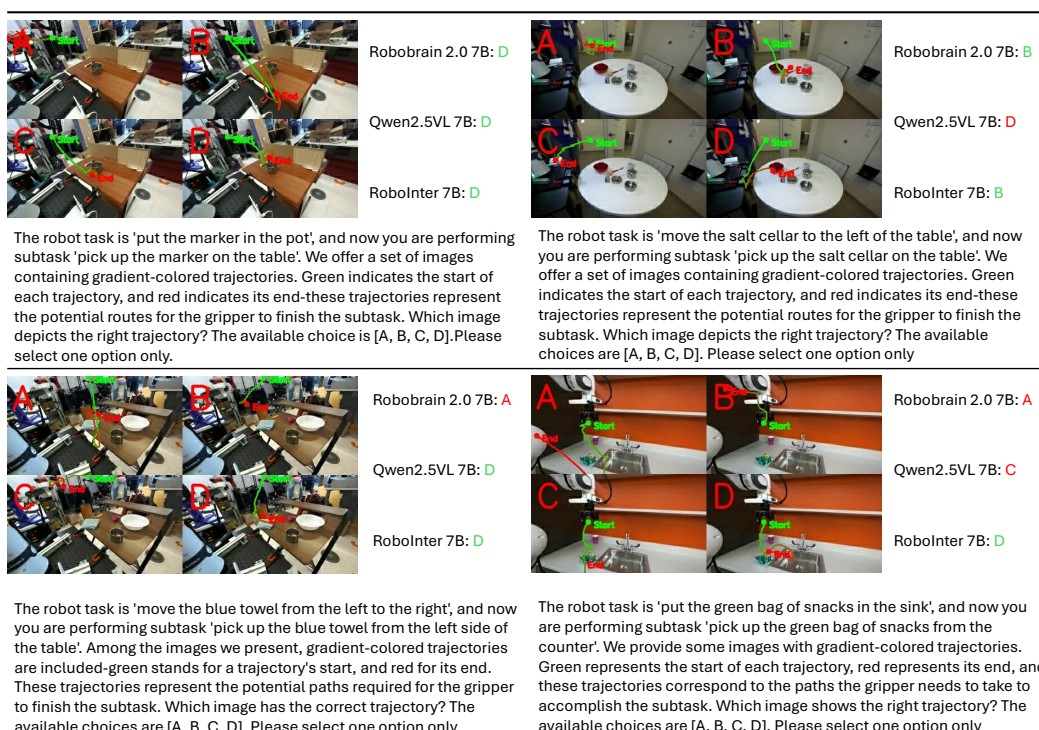

Figure 12: **Case study on RoboInter-VQA.**

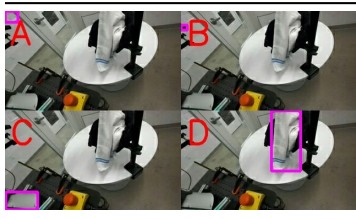
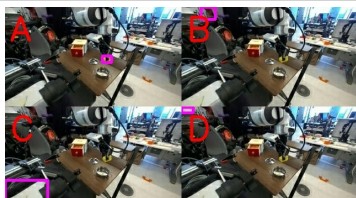

Robobrain 2.0 7B: D

Qwen2.5VL 7B: D

RoboInter 7B: D

The robot task is 'pick up the white cloth and put it on the table'. We provide images with purple boxes that represent the objects involved in the robotic arm's interactions, which image accurately depicts the bounding box of the object? The available choices are [[A, B, C, D]]. Please select one option only.

Robobrain 2.0 7B: D

Qwen2.5VL 7B: D

RoboInter 7B: A

The robot task is 'remove the yellow object from the pot and place it on the table'. We present images containing purple boxes, which indicate the objects that the robotic arm maybe interacts with, which image correctly shows the bounding box of the object? The available choices are [[A, B, C, D]]. Please select one option only.

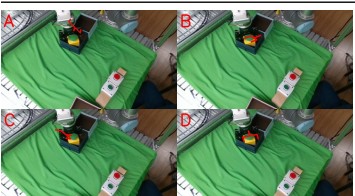

Robobrain 2.0 7B: A

Qwen2.5VL 7B: D

RoboInter 7B: D

The robot task is 'move an object from one box to another'. We provide some images that feature orange fork-like gripper patterns; these patterns stand for the potential poses the gripper needs to adopt to pick up an object. Which of the images below contains the correct grasping pose? The available choice is [A, B, C, D]. Please select one option only.

Robobrain 2.0 7B: A

Qwen2.5VL 7B: D

RoboInter 7B: B

The robot task is 'play the drum'. We present some images annotated with orange fork-like gripper patterns. These patterns correspond to the potential poses the gripper may adopt to grab an object-so which image below has the right grasping pose? The available choices are [A, B, C, D]. Please select one option only.

Figure 13: **Case study on RoboInter-VQA.**

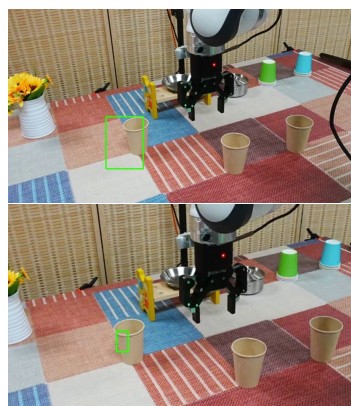

<image> \n The robot task is 'pickup the center bottle', and now you are performing subtask 'grasp the left bottle'\n\n Leveraging the observed scene, return the manipulated object's current bounding box.\n Return in JSON format: The box should be formatted as [[x1, y1], [x2, y2]], which means the left top points and right down points.

Answer: [0.5, 0.31, 0.69, 0.46]

<image> \n The robot task is 'pickup the center bottle', and now you are performing subtask 'grasp the left bottle'\n Leveraging the observed scene, report the expected gripper-object contact region.\n Return in JSON format: The region should be formatted as [[x1, y1], [x2, y2]], representing the left top points and right down points of the bounding box of the gripper at the moment of contact with the object.
Answer: [0.5, 0.31, 0.69, 0.46]

Figure 14: **Zero-shot evaluation on object grounding and grasp affordance tasks.**

## A.6 Annotation Workflow

### A.6.1 RoboInter-Tool

To enable efficient and consistent annotation for large-scale, high-precision robotic manipulation datasets, we develop *RoboInter-Tool*, a PyQt5-based multimodal annotation platform. The tool supports multi-granularity annotations at the video, clip, and frame levels, including language descriptions, gripper keypoint labeling, segmentation mask generation, and keyframe selection. Its modular design integrates annotation, visualization, and editing, allowing annotators to perform rapid validation and iterative refinement. *RoboInter-Tool* consists of two main components: a semantic segmentation annotation interface and a language annotation interface, with built-in cloud-based asynchronous quality verification.

**Semantic Segmentation Annotation Interface.** This interface focuses on frame-level precision, automated segmentation assistance, and multi-object flexibility (Figure 15). It contains three main components: (1)**Video annotation and visualization area.** The system supports frame-by-frame playback and review, enabling annotators to precisely identify keyframes of robot-object interaction. By clicking on video frames, annotators can label gripper keypoints and provide them to the SAM2 (Ravi et al., 2024), which then automatically generates segmentation masks of manipulated objects. In addition, annotators may specify contact frames and bind annotated objects to corresponding clips to facilitate consistent clip-level annotations. (2) **Tool control area.** The system provides unidirectional and bidirectional SAM-based segmentation modes, multi-object management via object IDs, and flexible editing operations such as undo, deletion, and object-specific modification. (3) **Shortcut and progress management area.** Keyboard shortcuts support frame navigation, playback control, contact-frame setting, and object binding. Real-time statistics and progress indicators display annotation status. Save-and-load functionality enables interruption and resumption for large-scale workflows.

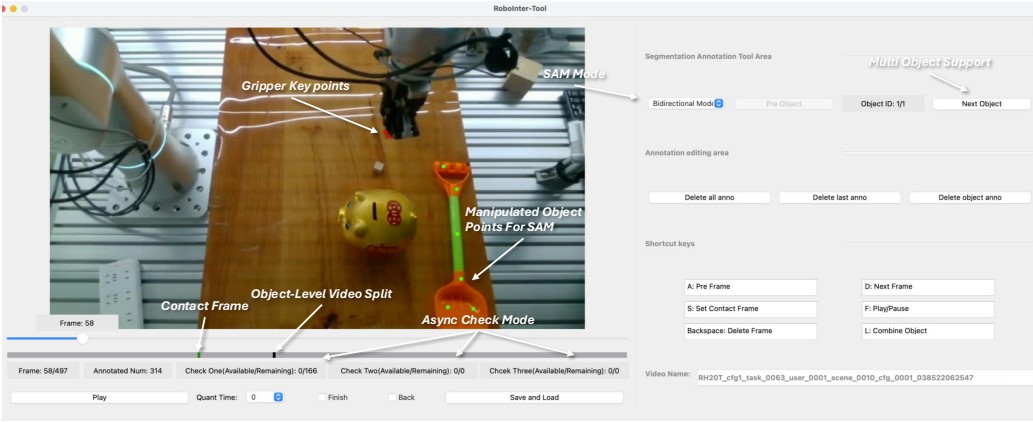

Figure 15: **Semantic Segmentation Annotation Interface.**

**Language Annotation Interface.** It supports both video-level and clip-level natural language descriptions of robotic manipulation videos. This interface is designed to enable multi-granularity annotations, ranging from global video descriptions to fine-grained primitive skill descriptions. As illustrated in Figure 16, the interface is organized into four main components: (1) **Video visualization area.** Similar to the semantic segmentation annotation interface, it supports frame-by-frame playback and provides a keyframe bar for previewing designated keyframes. (2) **Language annotation area.** Annotators can provide and modify global video-level language descriptions that provide high-level summaries of entire videos. For clip-level annotation, annotators define a video segment by marking its start and end frames and then select the corresponding primitive skills from a predefined template set. These templates are further parameterized by object and location, ensuring structured and consistent action descriptions. (3) **Shortcut and progress management area.** The interface provides convenient operations, including frame navigation, playback control, annotation deletion, and language modification via keyboard shortcuts, ensuring flexibility and efficiency during annotation. (4) **Pre/annotation results area.** The interface presents available pre-annotations or

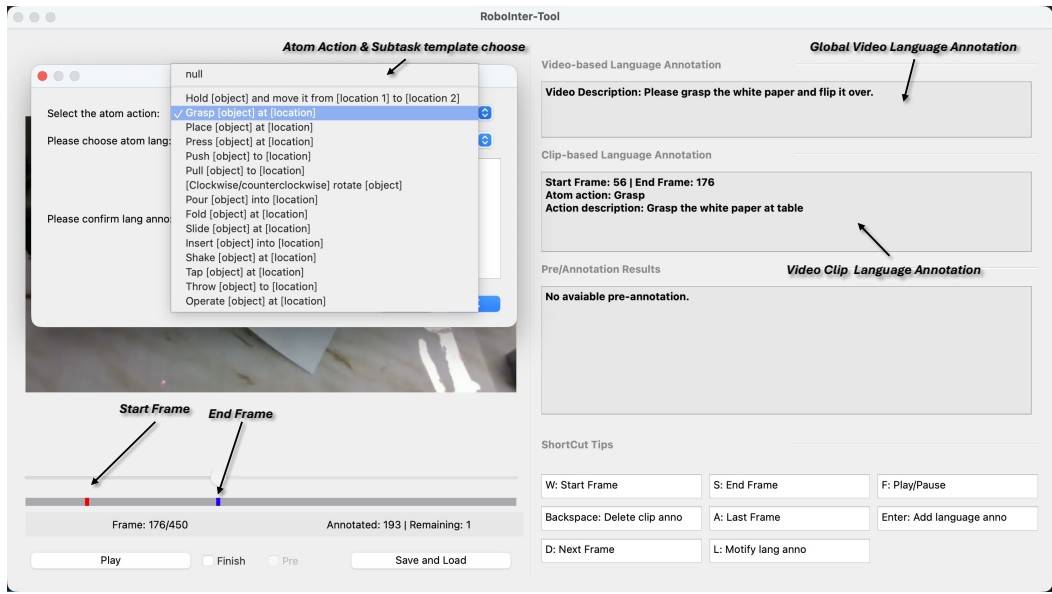

Figure 16: **Language Annotation Interface.**

current results for review, enabling annotators to validate and refine them. This functionality reduces redundant effort in large-scale annotation workflows while enhancing consistency and reusability.

### A.6.2 Principle of Semantic Segmentation Annotation

The semantic segmentation annotation process mainly encompasses three components: interacted object segmentation, contact frames identification, and gripper keypoints labeling. (1) **Interacted Object Segmentation.** Annotators specify prompt points in the area of manipulated objects, which serve as inputs to the SAM2. Based on these prompts, SAM2 produces pixel-level segmentation masks corresponding to the interacted objects. In scenarios involving multiple objects, annotators label each object sequentially according to the order of interaction, while the associated object IDs are automatically recorded, thereby ensuring precise segmentation and consistent tracking across interactions. (2) **Contact Frames.** Annotators identify contact frames, defined as the keyframes when the gripper first establishes physical contact with the target object. If the same object is manipulated multiple times within an episode, annotators use a combine function to bind different contact frames to that object, thereby preserving semantic continuity across disjoint interactions. (3)**Gripper keypoints.** To facilitate downstream keypoint-based optimization, our tool provides functionality for gripper keypoint annotation. Annotators label a predefined set of keypoints on the gripper in a frame-by-frame manner, capturing its motion trajectory. Only keypoints visible under the current camera view are annotated, thereby offering fine-grained geometric constraints that support subsequent 3D-to-2D projection optimization. Together, these three levels of annotation (from pixel-level masks to clip-level contact frames and geometric-level keypoints) form a comprehensive semantic segmentation annotation pipeline.

### A.6.3 Principle of Language Annotation

In the language annotation component, we establish a multi-granularity framework tailored for robotic manipulation videos, covering representations from global video semantics to fine-grained primitive skill. The annotation interface supports three complementary forms of annotation: (1) **Global video-level descriptions.** Annotators provide and refine high-level summaries of the entire task. To improve efficiency and reduce cognitive load, we first generate GPT-based pre-annotations, which are then inspected and corrected by human annotators. (2) **Clip-level descriptions.** Annotators segment each video into subtasks by marking start and end frames and selecting primitive skill categories from a predefined template library. Each template is parameterized by objects and locations, ensuring structured, consistent, and semantically aligned descriptions. *As the raw data from*

*DROID and RH20T often contain brief pauses, we additionally label the first and last non-static frames to improve temporal precision.* (3) **Primitive skills.** Building upon this interface, we define 15 primitive skills as the most elementary and semantically coherent units of robotic behavior. As shown in Table 20, primitive skill covering canonical manipulation behaviors such as *Pick*, *Place*, *Push*, etc. Importantly, a primitive skill is not limited to the single moment of gripper-object contact but also includes its preparation and completion. For example, a *Pick* starts when the gripper approaches the target object and ends when the object is securely held, while a *Place* covers the movement of the object to the target location until release and gripper withdrawal. Simple motions such as repositioning or empty approaches are not treated as independent primitive skills but are merged into neighboring clips. Typically, the first annotated clip begins when the robot initiates purposeful motion, while the final clip extends until the end of the episode, unless the terminal sequence involves static or invalid states, or the gripper is absent. Through this principled segmentation process, all valid primitive skills within each video are exhaustively annotated, ensuring that the dataset achieves completeness in temporal coverage, semantic fidelity, and structural rigor.

Table 20: **Definition and template of 15 primitive skills.**

| Primitive Skill | Definition | Template |
|---|---|---|
| Move with Object | The gripper holds an object and transports it from one location to another. | transfer [object] from [position] to [position] |
| Pick | The gripper approaches and securely picks up a specific object. | pick up [object] [position] |
| Place | The gripper releases and gently sets down an object at a designated location, ensuring stability (distinct from throwing). | place [object] [position] |
| Press | The gripper or tool applies downward force to actuate or press an item such as a button, switch, or block. | press [object] [position] |
| Push | The gripper or tool applies force to move an object away or along a surface in a specified direction. | push [object] [position] |
| Pull | The gripper or tool draws an object closer or along a path, e.g., opening a drawer, refrigerator door, or zipper. | pull [object] [position] |
| Twist | The gripper twists or turns an object clockwise or counterclockwise, such as unscrewing a cap or rotating a box. | twist [object] [clockwise/counterclockwise] |
| Pour | The gripper tilts a container to transfer liquids or granular materials into another container or onto a surface. | pour [object] [position] |
| Fold | The gripper manipulates a flexible material (e.g., fabric, paper) to fold it along one or more creases. | fold [object] [position] |
| Slide | The gripper moves an object laterally across a surface without lifting it, such as sliding items on a table. | slide [object] [position] |
| Insert | The gripper places an object precisely into an opening or slot, e.g., inserting a block into a rod or connecting parts. | insert [object] [position] |
| Shake | The gripper repeatedly oscillates or shakes an object to achieve a desired effect. | shake [object] [position] |
| Strick | The gripper or tool delivers a quick striking motion to an object, e.g., tapping with a hammer or stick. | strike [object] [position] |
| Throw | The gripper propels an object by releasing it during motion, causing it to be cast or dropped abruptly (distinct from placing). | throw [object] [position] |
| Other | Miscellaneous manipulations not covered by the above categories, including minor adjustments or rare operations. | manipulate [object] [position] |

## ChatGPT-4o Prompt to produce preliminary references for language annotations

You are highly skilled in embodiment task planning, breaking down intricate and long-term tasks into distinct primitive skills. All primitive skills are expressed as follows. Think of [object] as an object, think of [position] as a specific location in the environment. We have standardized the applicable scenarios behind each primitive skill — please strictly follow them.

**Primitive Skills:**

1. transfer [object] from [position] to [position]: Move the object from one position to another when it is already in the robot arm gripper.

2. pick up [object] [position]: Move the gripper to a specific position to grab or pick up an object.

3. place [object] [position]: Place the object at a designated position smoothly without sudden release (different from throw).

4. press [object] [position]: Apply pressure or press an object (buttons, switches, blocks, etc.).

5. push [object] [position]: Push an object, suitable for moving it or a handle in a direction.

6. pull [object] [position]: Pull an object (opposite of push), e.g., opening drawers, doors, or zippers.

7. twist [object] clockwise/counterclockwise: Twist or rotate objects (bottle caps, boxes, etc.).

8. pour [object] [position]: Pour liquid or granular material from a container into another container or position.

9. fold [object] [position]: Fold bendable materials (cloth, paper, etc.).

10. slide [object] [position]: Slide objects without lifting them (e.g., sweeping or sliding on a surface).

11. insert [object] [position]: Insert an object into a precise position (e.g., plug, connect block).

12. shake [object] [position]: Shake an object to achieve a specific effect.

13. strike [object] [position]: Strike an object using a tool (e.g., hit a drum with a hammer).

14. throw [object] [position]: Throw an object to a target or release it suddenly (different from place).

15. manipulate [object] [position]: Manipulate or adjust an object in situations not covered above.

**Input Information:**

- Task Description (in language)
- RGB Image (initial observation)

**Instructions:**

- Understand both the Task Description and the Image.
- Identify and sequence the primitive skills required.
- Replace [object] and [position] with specific, concrete nouns — avoid vague terms like "desired location".
- Plans must only use the 15 primitive skills above.
- Prefer using the first 8 primitive skills; use the last 7 only in special cases.
- Stick exactly to the given format.

Tip: You may first itemize the task-related objects to help you plan.

**Examples:**

*Example 1*
Task: "pick rxbar chocolate from bottom drawer and place on counter"; RGB Image: (omitted)
Answer:

```json
{
  "pull the handle of the bottom drawer to the outside of the counter",
  "pick up the rxbar chocolate in the bottom drawer",
  "transfer the rxbar chocolate from the bottom drawer to the top of the counter",
  "place the rxbar chocolate on the counter"
}
```

*Example 2*
Task: "open the fridge, put the tomatoes in the fridge, close the fridge"; RGB Image: (omitted)
Answer:

```json
{
  "pull the handle of the fridge to open the fridge",
  "pick up the tomatoes on the top of the desk",
  "transfer the tomatoes from the desk to the fridge",
  "place the tomatoes on the shelf in the fridge",
  "push the door of the fridge to the position of closing the fridge"
}
```

*Example 3*
Task: "Pour into the mug"; RGB Image: (omitted)
Answer:

```json
{
  "pick up the container on the top of the desk",
  "transfer the container from the top of the desk to the top of the mug",
  "pour the water of the container into the mug"
}
```

**Current Task**

- Task Description: ...
- RGB Image: see the following

The output follows exactly the form in the <Examples>. The output should be formatted as a list of dict objects, including the leading and trailing ```json and ```.

In addition:

- Output the plan again in English (same format).
- At the end of the output, translate the Task Description into English, formatted as <XXX>.

A.6.4   KEYPOINT ALIGNMENT AND ANNOTATION PIPELINE

To accurately capture and leverage gripper positional information in both simulation and real-world settings, we design a systematic keypoint annotation and projection pipeline (Figure 17). The pipeline unifies keypoint definition, EE Link offset calibration, and 3D-to-2D projection, thereby providing a consistent and extensible annotation framework that generalizes across different gripper models and offers a reliable foundation for downstream application.

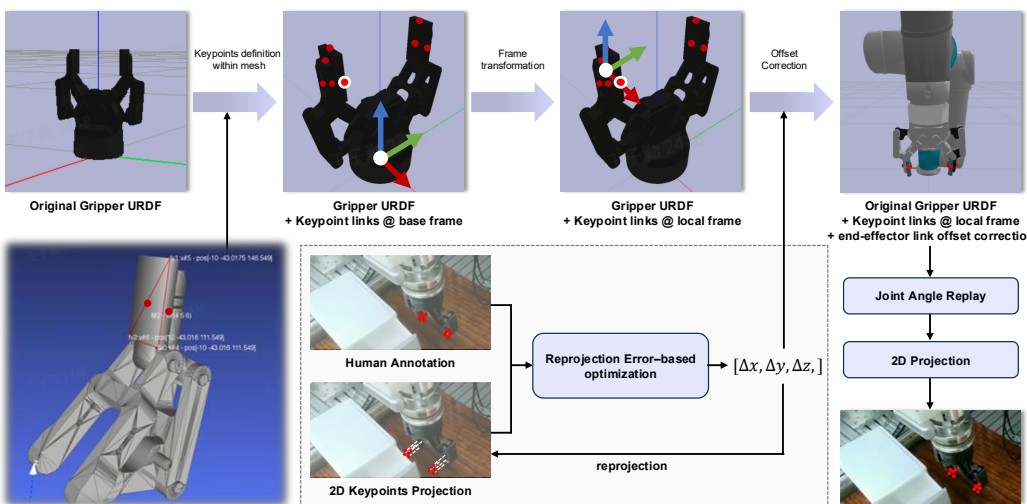

Figure 17: **Keypoint Annotation Pipeline**.

**Keypoint Definition.** We first define a set of representative keypoints for each gripper type according to its geometric characteristics, such as fingertips and the central points of the inner surfaces of the jaws. The locations and numbers of keypoints for different grippers are illustrated in Figure.18. These keypoints are designed to capture the geometric constraints and functional features that are most critical when the gripper interacts with objects or the environment. The process of keypoint definition and annotation consists of two steps: (1) **Computing keypoint coordinates on the mesh.** We first merge gripper link meshes in MeshLab to align their spatial positions with the URDF. From the unified mesh, precise 3D vertex coordinates in the gripper base frame are extracted, and the desired keypoints are annotated on the corresponding links. This ensures geometric accuracy and structural consistency with the real gripper. (2) **Updating the URDF with keypoints.** The computed 3D coordinates are added to the URDF as additional links. Since these coordinates are defined in the base frame rather than local link frames, they cannot be directly driven by joint kinematics. To resolve this, we apply coordinate transformation to convert them into the parent link frames. The updated URDF then correctly propagates keypoint poses during simulation according to joint configurations. (3) **3D-to-2D projection.** To faithfully reproduce each recorded trajectory in simulation, the robot's joint angles and gripper open-close states are used to reconstruct the corresponding motions, from which the 3D coordinates of predefined keypoints are obtained. Leveraging camera intrinsics and extrinsics, 3D keypoints are projected onto the image plane to generate the 2D keypoint annotations.

**EE Link Offset Calibration.** In real robotic systems, a fixed structural offset exists between the gripper and the end-effector link. If the original URDF is used directly in simulation, the reconstructed and projected keypoints deviate significantly from the annotated ground truth. Since this offset is not directly observable during data collection, we adopt an annotation-optimization strategy that estimates and corrects it by minimizing reprojection error. The procedure mainly involves two steps: (1) **2D keypoint annotation.** For each robot configuration, 15 episodes are randomly sampled and keypoints are manually annotated across all camera views, providing supervision for offset estimation. (2) **Offset optimization.** The EE-gripper offset is parameterized as a 3D translation vector and incorporated into the EE-gripper transformation. Given the recorded joint states, robot motions are replayed in simulation, and keypoints are projected onto the image plane using known camera intrinsics and extrinsics. The reprojection error (computed as the Euclidean distance

| Gripper Model | WSG-50 | Robotiq 2F-85 | Franka | AG-95 |
|---|---|---|---|---|
| **Number of Keypoints** | 10 | 10 | 4 | 10 |
| **Visualization of Keypoints** | | | | |

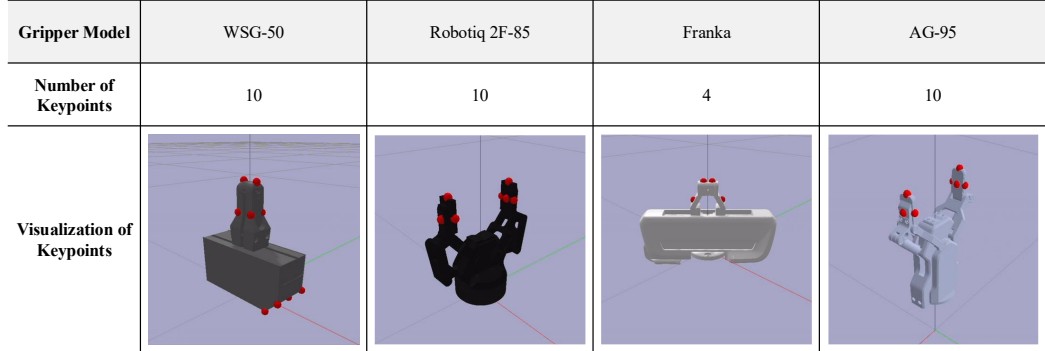

Figure 18: **Keypoint Annotation Visualization**.

between projected and annotated 2D points) is minimized to estimate the optimal offset. The calibrated offset is then integrated into the URDF, thereby aligning the simulated end-effector-gripper geometry with the configuration used during data collection.

**Projection accuracy verification.** We have defined a set of corresponding 3D keypoints, and manually annotated a subset of their 2D trajectories on the 2D image, which serve as the pseudo ground truth. Using the offset calibration, we project the 3D keypoints back onto the image plane and compute the per-pixel L2 distance to the human annotations. As seen in Table 21, across all configurations, the optimization procedure reduces the reprojection error by a large margin (from 46.25 to 5.31 on cfg.1, and 108.21 to 11.61 on cfg.4). This consistent improvement demonstrates that our calibration method significantly enhances 3D-2D geometric consistency.

Table 21: **2D reprojection errors across seven RH20T configurations**. We report reprojection errors ($\downarrow$) before and after optimization.

| Config | cfg_1 | cfg_2 | cfg_3 | cfg_4 | cfg_5 | cfg_6 | cfg_7 |
|---|---|---|---|---|---|---|---|
| Num of Val frames | 1295 | 1357 | 1136 | 1094 | 1017 | 1195 | 1154 |
| Before optimization | 46.25 | 40.63 | 67.35 | 108.21 | 43.68 | 69.71 | 66.24 |
| After optimization | **5.31** | **5.69** | **10.80** | **11.61** | **5.47** | **6.31** | **6.95** |

### A.6.5 QUALITY CONTROL PIPELINE

**Semantic Segmentation Annotation quality control.** For the quality control of semantic segmentation annotations, we implemented a multi-stage verification pipeline combining real-time self-inspection, cross-validation, and sampling-based acceptance tests. During annotation, point prompts for manipulated objects were asynchronously processed on the server by the Segment Anything Model (SAM), and the generated masks were returned for annotator review. Annotators were required to validate all masks, and any object failing three consecutive checks was marked as a hard sample and discarded, preventing low-quality data from entering the dataset. After each annotation round, cross-validation was conducted, with annotators reviewing and correcting each other's work to reduce bias and enhance consistency. In the final acceptance stage, the dataset was divided into 100 subsets, with 50 samples from each subset randomly validated by new annotators. Subsets achieving less than 90% valid annotations were returned for re-annotation. This repair-and-verification process was performed once, resulting in a final dataset with an overall high accuracy.

**Language Annotation quality control.** For language annotations, we established a hybrid pipeline combining automated pre-annotation with multi-stage human verification. Initially, ChatGPT is used for generating draft annotations, which are then inspected and refined by human annotators. A cross-checking stage then followed, where annotators reviewed each other's work to mitigate bias and omissions. The dataset was subsequently divided into 100 subsets, each undergoing sampling-based validation (50 samples per subset) by new annotators. Subsets falling below a 90% acceptance rate were returned for re-annotation. The repair-and-verification cycle was iterated 4 times until all subsets met the acceptance criteria, resulting in a final dataset with an overall high accuracy.

### A.6.6 POSTPROCESSING

**Semantic-Primitive Alignment for Task Descriptions.** In our annotation pipeline, semantic segmentation annotations and primitive skill annotations are defined at different granularities. Contact frames in semantic segmentation annotations are labeled at the progress level, whereas a single subtask may span multiple primitive skills. For instance, in the task *"Pick up the green cup on the table, transfer it above the blue cup, and place it on top"*, only the initial *pick* action contains a contact frame in semantic segmentation annotations, while the primitive-level annotation yields three clips: *pick*, *transfer*, and *place*. This mismatch necessitates alignment between subtask-level language descriptions and semantic segmentation annotations when constructing VQA pairs. To this end, we define clips derived from semantic segmentation annotations as semantic-based clips and those derived from primitive skill annotations as primitive-based clips. For simple tasks containing only a single primitive-based clip, we directly adopt the video-level language description as the task description. For more complex cases, where a semantic-based clip corresponds to multiple primitive-based clips, we extract the start and end timestamps of the semantic-based clip from the raw data, identify all primitive skills whose temporal overlap with this interval exceeds a predefined threshold, and concatenate their descriptions to form the complete task-level language description.

**Spatial and Temporal Correction in DROID.** The episode in DROID is individually calibrated, as its camera intrinsics and extrinsics are not complete and not fully reliable. We find there are misalignments between 2D trace timestamps and video timestamps, as well as positional misalignments. To address these two issues, we introduce complementary spatial position correction and temporal alignment, which together enhance data consistency and quality. The visualization is illustrated in Figure 19. (1) **Position Correction.** Using annotated object masks, we compute object bounding boxes and centroids, and compare them with gripper bounding boxes derived from 2D keypoint projections at contact frame. If the intersection-over-union exceeds a predefined threshold, no correction is applied. Otherwise, we assume that at the contact frame the TCP projection should approximately coincide with the object centroid. For eligible cases, we calculate the 2D offset between the TCP projection and the object centroid, and apply this offset uniformly to the entire trajectory and projected keypoints, thereby correcting systematic positional deviations. Objects box with extreme aspect ratios are excluded from correction. (2) **Temporal Correction.** For temporal alignment, we use CoTracker3 (Karaev et al., 2024) to detect the onset of gripper motion in the video and match it with the corresponding onset frame from TCP trajectory data. The difference between the two defines a correction offset, which is applied to synchronize video and trajectory. After alignment, shorter trajectories are padded and longer ones truncated, ensuring precise frame-level correspondence.

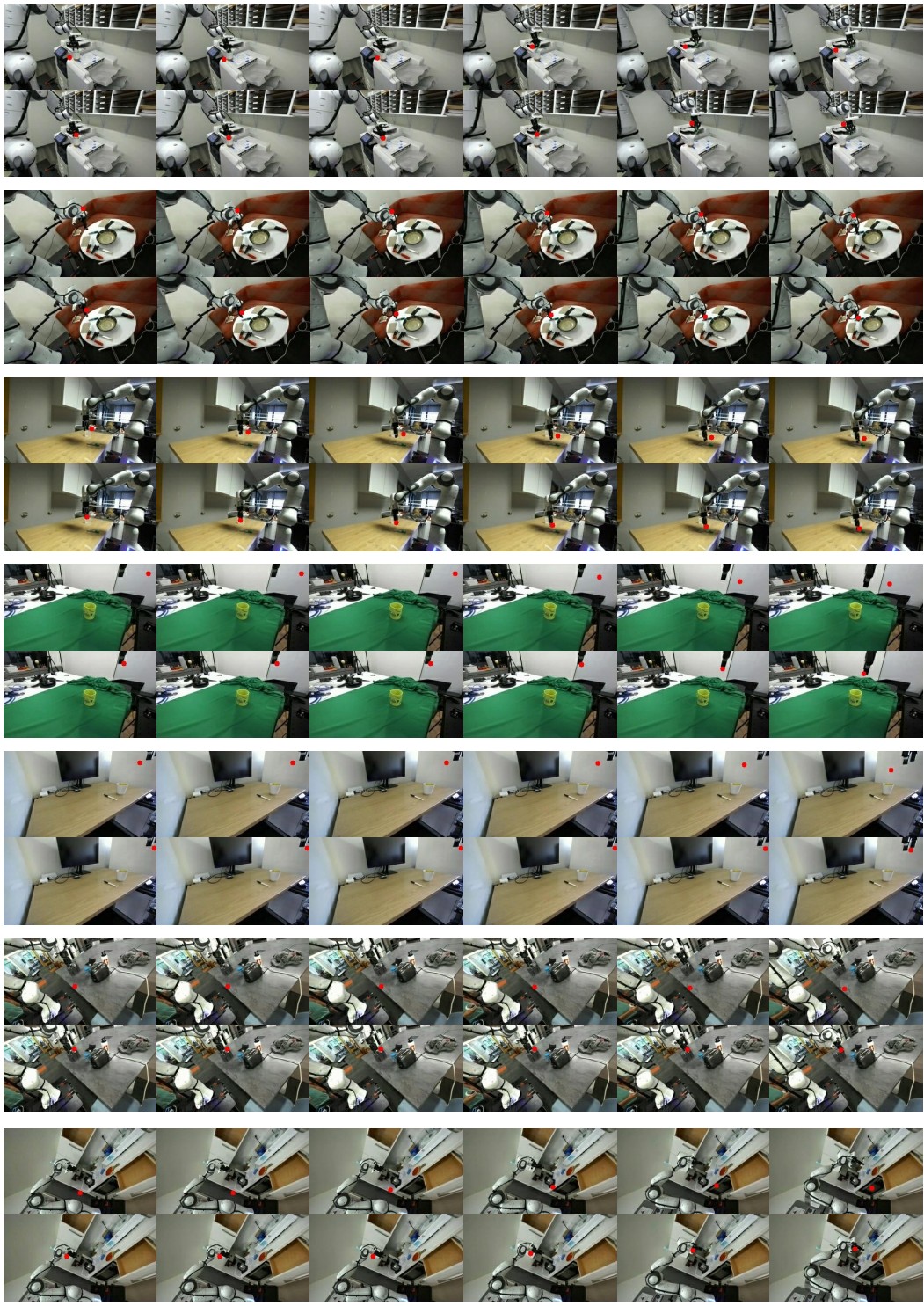

Figure 19: **Visualization of correction results on the DROID dataset.** For each episode, the first row shows the TCP 2D points before correction, while the second row presents the results after applying our correction method.

## A.7 DATASET OVERVIEW

### A.7.1 DATASET STATISTICS

As shown in Table 22, our dataset contains a large number of videos across diverse embodied scenarios, which can be utilized as large-scale pretraining data. Our dataset provides extensive coverage at multiple levels: video-level annotations, such as global language descriptions; frame-level annotations, including subtasks and keyframe counts; and local-region annotations, such as object masks and gripper keypoints. It also supports the full spectrum from generation to understanding, spanning both spatial and temporal domains.

Table 22: **Statistics of the raw dataset for our annotation.**

| Dataset | #Videos | #Objects | #Frames | Avg. Objects / Frame | Avg. Length | Embodiment | Scene Type | Language Seg |
|---|---|---|---|---|---|---|---|---|
| TableTop | 82k | 47M | 40M | 1.17 | 491 | - | - | - |
| In-the-Wild | 152k | 53M | 46M | 1.16 | 302 | - | - | - |
| All | 234k | 100M | 86M | 1.16 | 368 | 6 | 571 | 760k |

### A.7.2 VISUALIZATION FOR INTERMEDIATE REPRESENTATIONS

We visualize several examples of our annotated intermediate representations in Figure 20, Figure 21, Figure 22, and Figure 23. These results show the high diversity and labeling fidelity of our dataset.

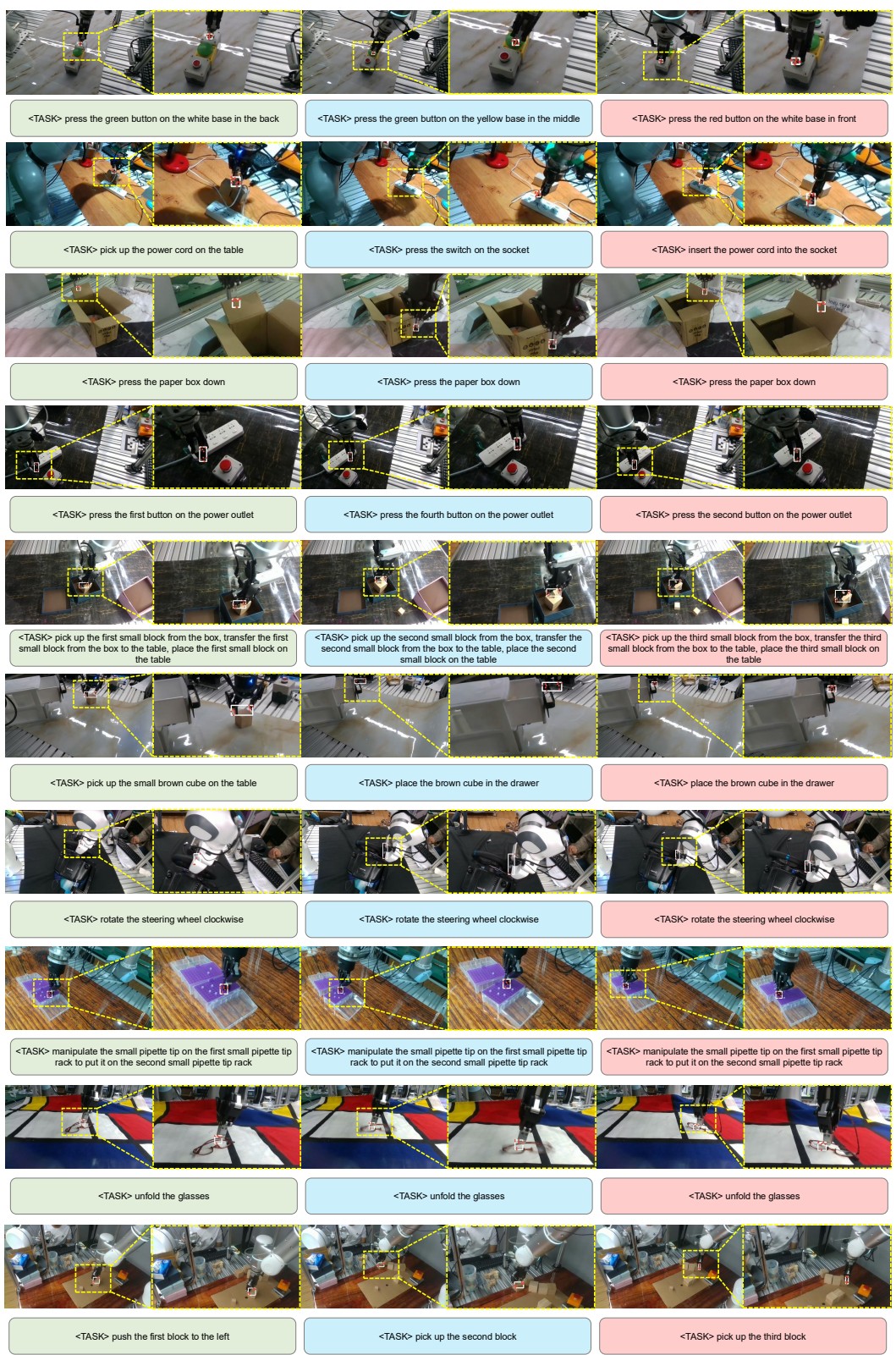

Figure 20: **Visualization of contact points (red dots) and contact boxes (white) in multi-task episodes.** Each subtask is annotated below, with two images per subtask: a global scene (left) and a zoomed-in interaction view (right).

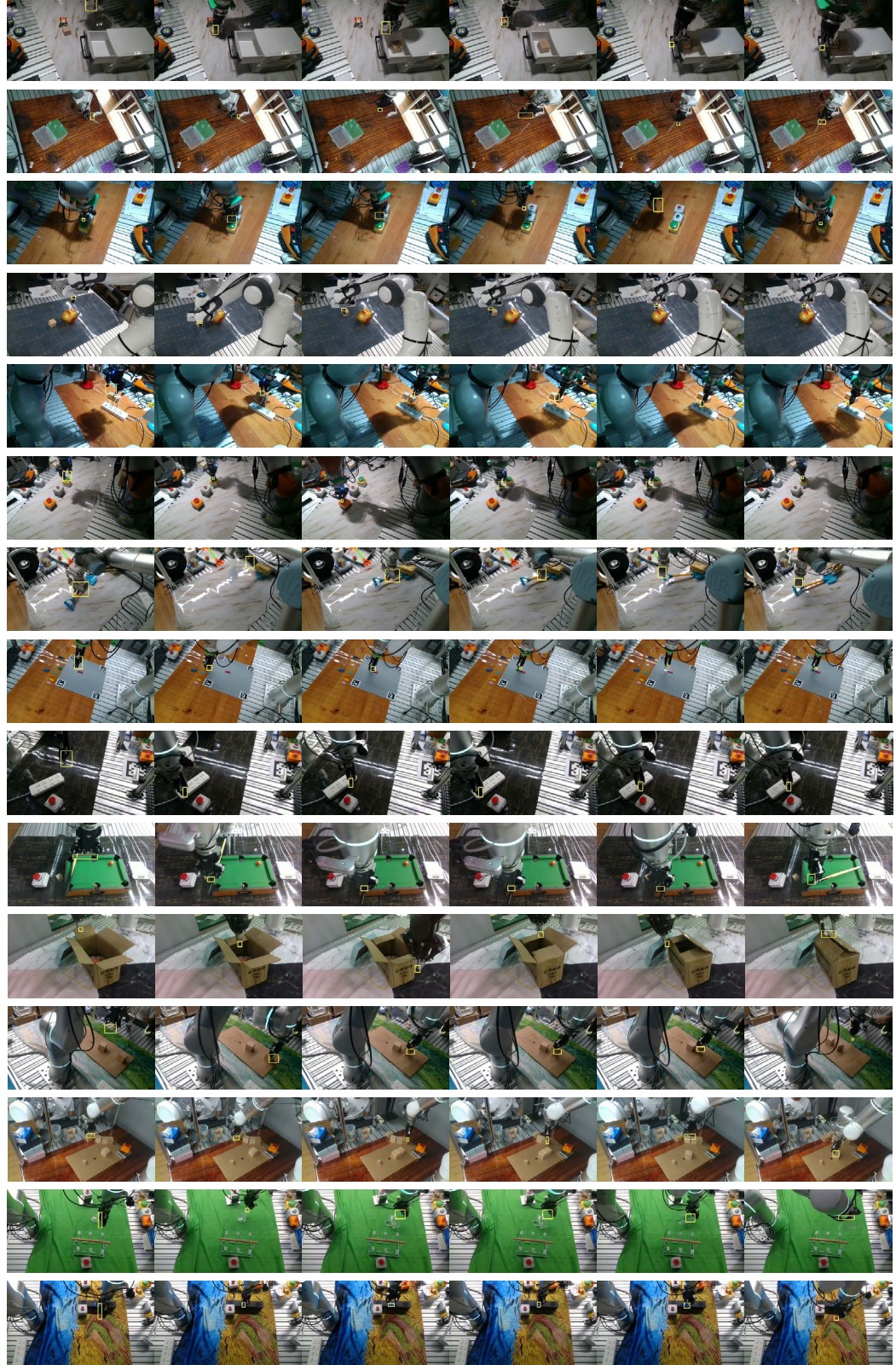

Figure 21: **Visualization of gripper bounding box (yellow) across consecutive frames.**

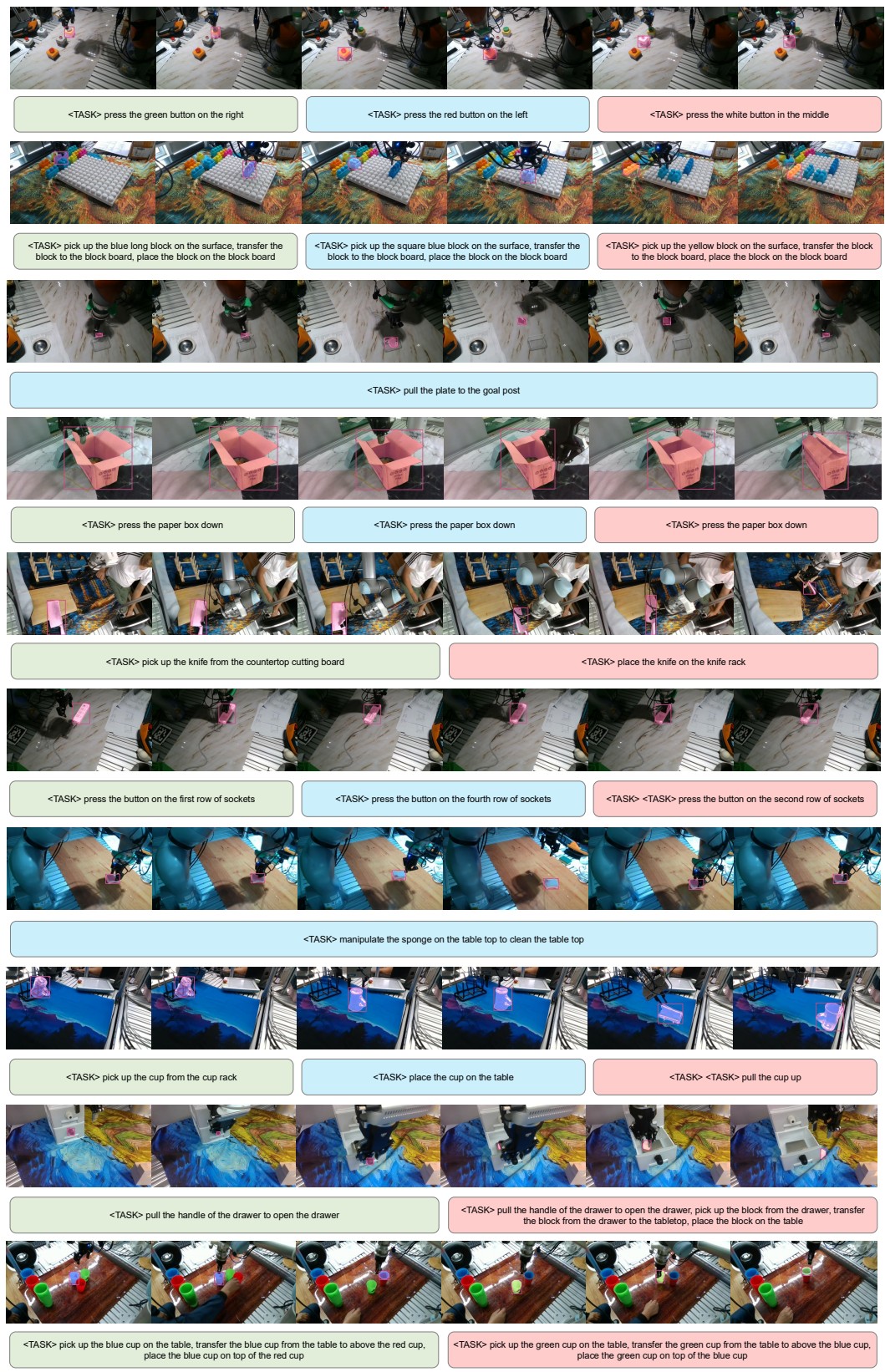

Figure 22: **Visualization of manipulated object segmentation masks and object bounding boxes (magenta) in multi-task episodes.** Each subtask is annotated below.

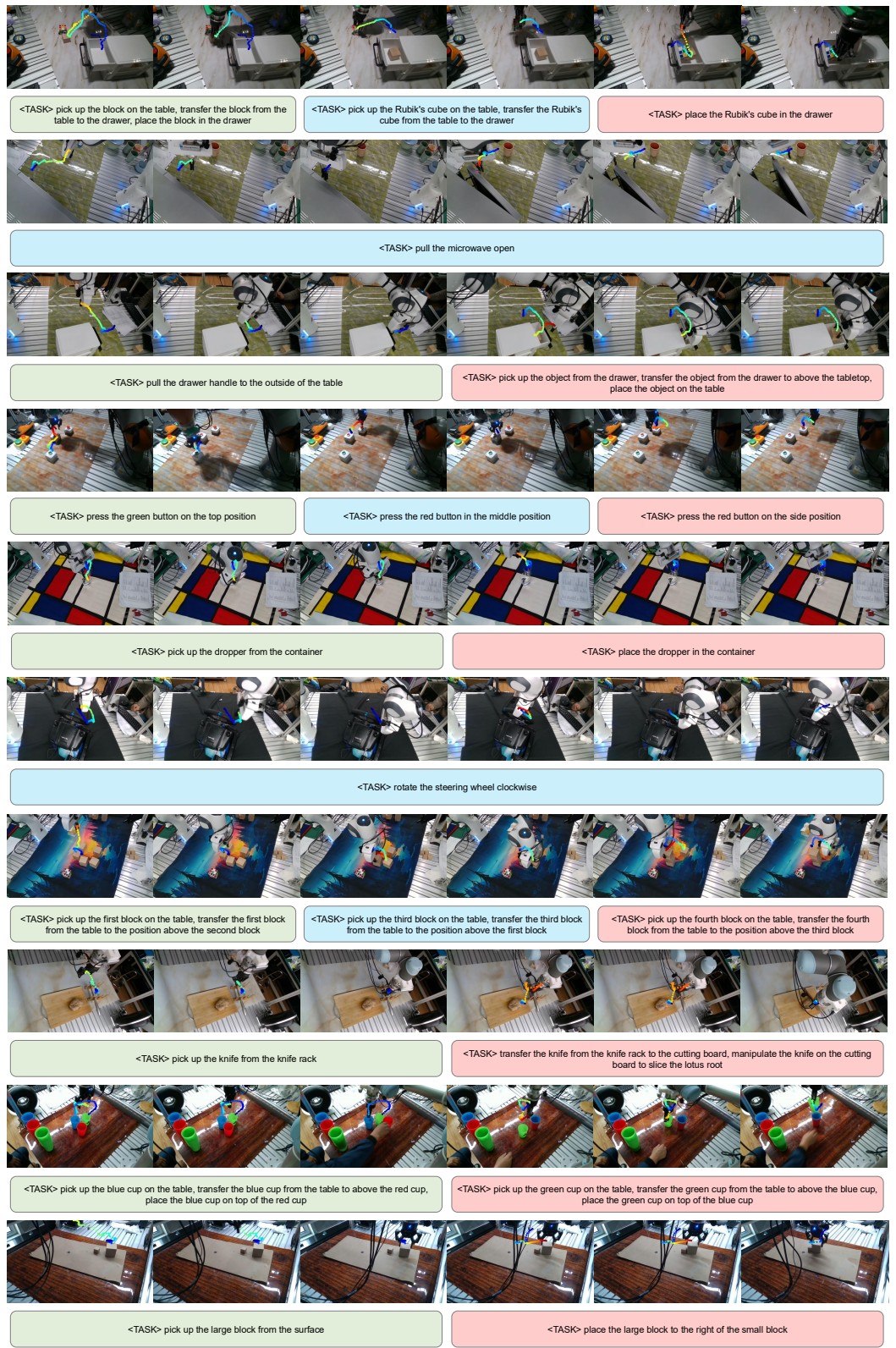

Figure 23: **Visualization of traces in multi-task episodes.** The color gradient encodes temporal progress within each subtask: points closer to red indicate positions near the subtask start, while those closer to blue denote positions approaching the subtask end. Each subtask is annotated below.

### A.7.3 DETAILS OF VQA DATASET

**Statistics.** Figure 24 presents an overview of the language characteristics of our dataset. The word cloud highlights the diversity of language instructions, while the accompanying histogram illustrates the object distribution. Figure 25 reports the length distribution of subtasks, providing insight into the temporal structure of trajectories. Finally, Figure 26 summarizes multiple aspects of the dataset, including the number distribution of subtasks, language token length distribution, skill distribution, and embodiment distribution, offering a comprehensive view of task complexity and multimodality.

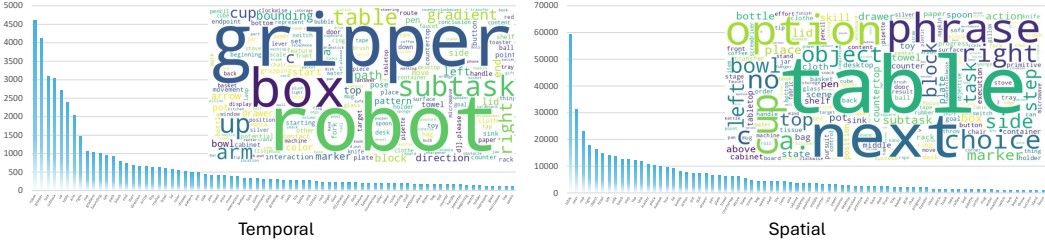

Figure 24: **Word cloud and object distribution.**

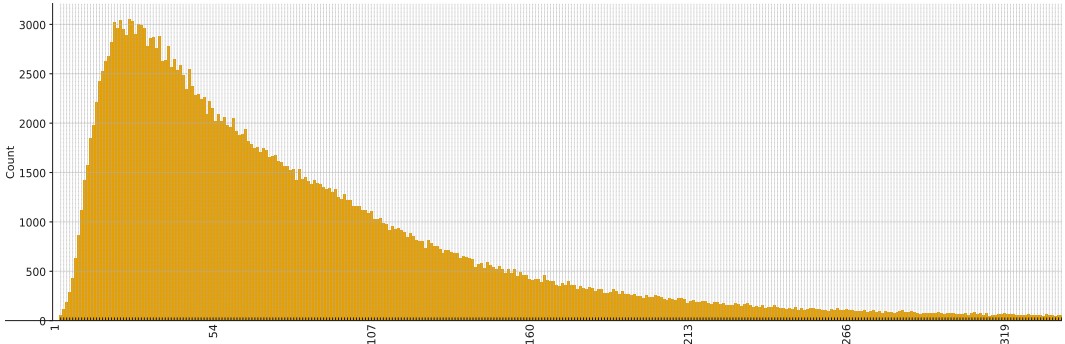

Figure 25: **Transition step-length distribution.**

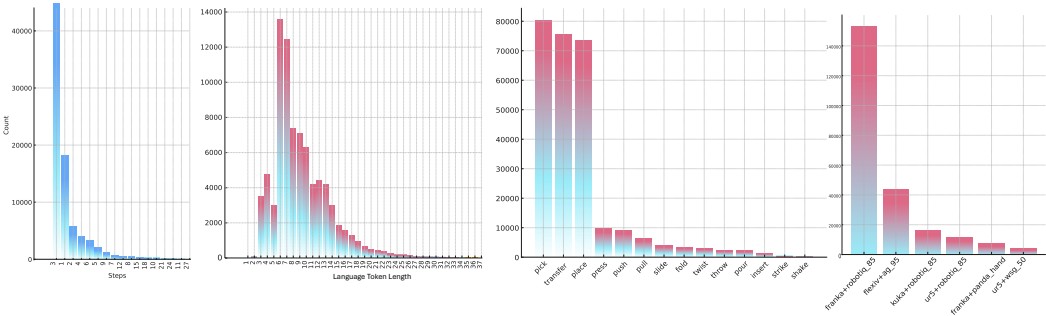

Figure 26: **More statistics.** From left to right: the subtasks number distribution, language token length distribution, skill distribution, and embodiment distribution.

Table 23: **Statistics of the constructed VQA on visual intermediate representation.** The choice and decision stand for the abilities of Understanding.

| Task Name | Task Type | Class | Number |
|---|---|---|---|
| Trace | Temporal&Choice | Tabletop | 3,610 |
| | | In-the-Wild | 8,245 |
| Grounding | Spatial&Choice | Tabletop | 8,158 |
| | | In-the-Wild | 57,572 |
| Trace dir | Temporal&Choice | Tabletop | 3,729 |
| | | In-the-Wild | 6,500 |
| Trace lang | Temporal&Choice | Tabletop | 3,610 |
| | | In-the-Wild | 8,245 |
| Contact | Spatial&Decide | Tabletop | 15,060 |
| | | In-the-Wild | 18,184 |
| Grounding | Spatial&Generation | Tabletop | 225,055 |
| | | In-the-Wild | 149,671 |
| Trace easy | Temporal&Generation | Tabletop | 33,803 |
| | | In-the-Wild | 31,282 |
| Trace hard | Temporal&Generation | Tabletop | 33,803 |
| | | In-the-Wild | 31,282 |
| Grasp Affordance (Box Mode) | Spatial&Generation | Tabletop | 115,266 |
| | | In-the-Wild | 78,004 |
| Grasp Affordance (Key point Mode) | Spatial&Generation | Tabletop | 115,266 |
| | | In-the-Wild | 78,004 |
| Gripper Detection | Spatial&Generation | Tabletop | 120,747 |
| | | In-the-Wild | 84,777 |
| Place Affordance | Spatial&Generation | Tabletop | 145,996 |
| | | In-the-Wild | 224,944 |
| Grasp pose | Spatial&Choice | Tabletop | 9,835 |

Table 24: **Statistics of the constructed VQA on task planning.** The choice and decision stand for the abilities of Understanding.

| Task Name | Task Type | Number |
|---|---|---|
| Planning_Task | Temporal & Generation | 82,939 |
| Planning_with_Context_Task | Temporal & Generation | 58,439 |
| Planning_Remaining_Steps_Task | Temporal & Generation | 58,439 |
| Generative_Affordance_Task | Temporal & Generation | 61,804 |
| Future_Prediction_Task | Temporal & Generation | 61,804 |
| Success_Negative_Task | Temporal & Decide | 79,929 |
| Success_Positive_Task | Temporal & Decide | 57,413 |
| Discriminative_Affordance_Positive_Task | Temporal & Decide | 39,060 |
| Discriminative_Affordance_Negative_Task | Temporal & Decide | 64,814 |
| Past_Description_Task | Temporal & Generation | 57,413 |
| Past_Multi_task_Selection | Temporal & Choice | 76,564 |
| Future_Multi_task_Selection | Temporal & Choice | 58,439 |
| Scene_Understanding | Spatial & Choice | 57,413 |
| Temporal_Understanding | Temporal & Choice | 6,585 |
| Past_Primitive_Selection | Temporal & Choice | 76,564 |
| Future_Primitive_Selection | Temporal & Choice | 58,439 |

**Spatial VQA.** We provide the annotation templates and visualize a subset of the Spatial VQA results, as shown below:

- **Grasp Pose Choice Task.** Evaluates whether the model can identify the correct grasp configuration among candidate options. Images are annotated with orange fork-like gripper patterns that represent potential gripper poses. The model must select the one that corresponds to the correct grasping pose. Visualizations are included in Figure.27.

- **Scene Understanding.** Requires selecting the correct scene (from images) that matches the long-horizon goal, assessing spatial grounding and environment recognition. Visualizations are included in Figure.37.

- **Contact Decide Task.** Assesses the model's judgment of physical contact between the gripper and the object. Given an image during a subtask, the model decides whether the gripper has already made contact, with binary choices [Yes, No]. Visualizations are included in Figure.27.

- **Grounding Choice Task.** Tests the model's ability to localize the correct object for interaction. Images include purple bounding boxes indicating possible target objects, and the model must choose the bounding box corresponding to the task-relevant object. Visualizations are included in Figure.28.

---

**Grasp Pose Choice Task**

- The robot task is <TASK>. We provide some images that feature orange fork-like gripper patterns; these patterns stand for the potential poses the gripper needs to adopt to pick up an object. Which of the images below contains the correct grasping pose? The available choice is <CHOICE>. Please select one option only.
- The robot task is <TASK>. Among the images we offer, there are orange fork-like gripper patterns—they represent the possible poses the gripper may use to grab an object. Which of the following images has the correct grasping pose? The available choice is <CHOICE>.Please select one option only.
- The robot task is <TASK>. We present some images with orange fork-like gripper patterns. These patterns correspond to the potential poses the gripper requires to pick up an object—so which image below contains the correct grasping pose? The available options are <CHOICE>. Please select one option only.
- The robot task is <TASK>. In the images we provide, there are orange fork-like gripper patterns: they represent the possible poses the gripper needs to use to pick up an object. Which of the following images contains the correct grasping pose? The available choices are <CHOICE>. Please select one option only.
- The robot task is <TASK>. We present some images annotated with orange fork-like gripper patterns. These patterns correspond to the potential poses the gripper may adopt to grab an object—so which image below has the right grasping pose? The available choices are <CHOICE>. Please select one option only.
- The robot task is <TASK>. We supply some images that include orange fork-like gripper patterns; these patterns represent the poses the gripper might need to take to pick up an object. Which of the images below contains the correct grasping pose? The available choices are <CHOICE>. Please select one option only.
- The robot task is <TASK>. Among the images we present, there are orange fork-like gripper patterns; they correspond to the potential poses the gripper needs to use to grab an object. Which image below has the correct grasping pose? The available choices are <CHOICE>. Please select one option only.
- The robot task is <TASK>. We offer some images featuring orange fork-like gripper patterns. These patterns stand for the poses the gripper might adopt to pick up an object—so which of the following images contains the correct grasping pose? The available choices are <CHOICE>. Please select one option only.
- The robot task is <TASK>. We provide some images that include orange fork-like gripper patterns; these patterns stand for the potential postures the gripper may use to grab an object. Which image below has the correct grasping pose? The available choices are <CHOICE>. Please select one option only.
- The robot task is <TASK>. The images we supply contain orange fork-like gripper patterns: these patterns represent the potential poses the gripper may rely on to grab an object. Which of the images below has the right grasping pose? The available choices are <CHOICE>. Please select one option only.

*Answer:* <CHOICE>

---

**Scene Understanding**

- Given four images labeled A, B, C, D, which image best corresponds to the scene required for accomplishing {long-horizon}? Please answer with a single option (A/B/C/D).
- Which of the images (A, B, C, D) best matches the scene necessary to achieve {long-horizon}? Please answer with a single option (A/B/C/D).
- Looking at the four images A, B, C, D, which one best represents the scene for {long-horizon}? Please answer with a single option (A/B/C/D).
- Among the four images (A, B, C, D) provided, which scene corresponds to {long-horizon}? Please answer with a single option (A/B/C/D).
- Which image (A, B, C, D) most accurately depicts the scene for {long-horizon}? Please answer with a single option (A/B/C/D).

*Answer:* A/B/C/D

## Contact Decide Task

- The robot task is <TASK>, and now you are performing subtask <SUBTASK>. In this image, do you think the gripper has already made contact with the object? The available choices are [Yes, No]. Please select one option only.
- The robot task is <TASK>, and now you are performing subtask <SUBTASK>. In this image, has the gripper made contact with the object? The available choices are [Yes, No]. Please select one option only.
- The robot task is <TASK>, and now you are performing subtask <SUBTASK>. In this image, do you think the gripper has already touched the object? The available choices are [Yes, No]. Please select one option only.
- The robot task is <TASK>, and now you are performing subtask <SUBTASK>. In this image, has the gripper touched the object? The available choices are [Yes, No]. Please select one option only.
- The robot task is <TASK>, and now you are performing subtask <SUBTASK>. In this image, do you think the gripper has already reached the object? The available choices are [Yes, No]. Please select one option only.
- The robot task is <TASK>, and now you are performing subtask <SUBTASK>. In this image, has the gripper reached the object? The available choices are [Yes, No]. Please select one option only.
- The robot task is <TASK>, and now you are performing subtask <SUBTASK>. In this image, do you think the gripper has already approached the object? The available choices are [Yes, No]. Please select one option only.
- The robot task is <TASK>, and now you are performing subtask <SUBTASK>. In this image, has the gripper approached the object? The available choices are [Yes, No]. Please select one option only.

*Answer:* Yes / No

## Grounding Choice Task

- The robot task is <TASK>. We offer some images containing purple boxes, and these boxes correspond to the objects that maybe interact with the robotic arm, which image describe the right object bounding box? The available choice is [<CHOICE>]. Please select one option only.
- The robot task is <TASK>. The images we present contain purple bounding boxes, which stand for the objects that the robotic arm maybe interacts with, which image depicts the correct object bounding box? The available choice is [<CHOICE>].Please select one option only.
- The robot task is <TASK>. Among the provided images, purple boxes are present to represent the objects that maybe engage with the robotic arm, which of the images shows the proper bounding box for the object? The available options are [<CHOICE>]. Please select one option only.
- The robot task is <TASK>. We offer some images where purple boxes are used to indicate the objects maybe interacting with the robotic arm, which image accurately describes the object's bounding box? The available choices are [<CHOICE>]. Please select one option only.
- The robot task is <TASK>. We present a set of images with purple boxes, and these boxes represent the objects maybe involved in the robotic arm's interactions, which image contains the correct bounding box for the target object? The available choices are [<CHOICE>]. Please select one option only.
- The robot task is <TASK>. We provide several images featuring purple boxes, which signify the objects that the robotic arm maybe interacts with, which image correctly illustrates the bounding box of the object? The available choices are [<CHOICE>]. Please select one option only.
- The robot task is <TASK>. The images we provide include purple boxes that denote the objects maybe interacting with the robotic arm, which image accurately represents the bounding box of the object? The available choices are [<CHOICE>]. Please select one option only.
- The robot task is <TASK>. We present images containing purple boxes, which indicate the objects that the robotic arm maybe interacts with, which image correctly shows the bounding box of the object? The available choices are [<CHOICE>]. Please select one option only.
- The robot task is <TASK>. We provide images with purple boxes that represent the objects involved in the robotic arm's interactions, which image accurately depicts the bounding box of the object? The available choices are [<CHOICE>]. Please select one option only.
- The robot task is <TASK>. The images we present feature purple boxes that signify the objects interacting with the robotic arm, which image correctly illustrates the bounding box of the object? The available choices are [<CHOICE>]. Please select one option only.

*Answer:* <CHOICE>

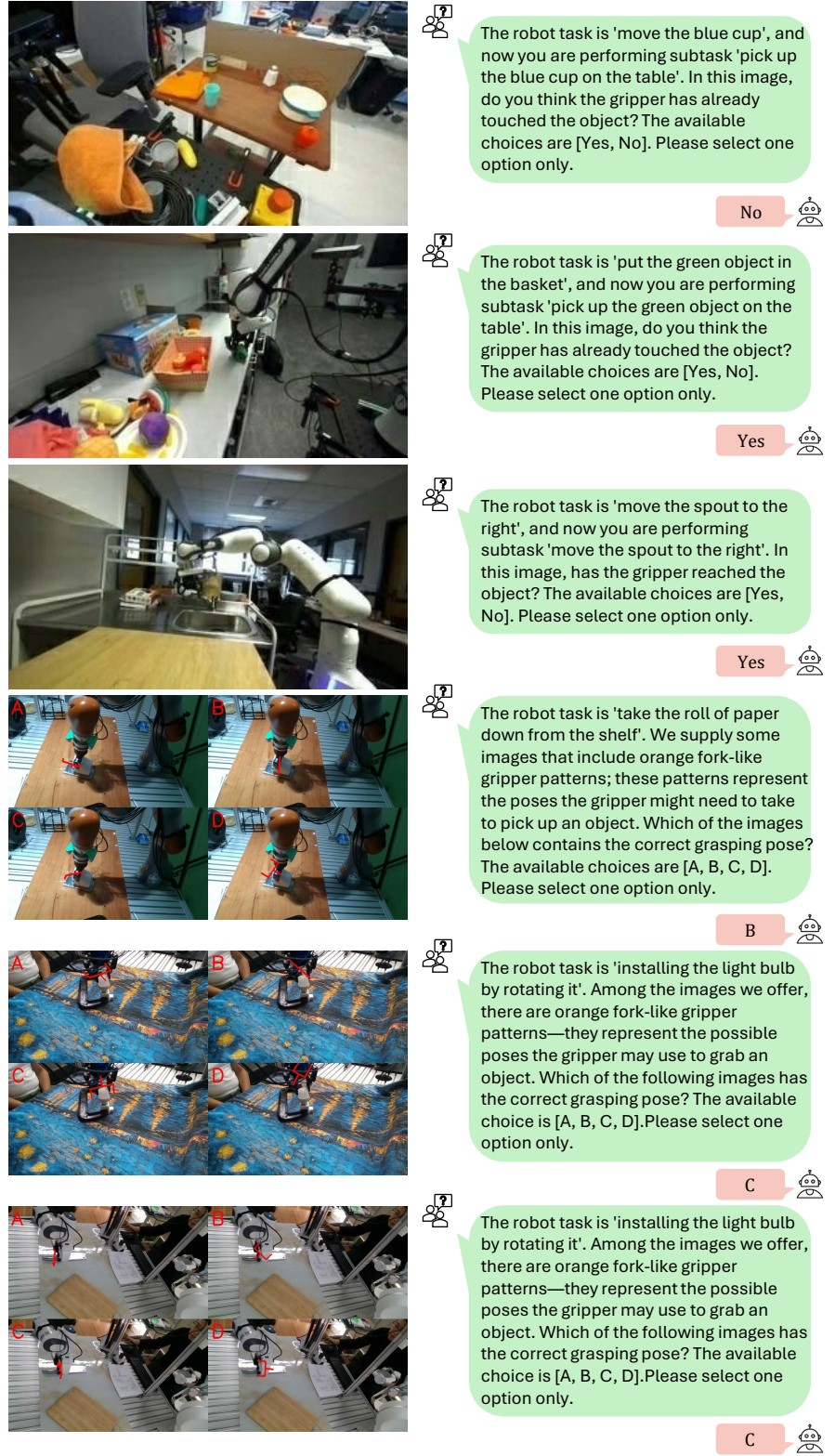

Figure 27: Contact decide task and grasp pose choice task.

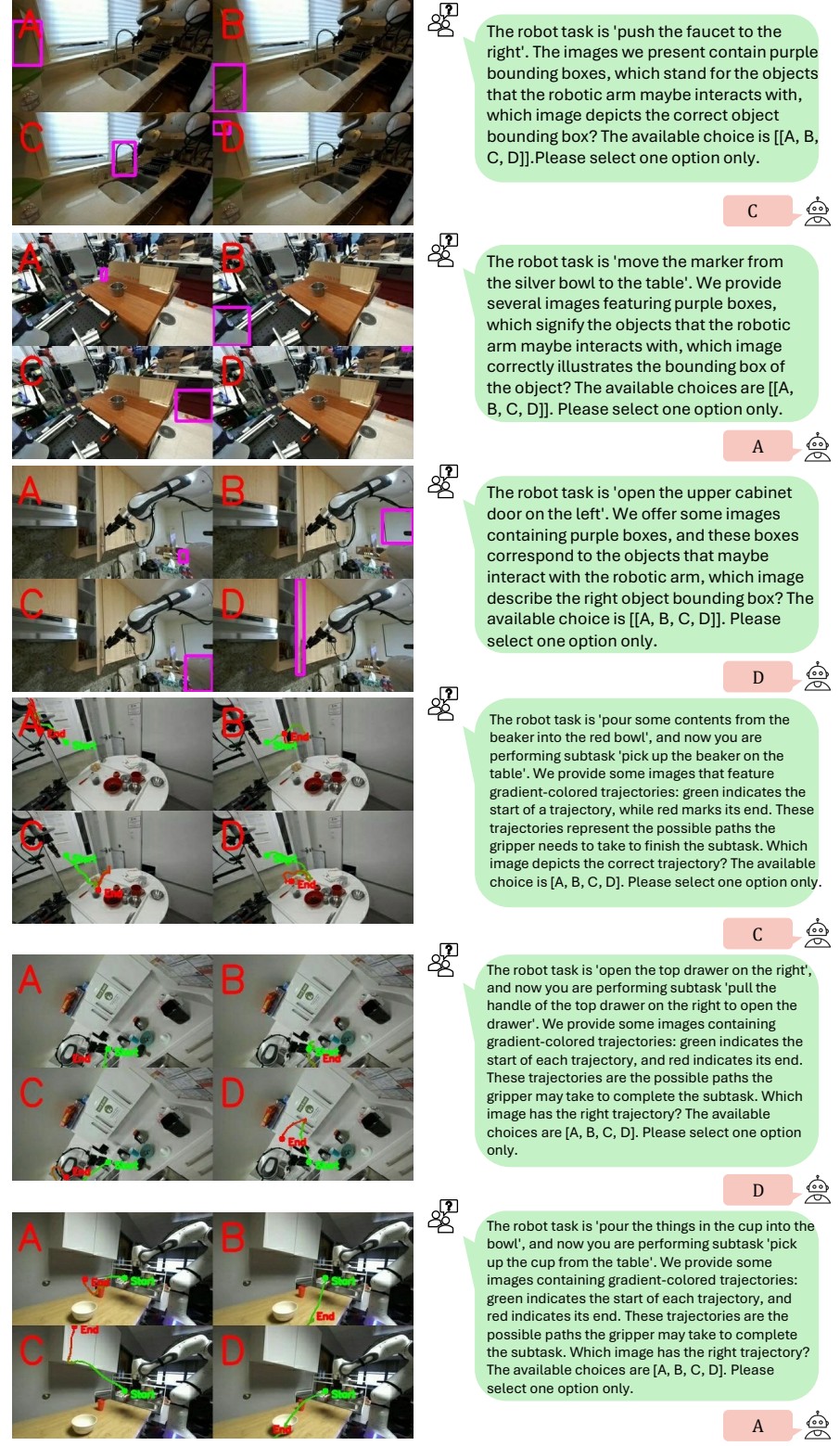

Figure 28: Grounding choice task and trajectory choice task.

**Temporal VQA.** We detailed the 18 types of temporal VQAs, aiming to evaluate the models' sequential understanding and decision-making abilities.

- **Trajectory Choice Task.** Evaluates the model's understanding of motion planning. Images depict candidate trajectories using a green-to-red gradient, and the model must identify the correct trajectory the gripper should follow to complete the subtask. Visualizations are included in Figure.28.

- **Trajectory Direction Choice Task.** Focuses on predicting the immediate motion direction of the gripper. Images display multiple colored arrows around the gripper, and the model must select which arrow correctly indicates the actual move direction. Visualizations are included in Figure.29.

- **Trajectory Sub-goal choice Task.** Requires the model to match a demonstrated trajectory with its corresponding subtask. Gradient-colored trajectories (green start, red end) are shown, and the model must decide which subtask best aligns with the trajectory among candidate options. Visualizations are included in Figure.30.

- **Discriminative affordance negative task.** Evaluates whether a proposed task is not the correct next action, requiring the model to reject infeasible affordances with a strict "No". Visualizations are included in Figure.31.

- **Discriminative affordance positive task.** Tests the model's ability to identify the correct next action among possible options, confirming feasible affordances with a strict "Yes". Visualizations are included in Figure.31.

- **Future multitask selection.** Given a long-horizon goal, asks the model to select the most appropriate next subtask from multiple choices, emphasizing decision-making under alternatives. Visualizations are included in Figure.33.

- **Future prediction task.** Requires anticipating the outcome or immediate follow-up task after completing a given step, testing temporal foresight in task sequences. Visualizations are included in Figure.34.

- **Future primitive selection task.** Focuses on predicting the next primitive skill to apply (e.g., grasp, push, move) to achieve a long-horizon objective, ensuring fine-grained planning ability. Visualizations are included in Figure.32.

- **Generative affordance task.** Prompts the model to generate the most feasible immediate action in the current state, highlighting short-horizon decision-making. Visualizations are included in Figure.32.

- **Past description task.** Asks the model to recall the most recently completed task, evaluating memory of temporal progress. Visualizations are included in Figure.34.

- **Past multitask selection.** Provides multiple subtasks and requires the model to identify which one has already been completed, testing retrospective recognition of task history. Visualizations are included in Figure.33.

- **Past primitive selection.** Requires identifying the most recently executed primitive skill, reinforcing low-level action traceability. Visualizations are included in Figure.35.

- **Planning remaining steps task.** After partial progress, asks for the next five sequential steps toward the goal, testing structured long-horizon planning ability. Visualizations are included in Figure.35.

- **Planning task.** Directly queries the model for the immediate next step toward a long-horizon goal without additional context, serving as the simplest planning probe. Visualizations are included in Figure.36.

- **Planning with context task.** Builds on prior completed tasks to decide the most logical next step, testing consistency and coherence in long-horizon reasoning. Visualizations are included in Figure.36.

- **Success negative task.** Asks the model to verify that a given task was not successfully completed, probing its ability to detect failure.

- **Success positive task.** Asks the model to confirm that a given task was successfully completed, probing its ability to detect success.

- **Temporal understanding.** Given multiple images, it requires identifying the one that corresponds to the execution state of a task, evaluating temporal grounding and process awareness. Visualizations are included in Figure.37.

---

**Trajectory Choice Task**

- The robot task is `<TASK>`, and now you are performing subtask `<SUBTASK>`. We provide some images that feature gradient-colored trajectories: green indicates the start of a trajectory, while red marks its end. These trajectories represent the possible paths the gripper needs to take to finish the subtask. Which image depicts the correct trajectory? The available choice is `<CHOICE>`. Please select one option only.
- The robot task is `<TASK>`, and now you are performing subtask `<SUBTASK>`. We offer a set of images containing gradient-colored trajectories. Green indicates the start of each trajectory, and red indicates its end—these trajectories represent the potential routes for the gripper to finish the subtask. Which image depicts the right trajectory? The available choice is `<CHOICE>`.Please select one option only.
- The robot task is `<TASK>`, and now you are performing subtask `<SUBTASK>`. The images we supply include gradient-colored trajectories: green denotes the start of a trajectory, and red denotes its end. These trajectories stand for the paths the gripper might need to follow to accomplish the subtask. Which image has the correct trajectory? The available options are `<CHOICE>`. Please select one option only.
- The robot task is `<TASK>`, and now you are performing subtask `<SUBTASK>`. We offer a set of images containing gradient-colored trajectories. Green indicates the start of each trajectory, and red indicates its end—these trajectories represent the potential routes for the gripper to finish the subtask. Which image depicts the right trajectory? The available choices are `<CHOICE>`. Please select one option only.
- The robot task is `<TASK>`, and now you are performing subtask `<SUBTASK>`. In the images we provide, there are gradient-colored trajectories: green represents the start point of a trajectory, and red represents its end point. These trajectories correspond to the paths the gripper may need to take to complete the subtask. Which image shows the correct trajectory? The available choices are `<CHOICE>`. Please select one option only.
- The robot task is `<TASK>`, and now you are performing subtask `<SUBTASK>`. We supply some images that include gradient-colored trajectories—green stands for the beginning of a trajectory, and red for its end. These trajectories represent the potential paths the gripper needs to follow to finish the subtask. Which image has the right trajectory? The available choices are `<CHOICE>`. Please select one option only.
- The robot task is `<TASK>`, and now you are performing subtask `<SUBTASK>`. We provide some images with gradient-colored trajectories. Green represents the start of each trajectory, red represents its end, and these trajectories correspond to the paths the gripper needs to take to accomplish the subtask. Which image shows the right trajectory? The available choices are `<CHOICE>`. Please select one option only.
- The robot task is `<TASK>`, and now you are performing subtask `<SUBTASK>`. Among the images we present, gradient-colored trajectories are included—green stands for a trajectory's start, and red for its end. These trajectories represent the potential paths required for the gripper to finish the subtask. Which image has the correct trajectory? The available choices are `<CHOICE>`. Please select one option only.
- The robot task is `<TASK>`, and now you are performing subtask `<SUBTASK>`. We provide some images containing gradient-colored trajectories: green indicates the start of each trajectory, and red indicates its end. These trajectories are the possible paths the gripper may take to complete the subtask. Which image has the right trajectory? The available choices are `<CHOICE>`. Please select one option only.
- The robot task is `<TASK>`, and now you are performing subtask `<SUBTASK>`. We present a set of images with gradient-colored trajectories—green marks where a trajectory begins, and red marks where it ends. These trajectories stand for the potential paths the gripper requires to accomplish the subtask. Which image shows the correct trajectory? The available choices are `<CHOICE>`. Please select one option only.

*Answer:* `<CHOICE>`

---

**Trajectory Direction Choice Task**

- The robot task is `<TASK>`, and now you are performing subtask `<SUBTASK>`. There are some different color arrows around gripper on the image, which color most likely represents the actual move direction of the robot gripper? Please choose only one option from `<CHOICE>`.
- The robot task is `<TASK>`, and now you are performing subtask `<SUBTASK>`. There are several arrows with different colors around the robot gripper on the image, which color describes the actual move direction of the robot gripper? Please choose only one option from `<CHOICE>`.
- The robot task is `<TASK>`, and now you are performing subtask `<SUBTASK>`. On the image, different arrows surround the gripper—some near its jaws, others by the target part. Which color most likely shows the gripper's actual move direction? Please choose only one option from `<CHOICE>`.
- The robot task is `<TASK>`, and now you are performing subtask `<SUBTASK>`. Several colored arrows are around the robot gripper in the image; some point toward the box, others away. Which color describes the gripper's real move direction? Please choose only one option from `<CHOICE>`.
- The robot task is `<TASK>`, and now you are performing subtask `<SUBTASK>`. Different arrows are around the gripper in the image—upward and downward ones. Which color most likely represents its actual movement direction? Please choose only one option from `<CHOICE>`.
- The robot task is `<TASK>`, and now you are performing subtask `<SUBTASK>`. On the image, various arrows surround the gripper—some point toward the box, others away. Which color most likely shows the gripper's actual move direction? Please choose only one option from `<CHOICE>`.
- The robot task is `<TASK>`, and now you are performing subtask `<SUBTASK>`. Several colored arrows are around the robot gripper in the image; some point up, others down. Which color describes its actual move direction? Please choose only one option from `<CHOICE>`.
- The robot task is `<TASK>`, and now you are performing subtask `<SUBTASK>`. Several arrows with different colors are around the gripper; some are horizontal, others vertical. Which color describes its actual move direction? Please choose only one option from `<CHOICE>`.

*Answer:* `<CHOICE>`

**Trajectory Subchoice Task**

- You are performing `<TASK>`. Now we provide the image that feature gradient-colored trajectories: green indicates the start of a trajectory, while red marks its end. The trajectory represent the path the gripper needs to take to finish the subtask. Which subtask is the best match for the trajectory among the following options: `<CHOICE>`? Please select one option only.
- You are performing `<TASK>`. Below is an image showing gradient-colored trajectories, where green signifies the starting point and red indicates the endpoint. The trajectory illustrates the path the gripper should follow to complete the subtask. Among the options `<CHOICE>`, which subtask does the trajectory best correspond to? Please select one option only.
- You are performing `<TASK>`. The image below displays gradient-colored trajectories, with green representing the start and red the end. These trajectories depict the path the gripper must take to accomplish the subtask. From the choices `<CHOICE>`, which subtask aligns best with the trajectory? Please select one option only.
- You are performing `<TASK>`. The image provided shows gradient-colored trajectories, where green marks the beginning and red the conclusion. These trajectories outline the path the gripper needs to follow to complete the subtask. Which subtask from the options `<CHOICE>` does the trajectory best represent? Please select one option only.
- You are performing `<TASK>`. The image below features gradient-colored trajectories, with green indicating the start and red the end. These trajectories illustrate the path the gripper should take to accomplish the subtask. Among the choices `<CHOICE>`, which subtask does the trajectory best correspond to? Please select one option only.
- You are performing `<TASK>`. The image provided displays gradient-colored trajectories, where green signifies the starting point and red marks the endpoint. These trajectories depict the path the gripper must follow to complete the subtask. From the options `<CHOICE>`, which subtask aligns best with the trajectory? Please select one option only.
- You are performing `<TASK>`. Below is an image showing gradient-colored trajectories, with green representing the start and red the end. The trajectory illustrates the path the gripper needs to take to finish the subtask. Which subtask from the choices `<CHOICE>` does the trajectory best represent? Please select one option only.

*Answer:* `<SUBTASK>`

**Planning Task**

- To advance toward {long-horizon}, what should be done next? Please answer in a short phrase.
- In order to fulfill {long-horizon}, what is the next best step? Please answer in a short phrase.
- Keeping {long-horizon} as the goal, what should be the next step to move forward? Please answer in a short phrase.
- To continue progressing toward {long-horizon}, what comes next? Please answer in a short phrase.
- Aiming for {long-horizon}, which step should you take now? Please answer in a short phrase.
- What should be the immediate next step to achieve {long-horizon}? Please answer in a short phrase.
- Considering {long-horizon} as the target, what task needs to be done next? Please answer in a short phrase.
- To move closer to {long-horizon}, what task should follow? Please answer in a short phrase.
- With {long-horizon} in mind, what is the most appropriate next action? Please answer in a short phrase.
- Toward achieving {long-horizon}, whats the next actionable task? Please answer in a short phrase.

*Answer:* `<task n>`

**Planning with Context Task**

- Having accomplished {past task 1 ∼ n-1}, what should be your next step to achieve {long-horizon}? Please answer in a short phrase.
- After completing {past task 1 ∼ n-1}, what action will best advance you toward {long-horizon}? Please answer in a short phrase.
- With {past task 1 ∼ n-1} done, what comes next to move closer to {long-horizon}? Please answer in a short phrase.
- Whats the next task to reach {long-horizon}, given that youve finished {past task 1 ∼ n-1}? Please answer in a short phrase.
- Taking into account {past task 1 ∼ n-1}, whats the best next move toward {long-horizon}? Please answer in a short phrase.
- With progress so far including {past task 1 ∼ n-1}, what should you do next to achieve {long-horizon}? Please answer in a short phrase.
- Whats the next step in pursuit of {long-horizon}, after completing steps {past task 1 ∼ n-1}? Please answer in a short phrase.
- So far, youve completed {past task 1 ∼ n-1}. Whats the next logical step to reach {long-horizon}? Please answer in a short phrase.
- Whats the next action towards {long-horizon}, considering the completed tasks ({past task 1 ∼ n-1})? Please answer in a short phrase.
- Given your progress ({past task 1 ∼ n-1}), whats the most logical next step for achieving {long-horizon}? Please answer in a short phrase.

*Answer:* `<task n>`

**Generative Affordance Task**

- Whats the most practical action you can start immediately? Please answer in a short phrase.
- Considering the current state, what task can you begin right away? Please answer in a short phrase.
- Whats something you can work on right now? Please answer in a short phrase.
- At this point, what action can you take immediately? Please answer in a short phrase.
- Considering whats possible now, what should you do first? Please answer in a short phrase.
- Whats the best task to start immediately? Please answer in a short phrase.
- What action is most feasible to take at this moment? Please answer in a short phrase.
- Whats the next task you can focus on right now? Please answer in a short phrase.
- Whats the most available task to handle at present? Please answer in a short phrase.
- Given the situation, whats the best action to take right now? Please answer in a short phrase.

*Answer:* `<task n>`

**Planning Remaining Steps Task**

- With {past task 1 ∼ n-1} completed, what are the next five actions to reach {long-horizon}? Please answer in sequential phrases.
- What five steps should you take next to achieve {long-horizon}, after accomplishing {past task 1 ∼ n-1}? Please answer in sequential phrases.
- Given that youve done {past task 1 ∼ n-1}, what are the next five things to do for {long-horizon}? Please answer in sequential phrases.
- Having completed {past task 1 ∼ n-1}, what five tasks will best advance you toward {long-horizon}? Please answer in sequential phrases.
- Considering the progress so far ({past task 1 ∼ n-1}), what five steps should follow to accomplish {long-horizon}? Please answer in sequential phrases.
- After finishing {past task 1 ∼ n-1}, what are the next five actions to take toward {long-horizon}? Please answer in sequential phrases.
- What are the next five tasks to complete after completing {past task 1 ∼ n-1} to achieve {long-horizon}? Please answer in sequential phrases.
- Considering {long-horizon}, what are the next five logical steps after {past task 1 ∼ n-1}? Please answer in sequential phrases.
- What are five actionable steps to follow {past task 1 ∼ n-1} in achieving {long-horizon}? Please answer in sequential phrases.
- To move toward {long-horizon}, what are the next five key actions after {past task 1 ∼ n-1}? Please answer in sequential phrases.

*Answer:* `<future task n ∼ n+m>`

**Future Prediction Task**

- Once {task n-1} is complete, what is expected to happen next? Please answer in a short phrase.
- After {task n-1}, what development is likely to take place? Please answer in a short phrase.
- What is most likely to occur after {task n-1} finishes? Please answer in a short phrase.
- Given the tasks so far, what might follow {task n-1}? Please answer in a short phrase.
- Following the completion of {task n-1}, what is anticipated to happen? Please answer in a short phrase.
- What outcome is expected to follow {task n-1}? Please answer in a short phrase.
- Whats the likely result after completing {task n-1}? Please answer in a short phrase.
- Whats predicted to happen after {task n-1}? Please answer in a short phrase.
- After {task n-1}, whats the next event you anticipate? Please answer in a short phrase.
- Following {task n-1}, what is predicted to occur? Please answer in a short phrase.

*Answer:* `<task n>`

**Past Description Task**

- What task was finished just now? Please answer in a short phrase.
- What did you complete most recently? Please answer in a short phrase.
- What was the last action you carried out? Please answer in a short phrase.
- Which task did you wrap up before this? Please answer in a short phrase.
- What was done immediately prior to this? Please answer in a short phrase.
- Which task was concluded most recently? Please answer in a short phrase.
- What was the previous action performed? Please answer in a short phrase.
- What task did you complete last? Please answer in a short phrase.
- What was the final task before now? Please answer in a short phrase.
- What action occurred just prior to this? Please answer in a short phrase.

*Answer:* `<task n>`

**Success Negative Task**

- Has {task n} been finished as intended? Please answer strictly with Yes or No.
- Is {task n} regarded as successfully completed? Please answer strictly with Yes or No.
- Is it confirmed that {task n} was done properly? Please answer strictly with Yes or No.
- Has {task n} been accomplished as expected? Please answer strictly with Yes or No.
- Can we confirm that {task n} was completed as required? Please answer strictly with Yes or No.
- Did {task n} achieve the desired results? Please answer strictly with Yes or No.
- Was {task n} executed successfully? Please answer strictly with Yes or No.
- Is {task n} considered a success? Please answer strictly with Yes or No.
- Can {task n} be marked as complete? Please answer strictly with Yes or No.
- Was {task n} carried out as planned? Please answer strictly with Yes or No.

*Answer:* `No`

**Success Positive Task**

- Has {task n} been finished as intended? Please answer strictly with Yes or No.
- Is {task n} regarded as successfully completed? Please answer strictly with Yes or No.
- Is it confirmed that {task n} was done properly? Please answer strictly with Yes or No.
- Has {task n} been accomplished as expected? Please answer strictly with Yes or No.
- Can we confirm that {task n} was completed as required? Please answer strictly with Yes or No.
- Did {task n} achieve the desired results? Please answer strictly with Yes or No.
- Was {task n} executed successfully? Please answer strictly with Yes or No.
- Is {task n} considered a success? Please answer strictly with Yes or No.
- Can {task n} be marked as complete? Please answer strictly with Yes or No.
- Was {task n} carried out as planned? Please answer strictly with Yes or No.

*Answer:* Yes

**Discriminative Affordance Positive Task**

- Is {task n} the task you should perform next? Please answer strictly with Yes or No.
- Are you expected to handle {task n} as the next step? Please answer strictly with Yes or No.
- Is {task n} the focus of your upcoming efforts? Please answer strictly with Yes or No.
- Is {task n} the next task to be addressed? Please answer strictly with Yes or No.
- Are you about to work on {task n} next? Please answer strictly with Yes or No.
- Is {task n} possible to start next? Please answer strictly with Yes or No.
- Can {task n} be executed in the next step? Please answer strictly with Yes or No.
- Is it realistic to begin {task n} in the following step? Please answer strictly with Yes or No.
- Is {task n} achievable as the next stage? Please answer strictly with Yes or No.
- Can you take up {task n} immediately after the current task? Please answer strictly with Yes or No.

*Answer:* Yes

**Discriminative Affordance Negative Task**

- Is {random task} the task you should perform next? Please answer strictly with Yes or No.
- Are you expected to handle {random task} as the next step? Please answer strictly with Yes or No.
- Is {random task} the focus of your upcoming efforts? Please answer strictly with Yes or No.
- Is {random task} the next task to be addressed? Please answer strictly with Yes or No.
- Are you about to work on {random task} next? Please answer strictly with Yes or No.
- Is {random task} possible to start next? Please answer strictly with Yes or No.
- Can {random task} be executed in the next step? Please answer strictly with Yes or No.
- Is it realistic to begin {random task} in the following step? Please answer strictly with Yes or No.
- Is {random task} achievable as the next stage? Please answer strictly with Yes or No.
- Can you take up {random task} immediately after the current task? Please answer strictly with Yes or No.

*Answer:* No

**Past Multi task Selection**

- For {long-horizon}, which of the following subtasks has already been completed?
  Choices: [A. `<task 1>`, B. `<task 2>`, C. `<task 3>`, D. `<task 4>`].
- In the context of {long-horizon}, which subtask has been finished so far?
  Choices: [A. `<task 1>`, B. `<task 2>`, C. `<task 3>`, D. `<task 4>`].
- Regarding {long-horizon}, which subtask among the following has already been accomplished?
  Choices: [A. `<task 1>`, B. `<task 2>`, C. `<task 3>`, D. `<task 4>`].
- To achieve {long-horizon}, which subtask has just been completed?
  Choices: [A. `<task 1>`, B. `<task 2>`, C. `<task 3>`, D. `<task 4>`].
- Which of the listed subtasks has already been executed for {long-horizon}?
  Choices: [A. `<task 1>`, B. `<task 2>`, C. `<task 3>`, D. `<task 4>`].

*Answer:* A/B/C/D

**Future Multi task Selection**

- For achieving {long-horizon}, which subtask is the most appropriate to execute next?
  Choices: [A. `<task 1>`, B. `<task 2>`, C. `<task 3>`, D. `<task 4>`].
- In order to complete {long-horizon}, which subtask should be executed next?
  Choices: [A. `<task 1>`, B. `<task 2>`, C. `<task 3>`, D. `<task 4>`].
- Which subtask should be the next step toward {long-horizon}?
  Choices: [A. `<task 1>`, B. `<task 2>`, C. `<task 3>`, D. `<task 4>`].
- Looking ahead to {long-horizon}, which subtask should be performed next?
  Choices: [A. `<task 1>`, B. `<task 2>`, C. `<task 3>`, D. `<task 4>`].
- For progressing toward {long-horizon}, which subtask is the most suitable next step?
  Choices: [A. `<task 1>`, B. `<task 2>`, C. `<task 3>`, D. `<task 4>`].

*Answer:* A/B/C/D

---

**Scene Understanding**

- Given four images labeled A, B, C, D, which image best corresponds to the scene required for accomplishing {long-horizon}? Please answer with a single option (A/B/C/D).
- Which of the images (A, B, C, D) best matches the scene necessary to achieve {long-horizon}? Please answer with a single option (A/B/C/D).
- Looking at the four images A, B, C, D, which one best represents the scene for {long-horizon}? Please answer with a single option (A/B/C/D).
- Among the four images (A, B, C, D) provided, which scene corresponds to {long-horizon}? Please answer with a single option (A/B/C/D).
- Which image (A, B, C, D) most accurately depicts the scene for {long-horizon}? Please answer with a single option (A/B/C/D).

*Answer:* `A/B/C/D`

---

**Temporal Understanding**

- Given four images labeled A, B, C, and D, which image corresponds to the state while executing {task n}? Please answer with a single option (A/B/C/D).
- Which of the four images (A, B, C, D) shows the state during the execution of {task n}? Please answer with a single option (A/B/C/D).
- Looking at the four images, which one depicts the state when {task n} is being carried out? Please answer with a single option (A/B/C/D).
- Among images A, B, C, and D, which represents the state while performing {task n}? Please answer with a single option (A/B/C/D).
- Which image (A, B, C, D) corresponds to the state during the execution of {task n}? Please answer with a single option (A/B/C/D).

*Answer:* `A/B/C/D`

---

**Past Primitive Selection**

- Which primitive skill has just been applied to achieve {long-horizon}?
  Choices: [A. `<skill 1>`, B. `<skill 2>`, C. `<skill 3>`, D. `<skill 4>`].
- In pursuit of {long-horizon}, which primitive skill was applied most recently?
  Choices: [A. `<skill 1>`, B. `<skill 2>`, C. `<skill 3>`, D. `<skill 4>`].
- Which skill was just executed for the purpose of {long-horizon}?
  Choices: [A. `<skill 1>`, B. `<skill 2>`, C. `<skill 3>`, D. `<skill 4>`].
- For achieving {long-horizon}, which primitive was applied last?
  Choices: [A. `<skill 1>`, B. `<skill 2>`, C. `<skill 3>`, D. `<skill 4>`].
- Which primitive skill had just been used in the process of {long-horizon}?
  Choices: [A. `<skill 1>`, B. `<skill 2>`, C. `<skill 3>`, D. `<skill 4>`].

*Answer:* `A/B/C/D`

---

**Future Primitive Selection**

- Which primitive skill should be applied next to achieve {long-horizon}?
  Choices: [A. `<skill 1>`, B. `<skill 2>`, C. `<skill 3>`, D. `<skill 4>`].
- For {long-horizon}, which primitive skill is the most suitable to apply next?
  Choices: [A. `<skill 1>`, B. `<skill 2>`, C. `<skill 3>`, D. `<skill 4>`].
- In order to accomplish {long-horizon}, which skill should be applied next?
  Choices: [A. `<skill 1>`, B. `<skill 2>`, C. `<skill 3>`, D. `<skill 4>`].
- Looking ahead to {long-horizon}, which primitive should be chosen next?
  Choices: [A. `<skill 1>`, B. `<skill 2>`, C. `<skill 3>`, D. `<skill 4>`].
- Which of the following skills should be applied as the next step for {long-horizon}?
  Choices: [A. `<skill 1>`, B. `<skill 2>`, C. `<skill 3>`, D. `<skill 4>`].

*Answer:* `A/B/C/D`

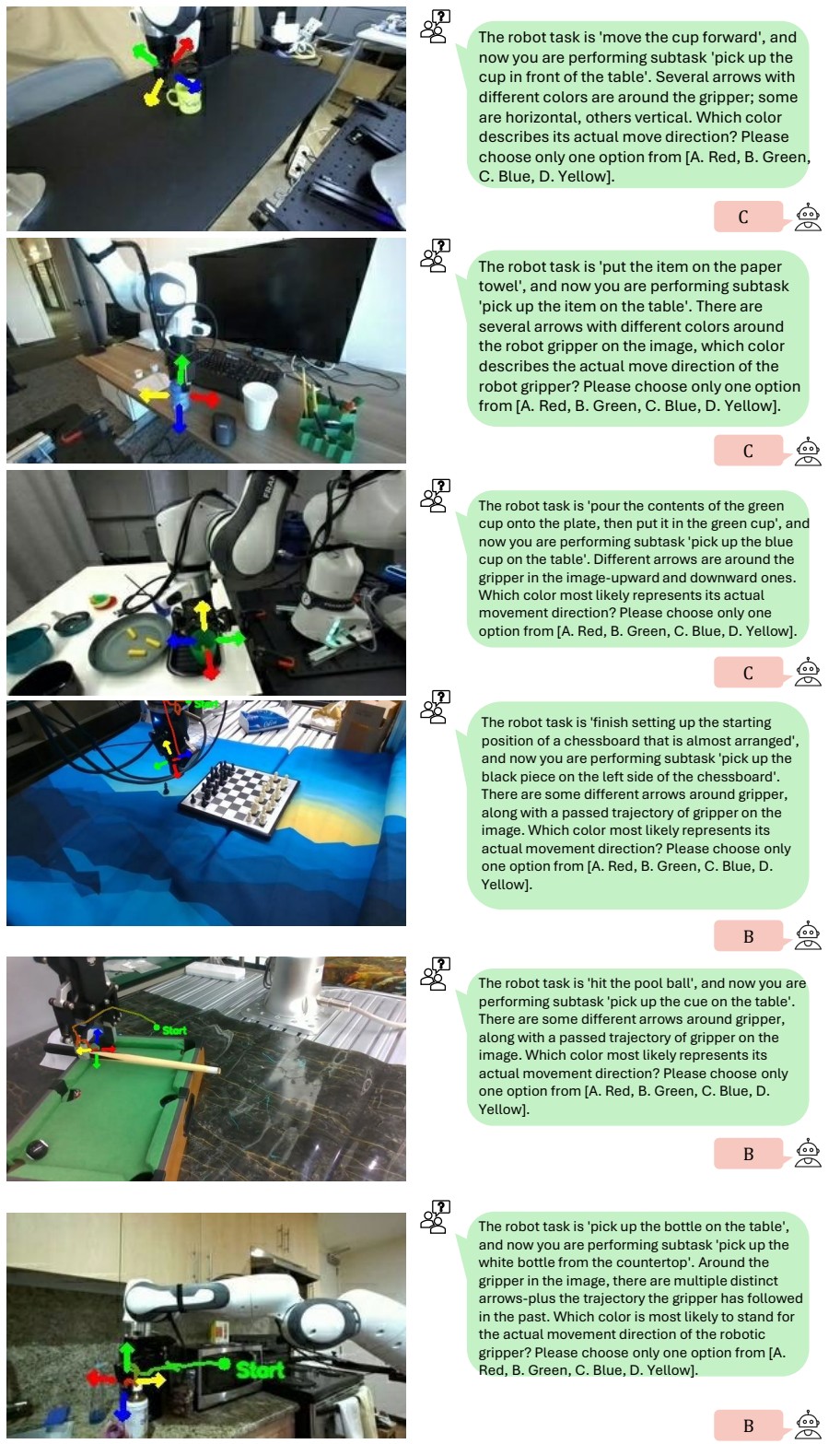

Figure 29: Trajectory direction choice task and trajectory direction choice with trajectory task.

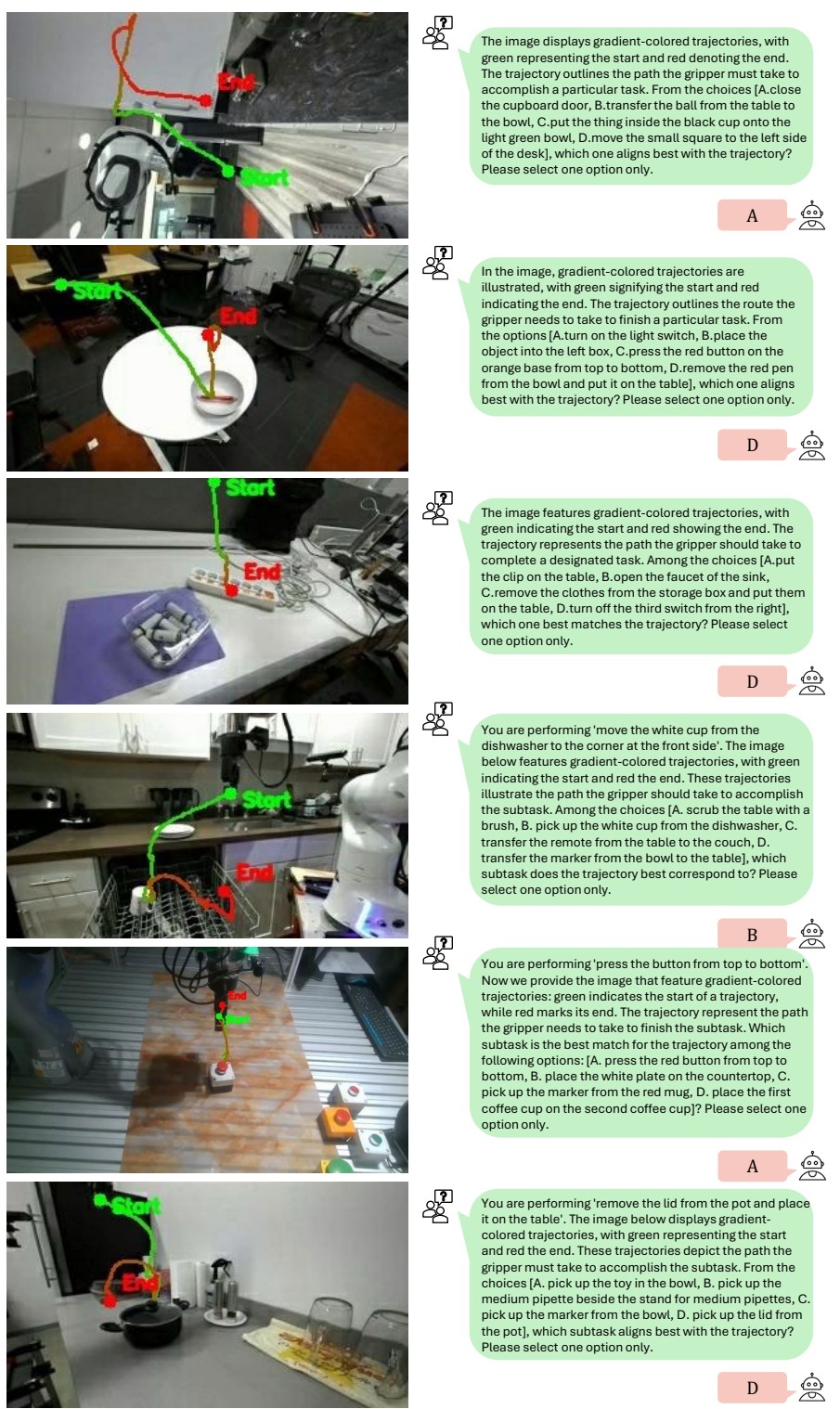

Figure 30: Trajectory sub-goal choice Task

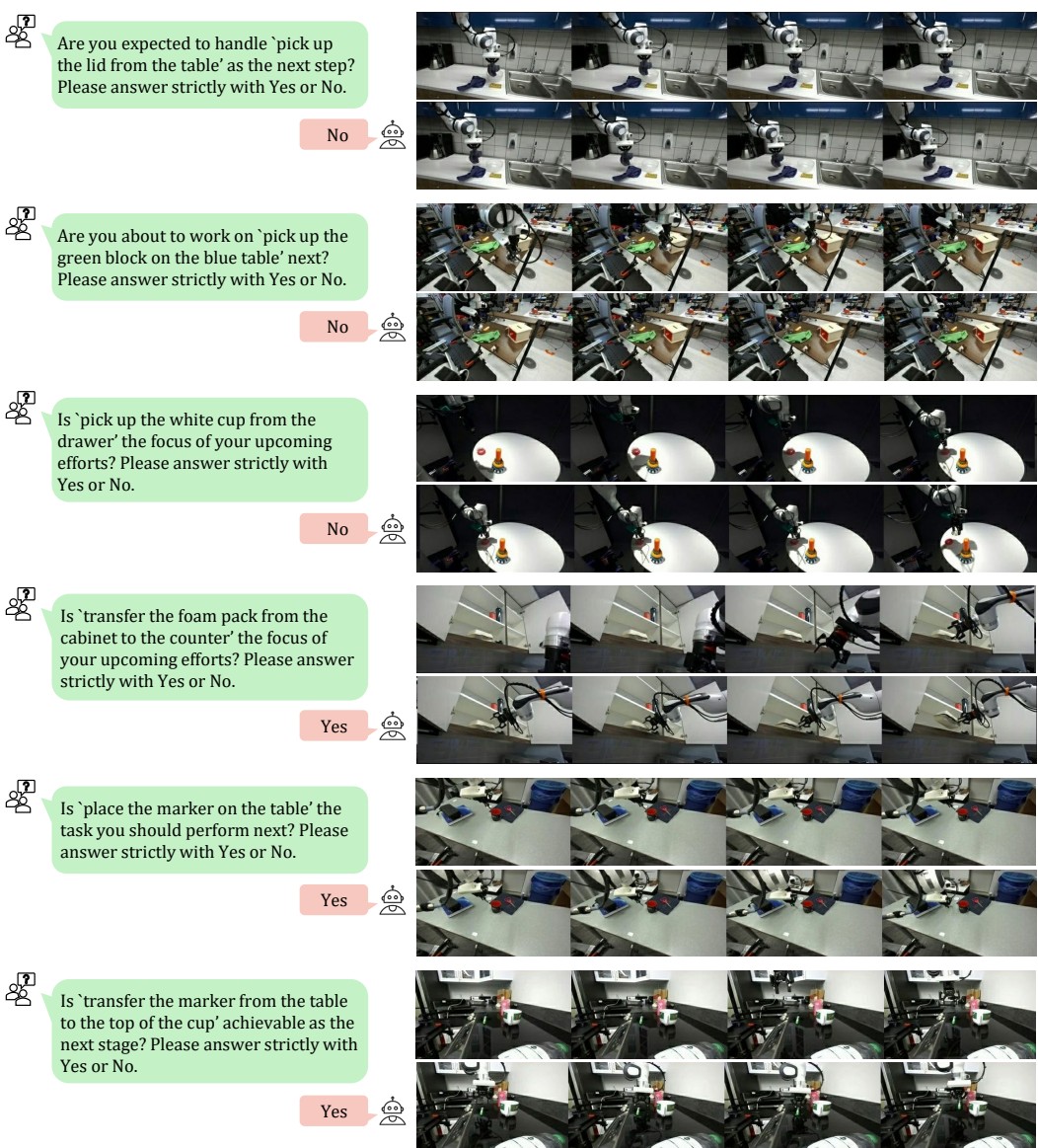

Figure 31: Discriminative tasks of task planning.

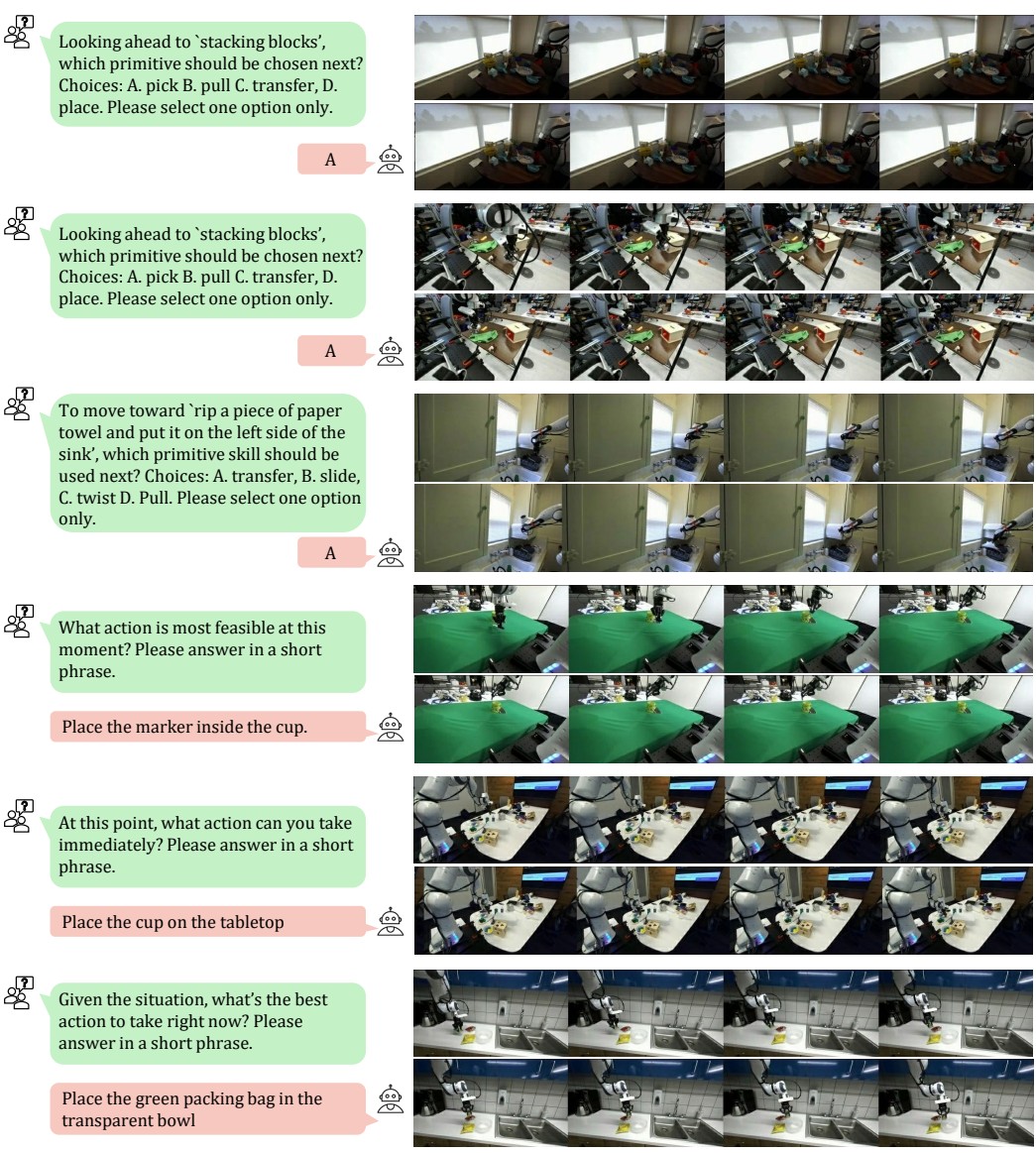

Figure 32: Future primitive selection task and generative affordance prediction task.

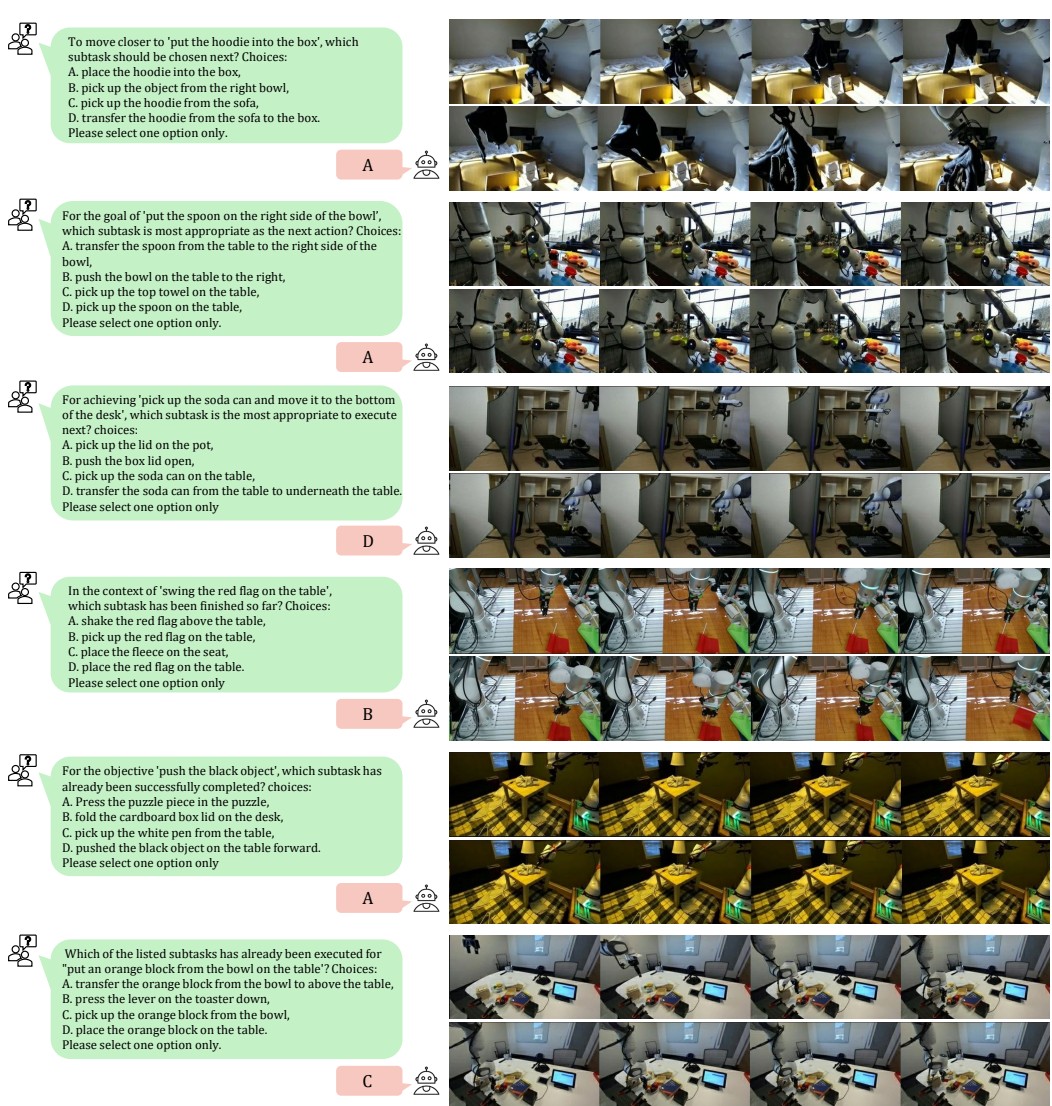

Figure 33: Future multi-task selection task and past multi-task selection task.

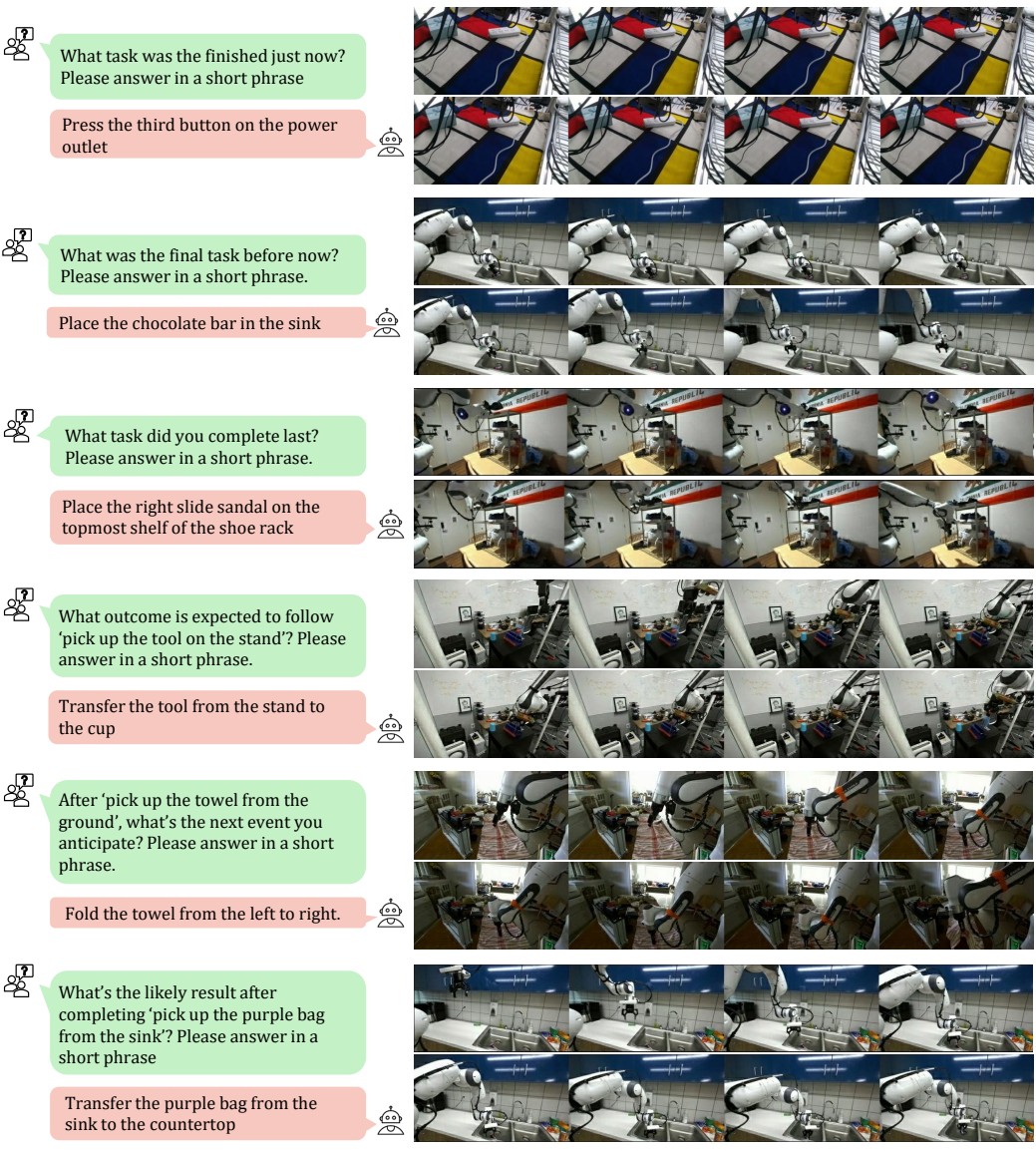

Figure 34: Past description task and future prediction task.

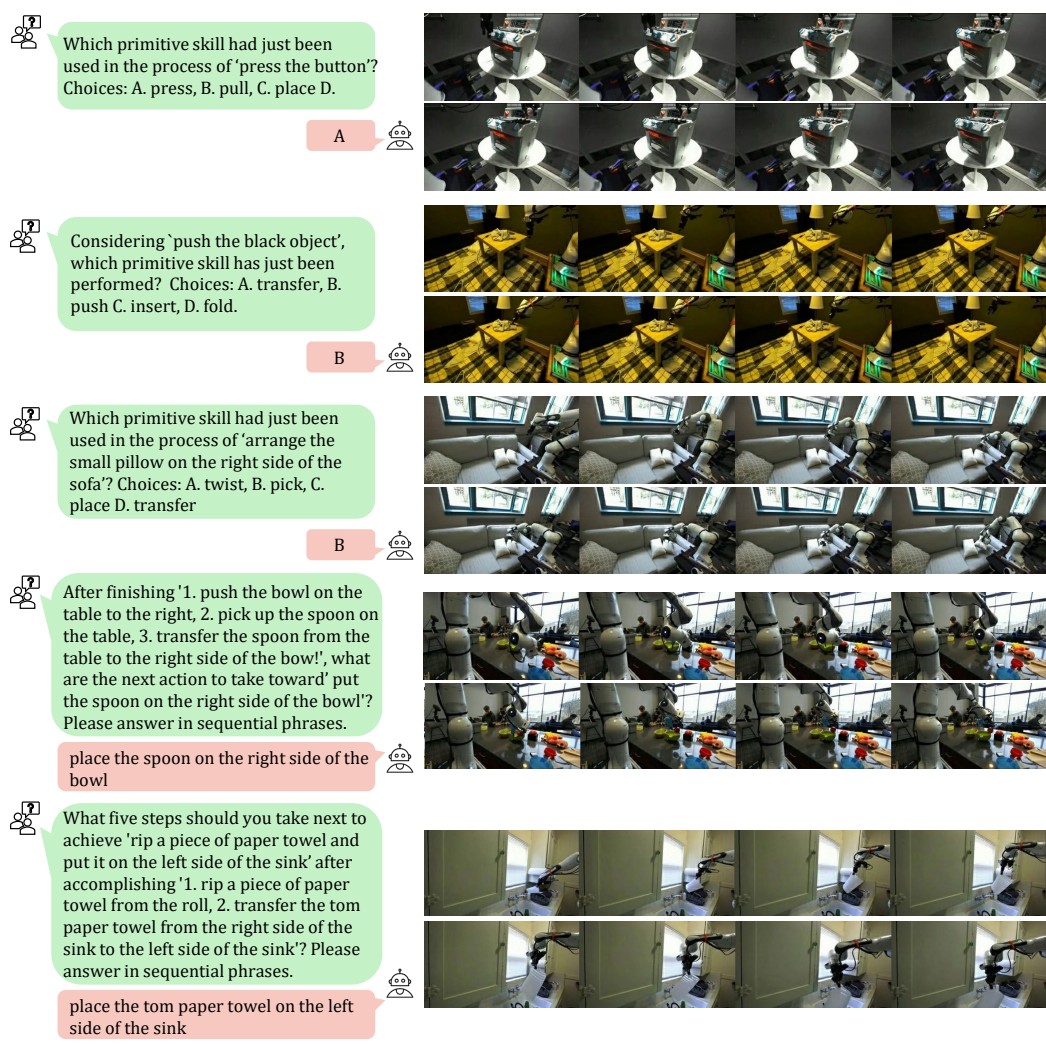

Figure 35: Past primitive skill selection task and planning remaining steps task.

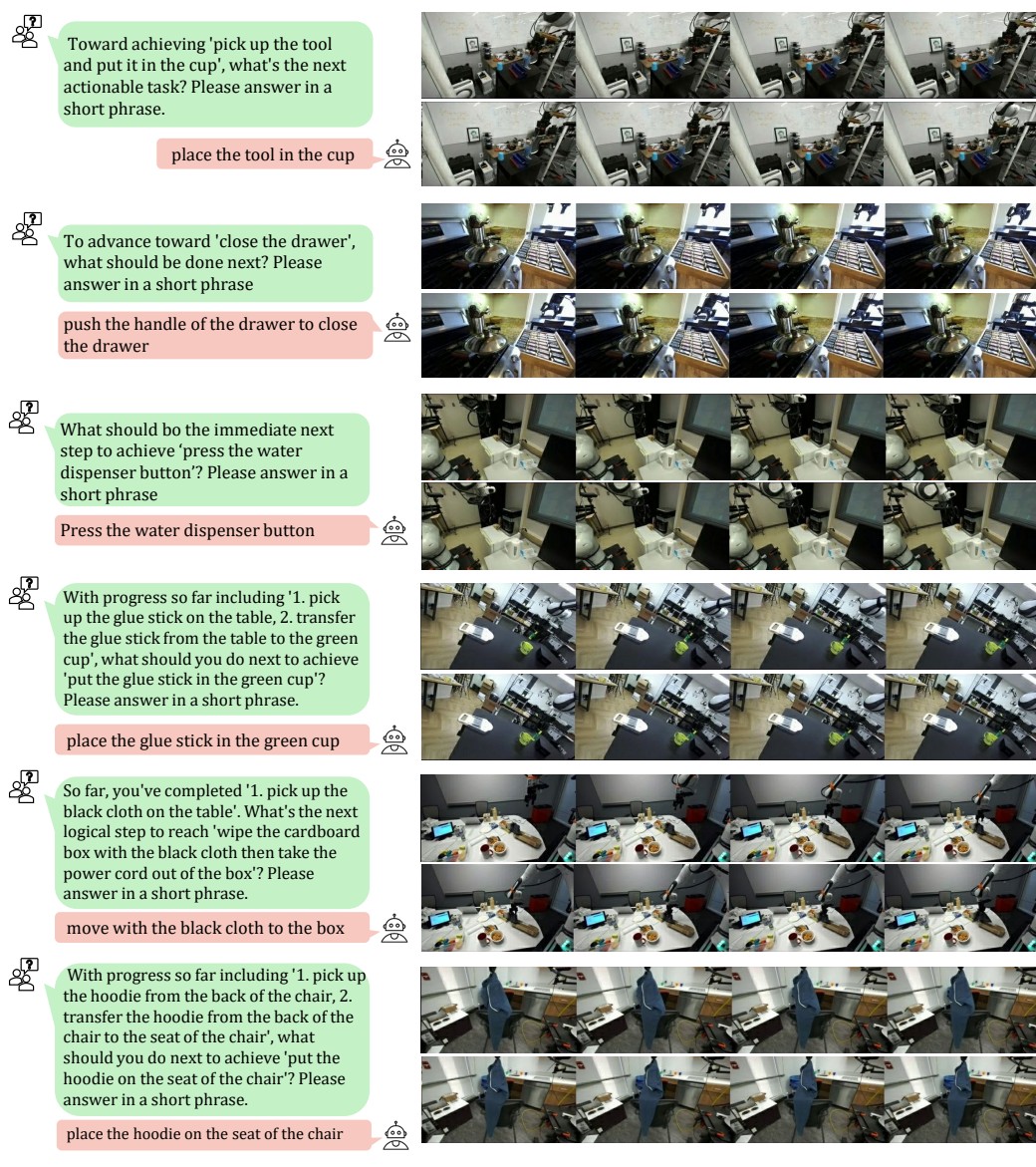

Figure 36: Planning task and planning with context task.

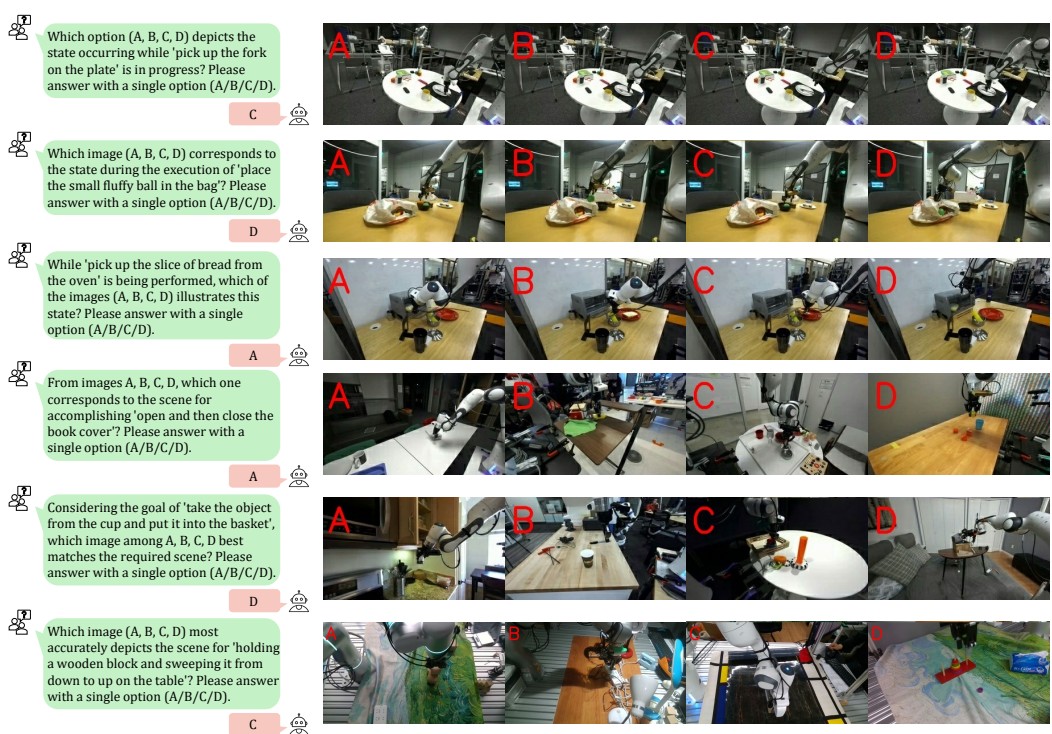

Figure 37: Scene understanding task and temporal understanding task.

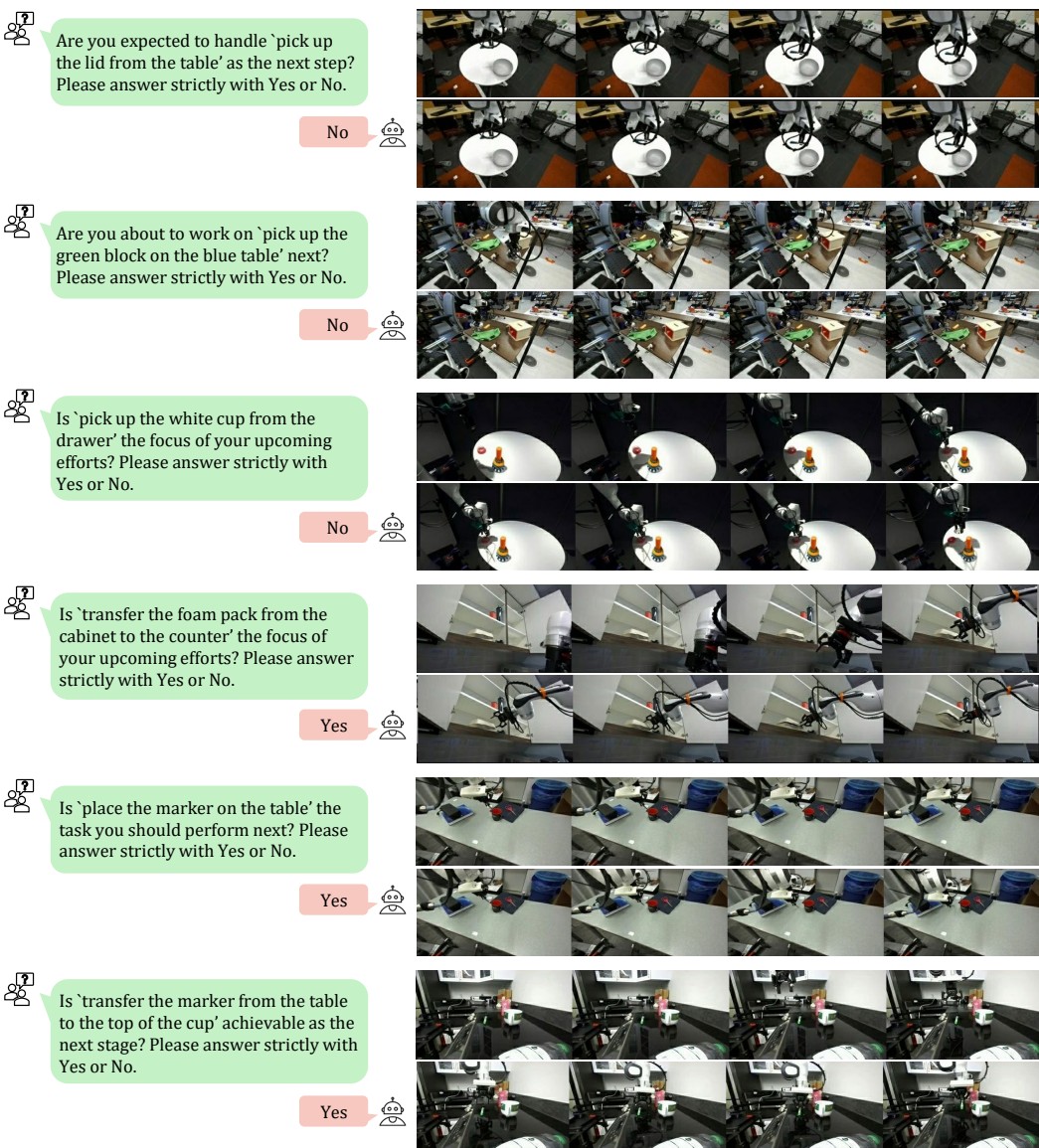

Figure 38: Discriminative tasks of task planning.

### A.7.4 COT DATA FOR ROBOINTER-VLA

**Flexible Chain-of-Thought Data.** For training the *Explicitly-Conditioned E2E* and *Modular Planner-to-Executor* settings, we construct *Flexible Chain-of-Thought* (F-CoT) data. A representative visualization is provided below; in practice, we serialize F-CoT as textual sequences as shown in Figure 39 and 40.

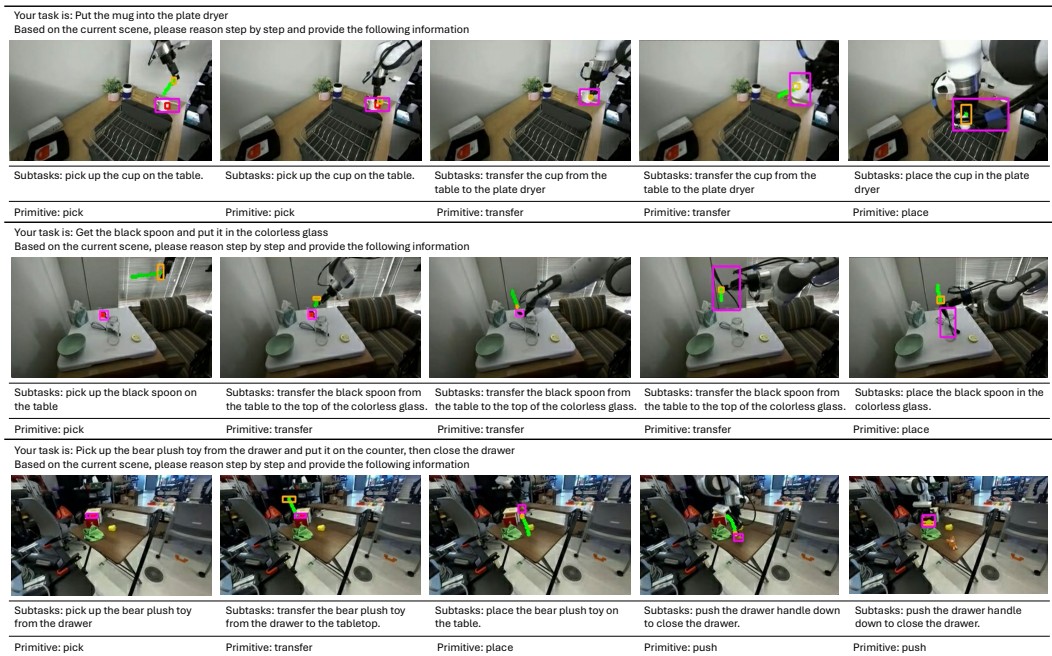

Figure 39: Visualization of the *F-CoT*.

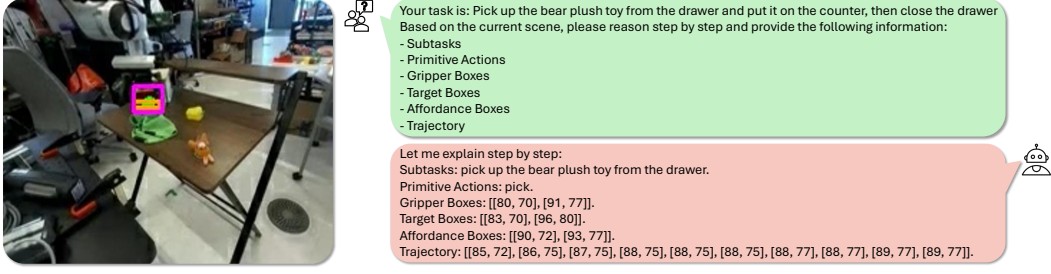

Figure 40: Visualization of the textual chain-of-though reasoning.

## A.8 FURTHER APPLICATION, LIMITATIONS AND LLM USAGE STATEMENT

**Further applications.** Beyond training VLMs and VLAs, *RoboInter-Data* also supports broader research directions: **(1) Expert generative models for individual intermediate representations.** In contrast to datasets centered on a single annotation type, RoboInter-Data spans diverse embodied scenes while providing large-scale, high-quality annotation for each intermediate representation, enabling the pre-training of specialized generative models tailored to specific representations. **(2) Human–robot interaction.** Intermediate representations such as traces, bounding boxes, and sub-task sequences naturally bridge high-level instructions and low-level control. Our dataset supports the development of intuitive and precise shared-autonomy systems grounded in these representations. **(3) Embodied world model learning.** All annotations in RoboInter-Data are temporally aligned with $640 \times 360$ raw manipulation videos. These intermediate representations can serve as structured control signals for controllable video generation, while the aligned information about the robot, objects, and environment provides strong supervision for learning structured embodied world models. **(4) Video action model learning.** The combination of action annotations, videos, and diverse intermediate representations makes RoboInter-Data well suited for training video action models with rich, multi-level supervision.

**Limitations.** While the RoboInter suite offers broad coverage across tasks and scenes, current annotations are primarily collected on fixed-base, single-arm robots, limiting embodiment diversity. Future extensions may incorporate more complex setups, such as mobile or dual-arm systems, to enhance cross-embodiment generalization. Additionally, while our planner exhibits strong reasoning capabilities, improving its robustness and generalization across unseen scenarios remains an open challenge. The executor, though effective, relies on relatively heavy inference due to long-chain reasoning; future work may explore lightweight architectures and optimization strategies to improve deployment efficiency.

**LLM Usage Statement.** We employed large language models (LLMs) solely for grammar refinement and minor linguistic polishing. All LLM-assisted edits were carefully reviewed and verified by the authors to ensure that no fabricated content or unintended alterations to the original meaning were introduced. The research ideas, experimental design, data analysis, and conclusions presented in this work were entirely conceived and executed by the authors without LLM assistance.

