# OpenReview forum: "RoboInter: A Holistic Intermediate Representation Suite Towards Robotic Manipulation"
_ICLR.cc/2026/Conference — ICLR 2026 Poster_

### Official Review · Reviewer_hDxo · 2025-10-27

**Soundness:** 4
**Presentation:** 4
**Contribution:** 3
**Rating:** 8
**Confidence:** 4

**Summary:**

This paper focuses on the plan-then-execute paradigm for robotic manipulation and tries to fill gap of lacking fine-grained intermediate supervision with the proposed RoboInter Manipulation Suite. RoboInter Manipulation Suite contributes in three aspects: 1) RoboInter-Data, a human-veriffed dataset with over 200k episodes across 571 diverse scenes, 2) RoboInter-VQA, 8 spatial and 20 temporal embodied QA categories to benchmark and enhance the embodied capabilities of current large vision-language models, and 3) RoboInter-VLA, a ffexible plan-then-execute framework with modular and end-to-end variants that link planning to execution. Experiments indicate that RoboInter-Data significantly improves the model's reasoning capabilities for embodied tasks. In closed-loop evaluations, RoboInter-VLA is shown to effectively complete both in-distribution and out-of-distribution tasks when compared to Pi0 and OpenVLA.

**Strengths:**

1. The dataset makes a significant contribution to the plan-then-execute paradigm and can also be utilized in other research paradigms.

2. Comprehensive experiments validate the contributions of the dataset effectively.

3. The insightful analysis of the variants of RoboInter-VLA provides valuable inspiration for other researchers in this field.

**Weaknesses:**

1. There is a lack of evaluation of other plan-then-execute methods trained on the proposed dataset, which would provide a more comprehensive validation of the dataset's contributions.

2. There is insufficient analysis regarding which types of intermediate representations (subtask, primitive, target box, affordance) are most beneficial for robotic manipulation tasks. Are all the mentioned representations utilized as inputs to RoboInter-VLA during both open-loop and closed-loop evaluations?

3. An analysis of inference time for RoboInter-VLA is missing.

**Questions:**

1. The RoboInter-Modular variant demonstrates superior performance in open-loop evaluations. However, it is not included in the closed-loop evaluation. What are the reasons for this exclusion?

---

> ### Author Response · Authors · 2025-11-23
> **Rebuttal by Authors (1/3)**
>
> We thank the reviewer for the summary and for highlighting the strengths of our work, including: (1) a large-scale, human-verified dataset with fine-grained intermediate supervision for manipulation, (2) RoboInter-VQA to benchmark embodied reasoning, (3) flexible modular and end-to-end VLA variants that link planning and execution, (4) demonstrating strong improvements in reasoning and manipulation performance.
>
> Below, we address the concerns in detail.
>
> ---
>
> **`1. Comparison with more baselines`**
> > W-1: There is a lack of evaluation of other plan-then-execute methods trained on the proposed dataset, which would provide a more comprehensive validation of the dataset's contributions.
>
> Thanks for the suggestion. We additionally evaluated **VLA-OS [1]** (revised in `Table.4` and `Section 5.2` of main paper), **a recent plan-then-execute baseline that supporting both an action-only variant and an explicit reasoning variant**.
>
> For each model family, we compare two settings: **without CoT (Vanilla) and with CoT (Oracle+Executor)**, corresponding to **using vs. not using our intermediate representation data**.
>
> | Model            | Backbone Size | OLS@0.1 | OLS@0.05 | OLS@0.03 | OLS@0.01 | Mean   |
> |------------------|---------------|---------|----------|----------|----------|--------|
> | VLA-OS (Vanilla)  | 3B            | 0.6180  | 0.3905   | 0.1928   | 0.0129   | 0.3035 |
> | Ours (Vanilla)   | 3B            | 0.6793  | 0.3608   | 0.1753   | 0.0189   | **0.3086** |
> | VLA-OS (+CoT)     | 3B            | 0.7260  | 0.4928   | 0.2734   | 0.0200   | 0.3780 |
> | Ours (+CoT)      | 3B            | 0.7511  | 0.4640   | 0.2705   | 0.0587   | **0.3861** |
>
> Training **VLA-OS on our dataset with CoT supervision yields an improvement of approximately +23%** over its Vanilla variant. This substantial gain provides **strong evidence for the effectiveness and quality of the intermediate representations from our dataset**. And our models achieve higher performance than VLA-OS across both Vanilla and CoT settings.
>
> Overall, these results confirm that **the proposed intermediate representation dataset** brings **consistent benefits across different plan-then-execute architectures**, reinforcing the general utility of our data.
>
> [1] Gao C, Liu Z, et al. VLA-OS: Structuring and Dissecting Planning Representations and Paradigms in Vision-Language-Action Models.
>
> ---
>
> **`2. Analysis of intermediate representations types`**
> > W-2: There is insufficient analysis regarding which types of intermediate representations (subtask, primitive, target box, affordance) are most beneficial for robotic manipulation tasks. Are all the mentioned representations utilized as inputs to RoboInter-VLA during both open-loop and closed-loop evaluations?
>
> **(1) Analysis regarding different types of intermediate representations**
>
> Our initial submission has reported some ablations in `Appendix Tables 15`. Below we **provide more results and analysis about the contributions of each intermediate representation**, these content has been revised in `Appendix A.5.3`:
>
> | Method        | OLS@0.1 | OLS@0.05 | OLS@0.03 | OLS@0.01 | Mean |
> |----------------|---------|----------|----------|----------|----------|
> | Vanilla         | 0.6793 | 0.3608 | 0.1753 | 0.0189 | 0.3086 |
> | + Subtask       | 0.6965 | 0.3676 | 0.1770 | 0.0171 | 0.3146 |
> | + Primitive Skill    | 0.6925 | 0.3658 | 0.1793 | 0.0156 | 0.3133 |
> | + Target Box    | 0.7001 | 0.3793 | 0.1879 | 0.0202 | 0.3218 |
> | + Affordance    | 0.7054 | 0.3958 | 0.2012 | 0.0263 | **0.3322** |
>
> The trend is clear: **coarser intermediate cues** (subtask, primitive skill) lead to **modest improvements**, while **finer-grained spatial cues** (target box and especially affordance) provide **substantially larger gains**. Notably, **Affordance is the most beneficial among these types**. Overall, **more fine-grained intermediate supervision lead to stronger performance.**
>
> **(2) Usage of representations**
>
> In **open-loop analysis**, our main goal is to **study how intermediate representations influence VLA learning**, so we evaluate **a broad set of almost all representative types**.
>
> For **real-world closed-loop control**, however, deploying the full set of representations is unnecessary and inefficient. **Covering too many representations will increase the latency and provide redundant information**. Consequently, as detailed in Appendix A.4.1, we adopt **a more compact CoT consisting of three key representations (subtask, gripper box, and affordance)**, which **preserves essential guidance while avoiding unnecessary inference overhead**.
>
> **More broadly**: we view **the role of different intermediate representations** in different robotic tasks **as an open and important research question**. One motivation for releasing this large-scale RoboInter-Data with diverse representations is also to **enable the community to investigate the utilization of different intermediate representation more systematically**  in future work.

---

> ### Author Response · Authors · 2025-11-23
> **Rebuttal by Authors (2/3)**
>
> **`3. An analysis of inference time for RoboInter-VLA`**
> > W-3: An analysis of inference time for RoboInter-VLA is missing.
>
> Thanks for raising this point. Follow the suggestion, we provide **a detailed analysis regarding the inference time for RoboInter-VLA below** (revised in `Appendix A.5.1`):
>
> As **our primary objective is to study how our large-scale intermediate representation dataset influence VLA learning**; thus, our **open-loop evaluations did not emphasize speed**. However, for **closed-loop deployment**, we applied **standard acceleration strategies** to ensure fair and practical comparisons.
>
> **(1) Inference time comparison on open-loop experiments**
>
> We compare RoboInter-VLA against **VLA-OS [1]**, a recent model that **also includes a vanilla action-only variant and an explicit reasoning variant (VLA-OS-I-Explicit)**. An detailed latency comparison:
>
>
> | Model               | Model Size | Reasoning Tokens | Latency    | Open-loop Score |
> |---------------------|------------|------------------|------------|------------------|
> | VLA-OS (Vanilla)     | 3B         | 0                | 0.1647s    | 0.3035           |
> | VLA-OS-I-Explicit    | 3B         | 64               | 2.2274s    | 0.3290           |
> | Vanilla             | 3B         | 0                | 0.1309s    | 0.3086           |
> | RoboInter-IC-E2E    | 3B         | 64               | 0.1309s    | 0.3218           |
> | RoboInter-EC-E2E    | 3B         | 64               | 2.2187s    | 0.3340           |
> | RoboInter-Modular   | 3B+3B      | 64               | 2.3594s    | 0.3543           |
>
> Across both frameworks (VLA-OS and RoboInter-VLA), we observe consistent trends:
> - **Explicit reasoning substantially improves open-loop performance**, VLA-OS (0.3035 → 0.3290) and RoboInter-VLA (0.3086 → 0.3340).
> - **RoboInter-VLA is faster than VLA-OS in both w/o reasoning or w/ reasoning settings**, RoboInter-IC-E2E v.s. VLA-OS (0.1309s v.s. 0.1647s), RoboInter-EC-E2E v.s. VLA-OS-I-Explicit (2.2187s v.s. 2.2274s).
> - **Performance improvement comes with increased latency**, primarily caused by autoregressive generation of CoT tokens. It also **indicates this phenomenon shares by most planner-to-executor architectures**.
>
> **(2) Real-world comparison with acceleration strategies**
>
> For real-world deployment, we further **implemented practical acceleration strategies: textual caching and chunked action execution for EC-E2E** (as described in `Appendix A.4.1`), and **asynchronous dual-frequency execution for the Modular variant**. The measured speeds are:
>
> | Model                            | Speed     | Latency   | ID Score | OOD Score |
> |----------------------------------|-----------|-----------|----------|-----------|
> | OpenVLA                          | ~3.8 Hz   | 263.2 ms  | 45.0     | 23.3      |
> | Vanilla                          | ~7.64 Hz  | 130.9 ms  | 65.0     | 38.3      |
> | RoboInter-IC-E2E                 | ~7.64 Hz  | 130.9 ms  | 76.7     | 58.3      |
> | RoboInter-EC-E2E (cache+chunk)   | ~2.56 Hz  | 390.6 ms  | 68.3     | 60.0      |
> | RoboInter-Modular (asynchronous) | ~6.92 Hz  | 144.5 ms  | 73.3     | 60.0      |
>
> In practical deployment, **all of our models (IC-E2E, EC-E2E, and Modular) maintain a reasonable execution speed** while consistently **delivering superior performance on both in-domain (ID) and out-of-domain (OOD) tasks**.
>
> **(3) Overall analysis**
>
> **Explicit intermediate reasoning yields stronger performance and generalization at the cost of additional latency in open-loop testing. In real-world deployment, practical optimization strategies used in RoboInter-VLA can mitigate these latency issues and allows the models to maintain strong performance.**
>
> [1] Gao C, Liu Z, et al. VLA-OS: Structuring and Dissecting Planning Representations and Paradigms in Vision-Language-Action Models.

---

> ### Author Response · Authors · 2025-11-23
> **Rebuttal by Authors (3/3)**
>
> **`4. Additional experiments with RoboInter-Modular`**
> > Q-1: The RoboInter-Modular variant demonstrates superior performance in open-loop evaluations. However, it is not included in the closed-loop evaluation. What are the reasons for this exclusion?
>
> Thanks for the question. In the open-loop setting, we mainly investigate **how the proposed intermediate representation dataset affect different architectural variants**, so we **included the RoboInter-Modular model for completeness**.
>
> However, for **closed-loop real-world deployment**, we **focus on comparing against *fully end-to-end* baselines such as OpenVLA and π0**. Therefore, **rather than the non-end-to-end Modular architecture**, which includes **an additional planning VLM and doubles the parameter count (3B v.s. 6B)**, we mainly utilize the **end-to-end IC-E2E and EC-E2E models** to ensure **a more direct and fairer comparison**.
>
> Following the suggestion, we conducted **an additional comparison using RoboInter-Modular**, and real-world results are shown below:
>
> | Model      | Objects Collecting (ID/OOD) ↑ | Cups Stacking (ID/OOD) ↑ | Towels Folding (ID/OOD) ↑ | Clutters Cleaning (ID/OOD) ↑ | ID→OOD Drop ↓ |
> |------------|------------------------------|--------------------------|---------------------------|------------------------------|----------------|
> | OpenVLA    | 53.3 / 20.0                  | 66.7 / 33.3              | 26.7 / 6.7                | 33.3 / 33.3                  | 21.7           |
> | π0    | 73.3 / 46.7                  | 80.0 / 53.3              | 53.3 / 40.0                | 46.7 / 40.0                  | 18.3           |
> | Vanilla    | 66.7 / 33.3                  | 80.0 / 46.7              | 46.7 / 20.0               | 66.7 / 53.3                  | 26.7           |
> | IC-E2E     | 86.7 / 53.3                  | 86.7 / 60.0              | 60.0 / 46.7               | 73.3 / 73.3                  | 18.4           |
> | EC-E2E     | 73.3 / 60.0                  | 80.0 / 73.3              | 46.7 / 40.0               | 73.3 / 66.7                  | 8.3            |
> | **Modular**    | 66.7 / 53.3                  | 86.7 / 73.3          | 53.3 / 40.0               | 80.0 / 73.3                  | 11.7           |
>
> The Modular variant achieves **strong real-world performance** and **competitive OOD generalization**. Its slightly larger ID→OOD drop relative to EC-E2E may be caused by the asynchronous two-module setup, while powerful, is somewhat more sensitive to distribution shifts. Nevertheless, these results indicate that **RoboInter-Modular achieves superior performance and significantly benefits from our intermediate representation dataset**.

---

> > ### Comment · Reviewer_hDxo · 2025-11-28
> >
> > Thanks for the response. It has resolved my concerns. I will maintain my rating.

---

> ### Author Response · Authors · 2025-11-28
> **Response to Reviewer hDxo**
>
> Dear Reviewer hDxo,
>
> We’re delighted that our response has addressed your concerns, and we truly appreciate that you maintain your positive rating 8. If you have any further questions or suggestions, please feel free to let us know.
>
> Sincerely,
> The Authors

---

### Official Review · Reviewer_v8Xb · 2025-10-30

**Soundness:** 3
**Presentation:** 3
**Contribution:** 3
**Rating:** 8
**Confidence:** 4

**Summary:**

This paper presents RoboInter Manipulation Suite, a unified resource of data, benchmarking, and training vision-language-action (VLA) models. A key contribution of this paper is to propose a new dataset for improving embodied reasoning capabilities of VLA models. The dataset consists of fine-grained annotations of 200k video data, including language, key frame, target object, gripper coordinate annotations. The annotations were labeled in the RoboInter-Tool. Based on this data, the authors designed the visual question answering (VQA) tasks that require vision-language models to answer spatial or temporal questions. Finally, this paper also proposes a VLA model trained on the collected data. Experiments show that RoboInter-VLA improves reasoning capabilities on both third-party benchmarks (e.g., RoboVQA) and the proposed RoboInter-VQA tasks. RoboInter-VLA was also deployed to the real-world environment, and it showed improved performance, especially in out-of-distribution robot manipulation tasks.

**Strengths:**

- [S1] The paper proposes a new data, a benchmark, and a VLA model for embodied reasoning, which could be a great resource for the community.
- [S2] Experiments validate the effectiveness of the collected data on both other benchmarks and the proposed benchmark.
- [S3] The paper is well-written and easy to understand.

**Weaknesses:**

- [W1] While a plan-then-execute framework improves reasoning and interpretability by generating intermediate representations, it inherently hurts the latency of robotic systems. The paper does not provide any analysis regarding the inference speed of RoboInter-VLA compared with other VLA models. This could improve the understanding of the trade-off between reasoning capabilities and real-time performance in robotic systems.

**Questions:**

- [Q1] Will the RoboInter-VQA dataset be made publicly available?

---

> ### Author Response · Authors · 2025-11-23
> **Rebuttal by Authors**
>
> We thank the reviewer for the constructive summary and for highlighting the strengths of our work, including: (1) establishing a unified suite with data, benchmark, and training, (2) a large-scale dataset with fine-grained intermediate annotations, (3) designing RoboInter-VQA to evaluate VLMs, (4) demonstrating strong performance gains.
>
> Below, we address the concerns in detail.
>
> ---
>
> **`1. Analysis of inference speed of RoboInter-VLA`**
> > W-1: While a plan-then-execute framework improves reasoning and interpretability by generating intermediate representations, it inherently hurts the latency of robotic systems. The paper does not provide any analysis regarding the inference speed of RoboInter-VLA compared with other VLA models. This could improve the understanding of the trade-off between reasoning capabilities and real-time performance in robotic systems.
>
> Thanks for raising this point. We agree that balancing reasoning capability and real-time performance **is still an active and central challenge in robotic system research**, especially for frameworks that generate intermediate plans. Follow the suggestion, we provide **a detailed analysis regarding the inference speed below** (revised in `Appendix A.5.1`):
>
> As **our primary objective is to study how our large-scale intermediate representation dataset influence VLA learning**; thus, our **open-loop evaluations did not emphasize speed**. However, for **closed-loop deployment**, we applied **standard acceleration strategies** to ensure fair and practical comparisons.
>
> **(1) Latency comparison on open-loop experiments**
>
> We compare RoboInter-VLA against **VLA-OS [1]**, a recent model that **also includes a vanilla action-only variant and an explicit reasoning variant (VLA-OS-I-Explicit)**. An detailed latency comparison:
>
> | Model               | Model Size | Reasoning Tokens | Latency    | Open-loop Score |
> |---------------------|------------|------------------|------------|------------------|
> | VLA-OS (Vanilla)     | 3B         | 0                | 0.1647s    | 0.3035           |
> | VLA-OS-I-Explicit    | 3B         | 64               | 2.2274s    | 0.3290           |
> | Vanilla             | 3B         | 0                | 0.1309s    | 0.3086           |
> | RoboInter-IC-E2E    | 3B         | 64               | 0.1309s    | 0.3218           |
> | RoboInter-EC-E2E    | 3B         | 64               | 2.2187s    | 0.3340           |
> | RoboInter-Modular   | 3B+3B      | 64               | 2.3594s    | 0.3543           |
>
> Across both frameworks (VLA-OS and RoboInter-VLA), we observe consistent trends:
> - **Explicit reasoning substantially improves open-loop performance**, VLA-OS (0.3035 → 0.3290) and RoboInter-VLA (0.3086 → 0.3340).
> - **RoboInter-VLA is faster than VLA-OS in both w/o reasoning or w/ reasoning settings**, RoboInter-IC-E2E v.s. VLA-OS (0.1309s v.s. 0.1647s), RoboInter-EC-E2E v.s. VLA-OS-I-Explicit (2.2187s v.s. 2.2274s).
> - **Performance improvement comes with increased latency**, primarily caused by autoregressive generation of CoT tokens. It also **indicates this phenomenon shares by most planner-to-executor architectures**.
>
> **(2) Real-world comparison with acceleration strategies**
>
> For real-world deployment, we further **implemented practical acceleration strategies**: **textual caching and chunked action execution for EC-E2E** (as described in `Appendix A.4.1`), and **asynchronous dual-frequency execution for the Modular variant**. The measured speeds are:
>
> | Model                            | Speed     | Latency   | ID Score | OOD Score |
> |----------------------------------|-----------|-----------|----------|-----------|
> | OpenVLA                          | ~3.8 Hz   | 263.2 ms  | 45.0     | 23.3      |
> | Vanilla                          | ~7.64 Hz  | 130.9 ms  | 65.0     | 38.3      |
> | RoboInter-IC-E2E                 | ~7.64 Hz  | 130.9 ms  | 76.7     | 58.3      |
> | RoboInter-EC-E2E (cache+chunk)   | ~2.56 Hz  | 390.6 ms  | 68.3     | 60.0      |
> | RoboInter-Modular (asynchronous) | ~6.92 Hz  | 144.5 ms  | 73.3     | 60.0      |
>
> In practical deployment, **all of our models (IC-E2E, EC-E2E, and Modular) maintain a reasonable execution speed** while consistently **delivering superior performance on both in-domain (ID) and out-of-domain (OOD) tasks**.
>
> **(3) Overall analysis**
>
> **Explicit intermediate reasoning yields stronger performance and generalization at the cost of additional latency in open-loop testing. In real-world deployment, practical optimization strategies used in RoboInter-VLA can mitigate these latency issues and allows the models to maintain strong performance.**
>
> [1] Gao C, Liu Z, et al. VLA-OS: Structuring and Dissecting Planning Representations and Paradigms in Vision-Language-Action Models.
>
> **`2. Dataset Release`**
> > Q-1: Will the RoboInter-VQA dataset be made publicly available?
>
> Yes. Our intermediate representation dataset will be publicly released as soon as possible and made openly accessible.

---

### Official Review · Reviewer_BJ8z · 2025-10-31

**Soundness:** 3
**Presentation:** 3
**Contribution:** 3
**Rating:** 6
**Confidence:** 4

**Summary:**

This paper introduces the RoboInter manipulation suite, which includes RoboInter-Tool, a lightweigh tGUI for semi-automatic
 per-frame annotation of embodied videos,and RoboInter-Data,a human-verified dataset with over 200k episodes across 571diverse scenes. Based on them, This paper also introduces a RoboInter-VQA benchmark and there 'plan-then-execute' framework (IC-E2E / EC-E2E / Modular). Experiments show the pre-trained Planner improves open- and closed-loop generalization.

**Strengths:**

1. The introduced RoboInter-Data is large-scale, including 200k episodes, 91M frames, 571 scenes.

2. This paper offers rich intermediate representation categories (sub-tasks, skills, affordances, boxes, traces, contact frames, etc.) during constructing RoboInter-VQA. These representations are critical for improving the abilities (like spatial intelligence) of embodied brain and explaination during bulding VLA systems.

3. This paper also introduces a full toolchain (RoboInter-Tool): an open-source GUI integrating SAM-2 and GPT-4o for semi-automatic segmentation, tracking and language annotation, reducing labeling cost.

4. Based on these rich intermediate representation, this paper also develop three flexible training framework (IC,EC,Module).

5. Extensive are conducted in multimodal benchmarks, RoboInter-VQA, and real-world tasks.

**Weaknesses:**

1. The paper does not introduce architectural or training-paradigm innovations for vision-language-action (VLA) models. Its contributions mainly lie in demonstrating that the proposed intermediate representations can be effective within existing VLA and plan-then-execute frameworks.

2. While the paper mentions prior works that employ explicit chain-of-thought (EC) representations (e.g., ECoT), it does not provide sufficient discussion or comparative analysis with these conceptually related methods. This omission makes it difficult to position RoboInter’s contributions relative to previous efforts.

3. In most real-world experiments, the IC variant performs better than the EC one, raising uncertainty about the practical utility of explicit intermediates. A more systematic analysis of when and how different forms of intermediate representation benefit VLA learning would strengthen the paper.

**Questions:**

Please see the weaknesses.

---

> ### Author Response · Authors · 2025-11-23
> **Rebuttal by Authors (1/2)**
>
> We thank the reviewer for the clear summary and for highlighting the strengths of our work, including:  (1) a lightweight semi-automatic annotation tool, (2) a large-scale human-verified dataset, (3) RoboInter-VQA to evaluate embodied spatial–temporal reasoning, (4) developing flexible plan-then-execute frameworks.
>
> Below, we address the concerns in detail.
>
> ---
>
> **`1. Contributions and Novelty`**
> > W-1:  The paper does not introduce architectural or training-paradigm innovations for vision-language-action (VLA) models. Its contributions mainly lie in demonstrating that the proposed intermediate representations can be effective within existing VLA and plan-then-execute frameworks.
>
> We appreciate the reviewer's observation. **Rather than** *introducing architectural/training-paradigm innovations or merely demonstrating the efficiency of intermediate representations as explored in prior work*, our primary contribution lies in **addressing a critical and widely acknowledged bottleneck**  in manipulation: **the absence of large-scale, high-quality intermediate representations dataset**.
>
> The **limited generalization observed in current VLAs** arises **less from architectural limitations** and far **more from the lack of data and structured supervision** to guide perception and planning. Our work fill this gap by **providing rich, human-verified intermediate labels at a scale not available in prior datasets**.
>
> **The target of proposed standard model variants** (IC-E2E, EC-E2E, Modular) is to clearly **evaluate the impact** of intermediate representations data. The classic plan-then-execute formulations allow us to **isolate and demonstrate the effectiveness of our proposed dataset** without conflating them with architectural innovations.
>
> Improving model architectures and training paradigms could further benefit from the structured supervision we provide, and we plan to explore this direction in future work.
>
> ---
>
> **`2. More comparative analysis with related methods`**
> > W-2:  While the paper mentions prior works that employ explicit chain-of-thought (EC) representations (e.g., ECoT), it does not provide sufficient discussion or comparative analysis with these conceptually related methods. This omission makes it difficult to position RoboInter's contributions relative to previous efforts.
>
> We would like to clarify that the **core contribution of our work is the largest and most diverse intermediate representation dataset for robotic manipulation, together with extensive applications and evaluations**. In response to this contribution, we have provided detailed discussion and comparative analysis in the `Introduction`, `Related Works` and `Table.1 `of our main paper.
>
> The **explicit chain-of-thought mechanism (EC)** in our model mainly **servers for the effectiveness demonstration of our data, rather than being presented as an architectural contribution as emphasized in prior works**. Follow the suggestion, we incorporated a discussion of related methods here (revised in `Related Works` of main paper):
> - **ECOT** [1]: the early method to introduce explicit CoT into VLA models, empolying a single VLM to **generate both CoT text tokens and discreate action tokens**. Due to the **heavy autoregressive calculation and rough mixture of action and text generation**, it suffers from **higher inference latency and more suboptimal performance**.
> - **VLA-OS** [2]: a recent method that **investigates different paradigms for integrating explicit planning and action generation** through **per-layer KV-cache interaction.** In contrast, **our work targets the data bottleneck** rather than architectural design. Methodologically, we differ from VLA-OS by using a **pretrained Planner VLM** and a strategy to **aggregate information from all input tokens and generated CoT tokens**, serving as an intuitive and efficient alternative. An additional performance comparasion:
>
> | Model          | Backbone Size | OLS@0.1 | OLS@0.05 | OLS@0.03 | OLS@0.01 | Mean   |
> |----------------|------------|---------|----------|----------|----------|--------|
> | VLA-OS (Vanilla) | 3B         | 0.6180   | 0.3905   | 0.1928   | 0.0129  | 0.3035 |
> | VLA-OS (+CoT)    | 3B         | 0.7260   | 0.4928   | 0.2734   | 0.0200   | 0.3780 |
> | Ours (Vanilla)  | 3B         | 0.6793  | 0.3608   | 0.1753   | 0.0189   | 0.3086 |
> | Ours (+CoT)     | 3B         | 0.7511  | 0.4640   | 0.2705   | 0.0587   | 0.3861 |
>
> Results show **our model maintains better performance over VLA-OS**. Training VLA-OS with our data (i.e., from Vanilla to CoT mode) **yields a >23% performance gain, showing the validity of our data**.
>
> [1] Zawalski M, Chen W, et al. Robotic control via embodied chain-of-thought reasoning.
>
> [2] Gao C, Liu Z, et al. VLA-OS: Structuring and Dissecting Planning Representations and Paradigms in Vision-Language-Action Models.

---

> ### Author Response · Authors · 2025-11-23
> **Rebuttal by Authors (2/2)**
>
> **`3. Systematic analysis of how different forms of representation usage benefit VLA learning`**
> > W-3:  In most real-world experiments, the IC variant performs better than the EC one, raising uncertainty about the practical utility of explicit intermediates. A more systematic analysis of when and how different forms of intermediate representation benefit VLA learning would strengthen the paper.
>
> Thanks for the thoughtful and constructive comment. We provide a more systematic and clarified analysis below.
>
> **(1) The practical utility of explicit intermediates are certain**
>
> The real-world utility of explicit intermediates (EC-E2E) become certain once viewed from **the perspective of in-distribution (ID) vs. out-of-distribution (OOD) generalization**.
>
> | Model | Open-loop ↑| ID ↑| OOD ↑| ID to OOD Drop ↓|
> |-------|-----------|------|------|----------------|
> | Vanilla | 0.3086 | 65.0 | 38.3 | 26.7 |
> | IC-E2E | 0.3218 | **77.3** | 58.3 | 19.0 |
> | EC-E2E | **0.3340** | 68.3 | **60.0** | **8.3** |
>
> **In OOD, EC-E2E outperforms IC-E2E**, such as *Object Collecting* (60.0% vs. 53.3%) and *Cup Stacking* (73.3% vs. 60.0%), and shows a much **smaller ID→OOD degradation (8.3% vs. 19.0%)**, indicating **stronger generalization under distribution shift** than IC-E2E.
>
> The same trend also holds **in open-loop testing, which behaves similarly to an OOD scenario** (as training scenes and evaluation scenes are strictly separated), where **EC-E2E consistently performs better than IC-E2E** (0.3340 vs. 0.3218).
>
> These results reveal: **IC-E2E outperforms EC-E2E on ID testing**, while **EC-E2E is stronger on OOD generalization**. This reflects **two distinct targets** in embodied control: **accurate action fitting v.s. generalization for out-of-distribution scenes**.
>
>
> **(2) Why do IC and EC behave differently?**
>
> The distinction arises from their **training objectives**.
> - **IC-E2E focuses exclusively on action prediction**. This makes it **better at fitting the ID distribution** and leveraging the strong prior of the pretrained VLM planner.
> - **EC-E2E**, in contrast, **jointly predicts explicit intermediates and actions**. This multimodal supervision **increases learning difficulty and introduces potential modality interference**, which can slightly reduce ID accuracy. However, the **explicit reasoning structure improves task abstraction, yielding better OOD generalization**.
>
>
> **(3) Additional validation experiments**
>
> We ran **two complementary real-world experiments**: **(1) Tool Inserting, an ID-oriented precision control task**, where accurate action modeling is essential; **(2) Object Sorting, an OOD-oriented generalization task** involving diverse object-container combinations via changing instructions. We have provided more details in `Appendix A.4.2` of revised paper.
>
> | Model | Tool Inserting (ID-1) | Tool Inserting (ID-2) | Object Sorting (OOD-1) | Object Sorting (OOD-2) |
> |--------|----------------------|------------------------|-------------------------|-------------------------|
> | π0 | 26.7 | 0.0 | 46.7 | 33.3 |
> | Vanilla | 33.3 | 26.7 | 66.7 | 26.7 |
> | RoboInter-IC-E2E | **53.3** | **40.0** | 66.7 | 40.0 |
> | RoboInter-EC-E2E | 46.7 | 20.0 | **80.0** | **53.3** |
>
> Across all tasks, each variant shows clear, complementary strengths:  **IC-E2E excels in ID-oriented action fitting**; **EC-E2E offers more stable and generalizable OOD performance**, owing to its explicit intermediate reasoning. Overall, **these results show that explicit intermediates (EC) remain highly useful, particularly where generalization is required**.

---

> > ### Comment · Reviewer_BJ8z · 2025-11-25
> > **Thanks for Rebuttal**
> >
> > My main concerns have been resolved, and I appreciate the authors’ effort in the revisions. I will raise my score.

---

> ### Author Response · Authors · 2025-11-26
> **Response to Reviewer BJ8z**
>
> Dear Reviewer BJ8z,
>
> We are happy to hear that our response addressed your concerns! Also, we appreciate your support of our work and your raising the score from 6 to 8. If you have any further questions or suggestions, please do not hesitate to let us know.
>
> Best regards,
> Authors

---

### Official Review · Reviewer_ryjd · 2025-11-01

**Soundness:** 4
**Presentation:** 3
**Contribution:** 3
**Rating:** 6
**Confidence:** 3

**Summary:**

The paper presents the RoboInter Manipulation Suite, which operationalizes a plan-then-execute paradigm for robotic manipulation. It first introduces RoboInter-Tool for semi-automatic, per-frame annotation, and uses it to build RoboInter-Data—a large-scale, scene-diverse dataset that includes object and gripper bounding boxes, contact frames, trajectories, affordances, atomic skills, and sub-tasks. Based on these annotations, the authors design RoboInter-VQA, which turns intermediate representations into concrete task types to train and evaluate VLMs’ embodied perception and planning. At the model level, the paper proposes RoboInter-VLA with three variants—implicitly conditioned end-to-end (IC-E2E), explicitly conditioned end-to-end (EC-E2E), and a modular Planner→Executor setup—and bridges planning and control with a composable intermediate “chain-of-thought” (F-CoT) in text or visual form. Experiments show that planners trained on the proposed data and VQA substantially outperform base and embodied VLMs on Where2Place, RoboRefIt, RoboVQA, and the RefCOCO family; in both open-loop (OLS) and real-world closed-loop (Franka, ID/OOD) evaluations, explicit intermediate supervision and modular designs yield faster convergence and stronger generalization.

**Strengths:**

This paper makes important systematic contributions to the field of robotic manipulation. Originality: It represents the first large-scale systematic effort to address the data scarcity problem of intermediate representations in robotic manipulation, with RoboInter-Tool providing an innovative semi-automatic annotation solution. Quality: High-quality annotations for 200,000 videos are ensured through human verification and multi-round cross-validation, with comprehensive experimental design covering spatial/temporal understanding generation tasks as well as open-loop/closed-loop evaluation, achieving significant improvements across multiple benchmarks. Clarity: Experimental settings and evaluation metrics are clearly defined, and ablation studies help understand the contributions of different components. Significance: This work addresses a key bottleneck constraining the development of the "plan-then-execute" paradigm, provides valuable data resources and tools for the robotic learning community, and demonstrates good generalization capabilities in real-world experiments.

**Weaknesses:**

The accuracy verification of camera parameter estimation and 2D-3D projection is insufficient, lacking quantitative analysis of inter-annotator consistency and failure case studies. Meanwhile, current experiments are primarily based on the Franka robotic platform, lacking cross-platform validation and generalization capability evaluation in more complex scenarios, with limited scale and diversity in real-world experiments. The systematic ablation studies on F-CoT design and various intermediate representations are inadequate. The time costs and economic overhead of human-in-the-loop annotation have not been quantitatively analyzed and require more detailed evaluation.

**Questions:**

Technical Implementation Details**: Could you provide more detailed analysis of camera parameter estimation accuracy and 2D-3D projection errors? What is the quantitative inter-annotator agreement (e.g., Cohen's kappa) for different annotation types?
What are the specific time costs and economic overhead for human-in-the-loop annotation?
Given that experiments primarily use Franka robots, how would you validate generalization to other robotic platforms ? Do you have plans for cross-embodiment evaluation?

---

> ### Author Response · Authors · 2025-11-23
> **Rebuttal by Authors (1/4)**
>
> We thank the reviewer for the clear summary and for highlighting the contributions of RoboInter, including: (1) a semi-automatic annotation tool, (2) a large-scale and diverse intermediate-representation dataset, (3) designing a VQA benchmark for embodied spatial-temporal reasoning, (4) VLA variants that improve open-loop and real-world performance. Below, we address the concerns in detail.
>
> ---
>
> **`1. Verification and analysis of camera parameter estimation accuracy and 2D-3D projection errors`**
> > W-1: The accuracy verification of camera parameter estimation and 2D-3D projection is insufficient
> Q-1: Could you provide more detailed analysis of camera parameter estimation accuracy and 2D-3D projection errors?
>
> Since ground-truth camera parameter are not always available in our setting, **we cannot evaluate camera parameter estimation directly**. However, the evaluation results can be  **indirectly observed** by **2D reprojection error**, because the success rate of 3D-2D projection process is mainly affected by camera parameter, the **2D reprojection error reflects both camera calibration accuracy and 3D-2D projection quality**.
>
> Specifically, we define a set of corresponding 3D keypoints and **manually annotate a subset of their 2D traces** on the 2D image, which serve as the **pseudo ground truth**. Using the **optimized camera parameters**, we **project the 3D keypoints back onto the image plane** and compute the **per-pixel L2 distance** to the human annotations.
>
> **Table R1. The 2D reprojection errors across seven configurations of the source data**
> |                      | **Error of cfg_1 ↓** | **Error of cfg_2 ↓** | **Error of cfg_3 ↓** | **Error of cfg_4 ↓** | **Error of cfg_5 ↓** | **Error of cfg_6 ↓** | **Error of cfg_7 ↓** |
> |----------------------|--------------|--------------|--------------|--------------|--------------|--------------|--------------|
> | **Val Num frames**          | 1295 | 1357 | 1136 | 1094 | 1017 | 1195 | 1154 |
> | **Before optim.** | 46.25 | 40.63 | 67.35 | 108.21 | 43.68 | 69.71 | 66.24 |
> | **After optim.**  | 5.31 | 5.69 | 10.80 | 11.61 | 5.47 | 6.31 | 6.95 |
>
> As seen in above table, across all configurations, **the optimization procedure reduces the reprojection error by a large margin** (e.g., from 46.25 to 5.31 on cfg_1, and 108.21 to 11.61 on cfg_4). This consistent improvement demonstrates that **our optimization method significantly enhances 3D-2D projection quality**.
>
>
> ---
>
> **`2. Quantitative analysis of inter-annotator consistency`**
> > W-2: lacking quantitative analysis of inter-annotator consistency and failure case studies
> Q-2: What is the quantitative inter-annotator agreement (e.g., Cohen's kappa) for different annotation types
>
>
> Our annotation workflow includes **three rounds of cross-validation** by experienced annotators, **ensuring consistent and high-quality labels**. In response to the reviewer's suggestion, we conducted a **representative inter-annotator agreement study**: 300 samples were randomly selected and independently re-annotated by the original annotators. We report **Cohen's kappa** (only appropriate for categorical tasks) for **16-class primitive skill selection**, **IoU for box-level annotations**, and **MSE for point-level annotations**.
>
>
> **Table R2. Results of inter-annotator agreement, Origin: our original annotations, Ann.1/2: annotator.**
> |                      | **Primitive Skill (Cohen's Kappa ↑)** | **Grounding Box (IoU ↑)** | **Affordance Box (IoU ↑)** | **Contact Points (MSE ↓)** |
> |----------------------|----------------------|------------------|------------------------|------------------------|
> | **Ann.1 vs. Origin** | 0.894                | 94.7%            | 89.7%                  | 0.055                  |
> | **Ann.2 vs. Origin** | 0.881                | 92.1%            | 90.9%                  | 0.089                  |
> | **Ann.1 vs. Ann.2** | 0.885                | 92.9%            | 90.1%                  | 0.071                  |
>
> The results show **consistently strong agreement across annotators**. **Cohen's Kappa values (>0.88) indicate high consistency for primitive skills** (generally, above 0.81 indicates near-perfect agreement), and **IoUs above 89% demonstrate high consistency in region-level annotations**. The **close match between *annotator-origin* and *annotator-annotator* comparisons** further confirms that **annotation inconsistency is minimal** and unlikely to affect downstream performance.
>
> **Failure Case Studies**: mostly arises in **contact-point annotations**, which is expected: contact points can be inherently **ambiguous with non-unique valid solutions** (e.g., initial-touch vs. full-grasp points). Even in these cases **the variance remains small and has limited practical impact**, whereas other annotations show fewer failure cases.

---

> ### Author Response · Authors · 2025-11-23
> **Rebuttal by Authors (2/4)**
>
> **`3. More experiments and cross-platform validation`**
>
> > W-3:  Meanwhile, current experiments are primarily based on the Franka robotic platform, lacking cross-platform validation and generalization capability evaluation in more complex scenarios, with limited scale and diversity in real-world experiments.
> Q-4: Given that experiments primarily use Franka robots, how would you validate generalization to other robotic platforms
> Q-5: Do you have plans for cross-embodiment evaluation
>
> We had conducted cross-embodiment evaluations on both the WidowX and Google Robot simulation platforms in origin submission (`Table.19`). Although most of our real-world experiments were performed on the Franka platform due to hardware limitation, our **dataset and model design are capable to support multi-embodiment generalization**. We summarize more analysis and **additional cross-platform experiments below** (revised in `Appendix A.6`):
>
> **(1) Both our data and model are inherently cross-platform**
>
> Our **RoboInter-Data** already **contains demonstrations from multiple robot embodiments** across 570+ scene configurations. The **annotation pipeline** can **scale efficiently to new embodiments**, and **RoboInter-VLA normalizes observations/actions/states across platforms** and enable **platform-agnostic training**.
>
> **(2) Open-loop cross-platform results**
>
> Actually, our **open-loop validation already includes multiple embodiments**, due to our multiple-platform data source of training and validation. RoboInter-VLA variants consistently outperforms the Vanilla baseline across all platforms:
>
> **Table R3. Detailed results of open-loop score across platforms.**
> | Model               | platform_1 | platform_2 | platform_3 | platform_4 | platform_5 | mean      |
> |---------------------|------------|------------|------------|------------|------------|-----------|
> |            | **Franka+Panda**    | **UR5+Robotiq** | **Flexiv+AG-95** | **Kuka+Robotiq** | **Franka+Robotiq** |           |
> |                     |            |            |            |            |            |           |
> | Vanilla             | 0.6756     | 0.9659     | 0.7646     | 0.7476     | 0.3608     | 0.7029    |
> | RoboInter-IC-E2E    | 0.7149     | 0.9929   | *0.7786*   | 0.7564     | 0.3810     | 0.7248   |
> | RoboInter-EC-E2E    | 0.8098     | 0.9876     | 0.7723     | *0.7748*   | 0.3930     | 0.7475    |
> | RoboInter-Modular   | *0.8572*   | **0.9962** | 0.7743     | 0.7609     | *0.4133*   | *0.7604* |
> | Oracle+Executor     | **0.8828** | *0.9942*   | **0.7809** | **0.7848** | **0.4640** | **0.7813** |
>
> **(3) Close-loop results in simulation**
>
> For closed-loop evaluation beyond Franka, we list results on **two additional robotic embodiments (*Google Robot* and *WidowX Robot*)** below.
>
> **Table R4: Google Robot results (Variant Aggregation) on SimplerEnv**
> | Model | Open/Close Drawer | Put in Drawer | Pick Coke Can | Move Near | **Avg** |
> |-------|-------------------|----------------|----------------|-----------|---------|
> | π0 | 25.6 | 20.5 | 75.2 | 63.7 | 46.3 |
> | Magma | 59.0 | 24.0 | 68.6 | 78.5 | 57.5 |
> | Vanilla | 27.5 | 48.6 | 86.2 | 85.0 | 61.8 |
> | **RoboInter-IC-E2E** | 55.3 | **58.3** | **87.7** | 84.0 | **71.3** |
>
> **Table R5. WidowX Robot Results (Visual Matching) on SimplerEnv**
> | Model | Put Spoon | Put Carrot | Stack Blocks | Put Eggplant | **Avg** |
> |-------|------------|-------------|---------------|----------------|---------|
> | π0 | 29.1 | 0 | 16.6 | 62.5 | 27.1 |
> | Magma | 37.5 | 29.2 | 20.8 | 91.7 | 44.8 |
> | Vanilla | 41.7 | 37.5 | 12.5 | 95.8 | 46.9 |
> | **RoboInter-IC-E2E** | **54.2** | 25.0 | **29.2** | 91.7 | **50.0** |
>
> **(4) Additional close-loop results in real worlds**
>
> To further evaluate the real-world generalization, we utilize models pretrained with Bridgev2 dataset and **conduct zero-shot real-world tests on the WidowX Robot** (details in `Appendix A.6.3`):
>
> **Table R6. Real-world WidowX Robot results**
> | Model | Pick Spoon | Pick Pikachu | Put Cup on Plate | Put Pikachu on Plate |
> |-------|------------|-----------|------------------|----------------|
> |       | In-Distribution  |  Out-of-Distribution | In-Distribution | Out-of-Distribution |
> | π0 | 33.3 | 6.7 | 20.0 | 0.0 |
> | Vanilla | **60.0** | 26.7 | 46.7 | 13.3 |
> | **RoboInter-IC-E2E** | **60.0** | **40.0** | **53.3** | **33.3** |
>
> **(5) Conclusion and future plans**
>
> Experiments above show that RoboInter contains **large-scale cross-platform data**, a **scalable annotation pipeline**, and a **RoboInter-VLA framework that exhibits strong cross-platform generalization** in both **open-loop and closed-loop** settings, across **simulation and real-world robots**.
>
> We are actively extending our work to additional single-arm and dual-arm platforms; while these require further hardware integration and setup, our pipeline has already been designed for efficient scaling. **We expect RoboInter to continue enabling broader research on generalizable robotic manipulation**.

---

> ### Author Response · Authors · 2025-11-23
> **Rebuttal by Authors (3/4)**
>
> **`4. More ablation studies on F-CoT`**
>
> > W-3:  The systematic ablation studies on F-CoT design and various intermediate representations are inadequate.
>
> Our initial submission has reported corresponding ablations in `Table 4` and `Appendix Tables 15`. Below we **provide a more systematic ablations to clarify the role of F-CoT and the contribution of different intermediate representations**, these content has been revised to `Table 5 and section 5.2` (main paper) and `Appendix A.5.3`.
>
> **(1) Different F-CoT designs**
>
> F-CoT serves as a ***flexible carrier* that composes multiple intermediate representations** from RoboInter-Data. We evaluate two forms:
> - **Textual F-CoT**: Intermediate representations are serialized into autoregressive text tokens.
> - **Visual F-CoT**: Intermediate representations are drawn as visual prompts on additional input images.
>
> **Table.R7: Ablation results of different F-CoT designs**
> | Model           | OLS@0.1 | OLS@0.05 | OLS@0.03 | OLS@0.01 | Mean |
> |-----------------|---------|----------|----------|----------|----------|
> | Vanilla         | 0.6793  | 0.3608   | 0.1753   | 0.0189   | 0.3086   |
> | Visual F-CoT   | 0.7056  | 0.4029   | 0.2240   | 0.0430   | 0.3439   |
> | Textual F-CoT  | **0.7124**  | **0.4133**   | **0.2332**   | **0.0584**   | **0.3543** |
>
> **Textual F-CoT consistently performs best**, indicating that **structured textual reasoning is a more stable and expressive interface** for composing heterogeneous intermediate representations.
>
>
> **(2) Combinations of intermediate representations**
>
> RoboInter provides rich supervision, but not all representations must be used simultaneously, especially given the autoregressive cost of generating F-CoT tokens. We therefore **ablate combinations of intermediate representations** within the open-loop Oracle+Executor setting:
>
> **Table.R8: Ablation results of different representations combinations**
> |                         | OLS@0.1 | OLS@0.05 | OLS@0.03 | OLS@0.01 | Mean |
> | ----------------------- | ------- | ------ | ----------- | ------ | ----------- |
> | Vanilla | 0.6793    | 0.3608   | 0.1753   | 0.0189| 0.3086|
> | + Subtask      | 0.6965    | 0.3676   | 0.177|  0.0171| 0.3146|
> | +S. + Primitive skill     | 0.6983    | 0.3681   | 0.1779 | 0.0194 | 0.3159|
> | +S. +P. + Target Box      | 0.7025    | 0.3849   | 0.1988 | 0.0294 | 0.3289|
> | +S. +P. +T.B. + Gripper Box      | 0.7212 | 0.4032   | 0.2048 | 0.0272 | 0.3391|
> | +S. +P. +T.B. +G.B. + Affordance | 0.7245 | 0.4083   | 0.2114 | 0.0297 | 0.3435|
> | +S. +P. +T.B. +G.B. +Aff. + Trace      | 0.7511    | 0.464   | 0.2705| 0.0587 |0.3861|
>
> Subtask and Primitive Skill offer only small improvements, which is expected: **coarse-grained signals** offer **limited constraints for actions** during execution. Spatially grounded representations (Target Box, Gripper Box, Affordance) **contribute substantially more with fine-grained cues**. The **largest gain comes from Trace**, which provides **dense, fine-grained temporal guidance**. Overall, **more fine-grained intermediate supervision lead to stronger performance** in our open-loop evaluations.
>
>
> **(3) Single intermediate representation**
>
> We further ablate how each intermediate representation individually affects performance below:
>
> **Table.R9: Ablation results of single representations**
> | Method        | OLS@0.1 | OLS@0.05 | OLS@0.03 | OLS@0.01 | Mean |
> |----------------|---------|----------|----------|----------|----------|
> | Vanilla         | 0.6793 | 0.3608 | 0.1753 | 0.0189 | 0.3086 |
> | + Subtask       | 0.6965 | 0.3676 | 0.1770 | 0.0171 | 0.3146 |
> | + Primitive skill    | 0.6925 | 0.3658 | 0.1793 | 0.0156 | 0.3133 |
> | + Target Box    | 0.7001 | 0.3793 | 0.1879 | 0.0202 | 0.3218 |
> | + Affordance    | 0.7054 | 0.3958 | 0.2012 | 0.0263 | 0.3322 |
>
> The trend is consistent: **coarse-grained representations (Subtask, Primitive skill) give modest gains**, whereas **spatially precise cues (Target Box, Affordance) provide much stronger improvements**.

---

> ### Author Response · Authors · 2025-11-23
> **Rebuttal by Authors (4/4)**
>
> **`5. Detailed time costs and economic overhead`**
> > W-5:  The time costs and economic overhead of human-in-the-loop annotation have not been quantitatively analyzed and require more detailed evaluation.
> Q-3: What are the specific time costs and economic overhead for human-in-the-loop annotation?
>
> Below, we provide **the quantitative analysis of the time, labor, and economic overhead** of our human-in-the-loop (HITL) annotation pipeline.
>
> **(1) Time and labor breakdown**
>
> - Data Scale: 90k language annotations, 230k image annotations, plus full three-rounds cross-validation.
> - Workforce: 30 ~ 40 trained annotators.
> - Workload: 4 hours/day per annotator to maintain consistency and quality.
> - Duration: ~3 months of annotation + ~1 month of dedicated quality assurance.
>
> **(2) Economic overhead and efficiency gains**
>
> Our custom **RoboInter-Tools** can provide automated segmentation proposals and real-time visual feedback, which significantly reduced manual workload and cost.
>
> - Without RoboInter-Tool: Fully manual annotation of comparable scale is estimated at more than $102,669.
> - Actual Cost with RoboInter-Tools: **only $33,330, including all labor, QA, and management**.
> - Overall Savings: **~68% cost reduction, while maintaining high-fidelity ground truth (>95% final accuracy)**.

---

> > ### Comment · Reviewer_ryjd · 2025-11-26
> >
> > My core concerns have been resolved and the cross-platform experiments are helpful. I will raise my score

---

> ### Author Response · Authors · 2025-11-26
> **Response to Reviewer ryjd**
>
> Dear Reviewer ryjd,
>
> We are pleased to know that our reply resolved your concerns. We appreciate your support of our work and your raising the score from 6 to 8. If you have any further feedback, we would be glad to address it.
>
> Best regards,
> Authors

---

### Author Response · Authors · 2025-11-23
**General Response**

**We sincerely thank all reviewers (*ryjd, BJ8z, v8Xb, hDxo*) for the thoughtful and constructive feedback**. We appreciate the recognition of RoboInter as a systematic contribution that jointly advances **data**, **benchmarking**, and **modeling** for the plan–then–execute VLA paradigm. Reviewers **highlighted the importance of:**

**1. Tackling the data scarcity of intermediate representations** in robotic manipulation and **providing a large-scale, human-verified annotation data** (*ryjd, BJ8z, v8Xb, hDxo*);

**2. Designing RoboInter-VQA to probe spatial and temporal embodied reasoning of VLM** beyond standard VQA benchmarks (*ryjd, BJ8z, v8Xb, hDxo*);

**3. Demonstrating that explicit intermediate supervision and plan–then–execute designs improve generalization** in both open-loop and real-world closed-loop evaluations (*ryjd, v8Xb, hDxo*);

**4. Providing detailed experiments and analyses** across multimodal benchmarks, RoboInter-VQA, open-loop testing, and real-world tasks. (*ryjd, BJ8z, v8Xb, hDxo*).

----

We are encouraged that reviewers found the problem well-motivated and the results compelling. In response to the raised concerns, we have added new analyses, ablations, and experiments to further clarify the technical choices and strengthen the empirical validation. **Blow, we provide a summary of key revisions and additions:**

**1. Camera & annotation quality:** Added 2D reprojection error analysis and inter-annotator agreement. (*ryjd*)

**2. Cross-embodiment and additional real-world experiments:** Expanded open-/closed-loop results to more robot platforms and additional real-world tasks. (*ryjd, BJ8z, hDxo*)

**3. F-CoT & representation ablations:** Systematically ablated F-CoT designs (text vs. visual) and individual/combined intermediate representations. (*ryjd*, *hDxo*)

**4. More CoT baselines:** Strengthened comparison to ECoT, VLA-OS, and trained VLA-OS on RoboInter-Data to highlight dataset benefits. (*BJ8z, hDxo*)

**5.Latency & cost analysis:** Added inference-speed comparisons and human-in-the-loop cost analysis. (*ryjd*, *v8Xb*, *hDxo*)

We appreciate the reviewers’ feedback and remain committed to improving the work. All revisions are marked in *blue color* in the updated paper. The following sections address each reviewer’s comments in detail.

---

### Author Response · Authors · 2025-12-01
**A Summary of Our Rebuttal and Reviewers' Corresponding Response**

We sincerely thank all Reviewers, ACs, and SACs for their constructive feedback and engagement during the discussion. To clearly summarize our interactions with Reviewers and how each concern was addressed, we outline the main points below:

-----
## **Reviewer ryjd**
**Issues raised**

(1) Quantitative verification of camera calibration accuracy and 2D-3D projection accuracy; (2) Inter-annotator consistency; (3) How to validate the generalize across different robot embodiments; (4) More ablation studies on F-CoT and intermediate representations; (5) How about annotation cost.

**Our rebuttal**

(1) Comprehensive camera calibration and 2D reprojection error analysis; (2) Cohen’s kappa / IoU / MSE inter-annotator agreement statistics and failure-case study; (3) Cross-platform analysis and experiments (open-loop results on multiple industrial arms; closed-loop results on SimplerEnv and WidowX Robot); (4) Ablations on F-CoT designs, combinations of intermediate representations, and single representations; (5) Quantitative time and cost breakdown of human-in-the-loop annotation.

**Reviewer's response**

"My core concerns have been resolved, and the cross-platform experiments are helpful. **I will raise my score.**" **(26 Nov 2025, 15:29)**

-----
## **Reviewer BJ8z**

**Issues raised**

(1) No architectural or training-paradigm innovations; (2) Additional discussion or comparative analysis with explicit chain-of-thought methods; (3) Systematic analysis of the utility of explicit intermediates and how IC/EC variants differently benefit VLA learning.

**Our rebuttal**

(1) Comprehensive clarification of our motivation and contributions on the intermediate representations dataset; (2) Extended comparative analysis with ECoT / VLA-OS and additional comparison experiments; (3) Analysis of the utility of explicit intermediates (from the perspective of ID vs. OOD), explanations of why IC/EC variants differently benefit (training objectives), and additional real-world experiments for validation.

**Reviewer's response**

"My main concerns have been resolved, and I appreciate the authors’ effort in the revisions. **I will raise my score.**" **(25 Nov 2025, 22:42)**

-----
## **Reviewer v8Xb**

**Issues raised**

(1) Requested analysis of inference latency; (2) Question on dataset release.

**Our rebuttal**

(1) Detailed latency analysis and comparison tables for RoboInter-VLA vs. VLA-OS, and real-world execution speeds with acceleration strategies (caching, chunking, asynchronous execution); (2) Confirmation of public dataset release.

-----
## **Reviewer hDxo**

**Issues raised**

(1) Evaluation of additional baselines trained on our proposed dataset; (2) Better analysis of which intermediate representations matter most and how they are used; (3) Inference-time analysis; (4) Why lacking real-world validation of the Modular variant.

**Our rebuttal**

(1) Added full comparison with VLA-OS (Vanilla & CoT); (2) Fine-grained ablations for subtask/primitive/box/affordance cues and clarification of intermediate representation usage; (3) Detailed latency analysis and comparison tables for RoboInter-VLA vs. VLA-OS, plus real-world execution speeds with acceleration strategies (caching, chunking, asynchronous execution); (4) Clarification of focusing on end-to-end models, and extra real-world closed-loop results of the Modular variant.

**Reviewer's response**

"Thanks for the response. It has resolved my concerns. **I will maintain my rating.**" **(28 Nov 2025, 16:04)**

---

### Meta-Review · Area_Chair_KmLW · 2026-01-08

**Summary:**

This paper presents RoboInter, a valuable manipulation suite combining a semi-automatic annotation tool, a large-scale human-verified dataset with rich intermediate representations, a VQA benchmark for embodied spatial–temporal reasoning, and plan-then-execute VLA variants that leverage intermediate supervision to improve planning and control. Reviewers broadly agree that the main contribution is the high-impact resource (tool+data+benchmark) and the strong, clearly reported empirical gains across multiple benchmarks and real-robot evaluations.

After the rebuttal, borderline reviewers raise their scores. Overall, the work meaningfully advances intermediate-representation supervision for manipulation and is likely to be useful to the community; remaining limitations (broader real-world scale and further embodiment coverage) are reasonable future work.

**Reviewer Concerns:**

Broader real-world scale and further embodiment coverage are the limitations.

**Reviewer Scores:**

Reviewers broadly agree that the main contribution is the high-impact resource (tool+data+benchmark) and the strong, clearly reported empirical gains across multiple benchmarks and real-robot evaluations. The ratings are 6,8,8,6.

---

### Decision · Program_Chairs · 2026-01-26

Accept (Poster)